# Setd2 supports GATA3[+]ST2[+] thymic-derived Treg cells and suppresses intestinal inflammation

Zhaoyun Ding[1,11], Ting Cai[1,2,11], Jupei Tang[1], Hanxiao Sun [3], Xinyi Qi[4], Yunpeng Zhang[5], Yan Ji[1], Liyun Yuan[6], Huidan Chang[6], Yanhui Ma [7], Hong Zhou[8], Li Li [9,10,12] ✉, Huiming Sheng [3,12] ✉ & Ju Qiu [1,12] ✉

Treg cells acquire distinct transcriptional properties to suppress specific inflammatory responses. Transcription characteristics of Treg cells are regulated by epigenetic modifications, the mechanism of which remains obscure. Here, we report that Setd2, a histone H3K36 methyltransferase, is important for the survival and suppressive function of Treg cells, especially those from the intestine. Setd2 supports GATA3[+]ST2[+] intestinal thymic-derived Treg (tTreg) cells by facilitating the expression and reciprocal relationship of GATA3 and ST2 in tTreg cells. IL-33 preferentially boosts Th2 cells rather than GATA3[+] Treg cells in *Foxp3^{Cre-YFP}Setd2^{flox/flox}* mice, corroborating the constraint of Th2 responses by Setd2 expression in Treg cells. SETD2 sustains GATA3 expression in human Treg cells, and SETD2 expression is increased in Treg cells from human colorectal cancer tissues. Epigenetically, Setd2 regulates the transcription of target genes (including *Il1rl1*) by modulating the activity of promoters and intragenic enhancers where H3K36me3 is typically deposited. Our findings provide mechanistic insights into the regulation of Treg cells and intestinal immunity by Setd2.

To prevent inflammation-induced tissue damage and autoimmunity, Foxp3[+] T regulatory (Treg) cells are required to repress overt activation of CD4[+] conventional T cell subsets, including T helper 1 (Th1), Th2, Th17, and T follicular helper (Tfh) cells[1,2]. Differentiated from naïve CD4[+] T cells under distinct cytokine milieus, CD4[+] conventional T cell subsets are regulated by master transcription factors, which are important for them to exhibit immunodefensive functions against the invasion of various pathogens[3]. Comparatively, Treg cells acquire the expression of master transcription factors for their specialized functions to prohibit different Th-cell responses[4,5]. Among them, GATA3, as a transcription factor driving type 2 immunity, is fundamental for Treg homeostasis and functions[6–8]. Specific deletion of GATA3, or

[1]CAS Key Laboratory of Tissue Microenvironment and Tumor, Shanghai Institute of Nutrition and Health, University of Chinese Academy of Sciences, Chinese Academy of Sciences, Shanghai 200031, China. [2]Medical Research Center, Guangdong Provincial People's Hospital, Guangdong Academy of Medical Sciences, Guangzhou 510180, China. [3]Department of Laboratory Medicine, Tongren Hospital, Shanghai Jiao Tong University School of Medicine, Shanghai 200336, China. [4]Shanghai Jiao Tong University School of Medicine (SJTUSM), Shanghai 200025, China. [5]Department of General Surgery, Tongren Hospital, Shanghai Jiao Tong University School of Medicine, Shanghai 200336, China. [6]CAS Key Laboratory of Computational Biology, Bio-Med Big Data Center, Shanghai Institute of Nutrition and Health, University of Chinese Academy of Sciences, Chinese Academy of Science, Shanghai 200031, China. [7]Department of Laboratory Medicine, Xin Hua Hospital, Shanghai Jiao Tong University School of Medicine, Shanghai, China. [8]School of Life Science, Anhui Medical University, Hefei 241000, China. [9]State Key Laboratory of Oncogenes and Related Genes, Renji-Med X Clinical Stem Cell Research Center, Ren Ji Hospital, School of Medicine and School of Biomedical Engineering, Shanghai Jiao Tong University, Shanghai 200127, China. [10]Med-X Research Institute, Shanghai Jiao Tong University, Shanghai, China. [11]These authors contributed equally: Zhaoyun Ding and Ting Cai. [12]These authors jointly supervised this work: Li Li, Huiming Sheng and Ju Qiu. ✉e-mail: lil@sjtu.edu.cn; HMSHENG@shsmu.edu.cn; qiuju@sibs.ac.cn

concomitant loss of GATA3 and T-bet, in Treg cells causes inflammation characterized by increased Th1, Th2, and Th17 responses[7–9]. Notably, GATA3 expression in Treg cells is especially important for their tempering of Th2 responses in the intestine and the skin[8,10]. At the molecular level, GATA3 has been shown to interact with Foxp3 to induce the expression of factors modulating Treg functionality, including GATA3 itself, NFAT, and Runx3[8].

Through crosstalk with trillions of microorganisms, intestinal conventional T (Tconv) cells become highly activated, but this activation is efficiently combated by the suppressive function of intestinal Treg cells[11]. Intestinal Treg cells consist of thymic-derived Treg (tTreg) cells marked by Helios and Neuropilin-1 (Nrp1) expression[12,13], and Helios⁻ peripherally induced Treg (pTreg) cells that are converted from conventional T cells under the influence of factors such as TGF-β and retinoic acid from the tissue microenvironment[14]. Intestinal Treg cells have been found to express prominent levels of master transcription factors essential for their suppressive function. A major population of colonic pTreg cells is induced by microbiota and expresses the type 17 transcription factor RORγt[15,16]. Intestinal Treg cells also express higher levels of GATA3 than Treg cells from the spleen or lymph nodes[6]. This is probably because triggers inducing GATA3 expression in Treg cells are more abundant in the intestine, including the T-cell receptor (TCR) signaling activators, IL-2 and IL-33[17–19]. ST2 is the receptor of IL-33 and a downstream target of GATA3[20]. Similar to GATA3, ST2 is expressed at a higher level in intestinal Treg cells than in splenic Treg cells and supports the function of Treg cells[6,19,21]. Expression of both GATA3 and ST2 in Treg cells has been implicated in the pathogenesis of colitis and colon cancer in mice and humans[6,7,19,21–24], suggesting that they are potential diagnostic or therapeutic targets for intestinal inflammatory diseases. Currently, the molecular mechanisms regulating GATA3 and ST2 expression in intestinal Treg subsets are incompletely understood.

Epigenetic modifications, including histone methylation, histone acetylation, and DNA methylation, are intricately orchestrated in Treg cells to imprint the transcriptional properties of Treg cells[25]. Setd2, the only known mammalian histone H3 lysine 36 (H3K36) methyltransferase mediating H3K36me3 modification[26], has been found to be important for the development and function of immune and nonimmune cells[27–30]. Setd2 has been shown to regulate gene/protein expression by affecting DNA methylation[29,31], histone modifications[29,32,33], alternative splicing[34], transcriptional elongation[32,35], and posttranslational modifications[30,36]. The expression and function of Setd2 and H3K36me3 modification in Treg cells have not been explored.

Here, we investigate the molecular regulation of Treg cells by Setd2. We demonstrate that the support of GATA3 expression in Treg cells by Setd2 is conserved in both mice and humans, and Setd2 is essential for Treg cells to specifically suppress Th2 responses. Our data provide mechanistic insights into the regulation of Treg cells by Setd2, which improves the understanding of the cross-regulation of epigenetics and transcriptomics of Treg cells under steady state and during inflammation.

## Results

### Setd2 is essential for Treg cells to repress T cell activation

Setd2 is an H3K36 methyltransferase that is conserved across species. Setd2 was similarly highly expressed by both splenic and intestinal Treg cells (Supplementary Fig. 1a–c), and the level of H3K36me3 was comparable in splenic and intestinal Treg cells (Supplementary Fig. 1d, e). To investigate the function of Setd2 in Treg cells, we crossed *Setd2^flox/flox^* mice with *Foxp3^Cre-YFP^* mice to generate mice with Treg-specific deletion of Setd2 (*Foxp3^Cre-YFP^Setd2^f/f^*). Setd2 expression was sufficiently abrogated in Treg cells of *Foxp3^Cre-YFP^Setd2^f/f^* mice compared with control mice (*Foxp3^Cre-YFP^Setd2^f/+^*), while the expression of Setd2 in CD4⁺ non-Treg cells (conventional T cells, Tconv) was unaffected (Fig. 1a, b and Supplementary Fig. 1a).

We noticed a slight but significant increase in Setd2 expression by Tconv cells compared with Treg cells in *Foxp3^Cre-YFP^Setd2^f/+^* mice (Fig. 1a, b). This was possibly due to one deficient allele of *Setd2* in Treg cells of *Foxp3^Cre-YFP^Setd2^f/+^* mice. The level of H3K36me3 was drastically reduced in the Treg cells but not CD4⁺ Tconv cells from the spleen and intestine of *Foxp3^Cre-YFP^Setd2^f/f^* mice (Fig. 1c and Supplementary Fig. 1f), confirming that Setd2 is the dominant H3K36 methyltransferase mediating H3K36me3 modification in Treg cells.

The aberrant function of Treg cells can cause systemic inflammation. We observed that the proportions of effector/effector memory (CD44⁺CD62L^low^) CD4⁺ Tconv cells and CD4⁻ T cells, the majority of which were CD8⁺ T cells or γδT cells, were significantly higher or tended to be increased in the spleen, inguinal lymph nodes (ILNs), mesenteric lymph nodes (MLNs) and large intestine of *Foxp3^Cre-YFP^Setd2^f/f^* mice (Fig. 1d, e), whereas the percentages of naïve (CD44⁻CD62L^high^) CD4⁺ Tconv cells and CD4⁻ T cells were decreased (Fig. 1f, g). The percentages of central memory (CD44⁺CD62L^high^) CD4⁺ Tconv cells but not CD4⁻ T cells were reduced in the large intestine (Fig. 1h, i). For Treg cells, the percentages of naïve cells were also decreased in the spleen, MLNs, and large intestine, and percentages of central memory cells were decreased in the large intestine, possibly reflecting a loss of the quiescence status of the Treg cells (Fig. 1j–l). We observed comparable absolute numbers of Tconv cells and CD4⁻ T cells in the spleen, ILNs, MLNs, and large intestine from *Foxp3^Cre-YFP^Setd2^f/f^* mice with those from controls (Supplementary Fig. 1g). The data suggest that Setd2-deficient Treg cells possess suppressive function but a reduced capability to inhibit T cell activation in the secondary lymphoid organs (SLOs) and the intestine. The development of Treg cells in the thymus was normal in *Foxp3^Cre-YFP^Setd2^f/f^* mice, as indicated by the similar percentages of thymic Treg cells and two Treg precursor subsets (Foxp3⁻CD25⁺ and Foxp3^low^CD25⁻ cells) among CD4⁺ single-positive T cells (Fig. 1m)[37]. Interestingly, while no difference in the percentages of spleen and ILN Treg cells were found, Treg cells in the MLNs and the intestine were significantly reduced in *Foxp3^Cre-YFP^Setd2^f/f^* mice (Fig. 1n). A trend towards a decrease in numbers of Treg cells was observed in the MLNs and large intestine of *Foxp3^Cre-YFP^Setd2^f/f^* mice although this didn't reach a statistical significance (Supplementary Fig. 1g). Furthermore, we found no difference in percentages or numbers of Treg cells in the liver, lung or visceral adipose tissue from *Foxp3^Cre-YFP^Setd2^f/f^* mice compared with controls (Fig. 1o and Supplementary Fig. 1h). These data suggest that Setd2 is important for the homeostasis of Treg cells in the intestine and gut-associated lymphoid tissues (GALTs).

### Setd2 maintains Treg homeostasis by sustaining Treg survival

The maintenance of Treg cells may be affected by the proinflammatory environment in *Foxp3^Cre-YFP^Setd2^f/f^* mice. Therefore, we generated bone marrow chimeric mice with mixed bone marrow cells from *Foxp3^Cre-YFP^Setd2^f/f^* mice and littermate control mice to place Setd2-deficient and control Treg cells in the same environment (Fig. 2a). The bone marrow chimeric mice set up a system for Treg cells to compete to expand and survive. Strikingly, we found that Treg cells derived from Setd2-deficient Treg cells constituted significantly lower proportions in the spleen, MLNs, and intestine than Treg cells derived from the *Foxp3^Cre-YFP^Setd2^f/+^* donors (Fig. 2b). This was unlikely to be due to decreased Treg proliferation in the absence of Setd2 as was indicated by the similar frequencies of Ki67⁺ cells in Treg cells derived from *Foxp3^Cre-YFP^Setd2^f/f^* and control donors (Supplementary Fig. 2a). Furthermore, increased proportions of Tconv cells derived from *Foxp3^Cre-YFP^Setd2^f/f^* donors, which might happen if Setd2-deficient Treg cells lost Foxp3 expression and converted to Tconv cells, was not observed (Fig. 2c). Next, we analyzed Treg cells from female *Foxp3^Cre-YFP/+^Setd2^f/f^* mice in which Setd2-deficient and Setd2-competent Treg cells could be distinguished by YFP expression in the same host due to random X-chromosome inactivation. We

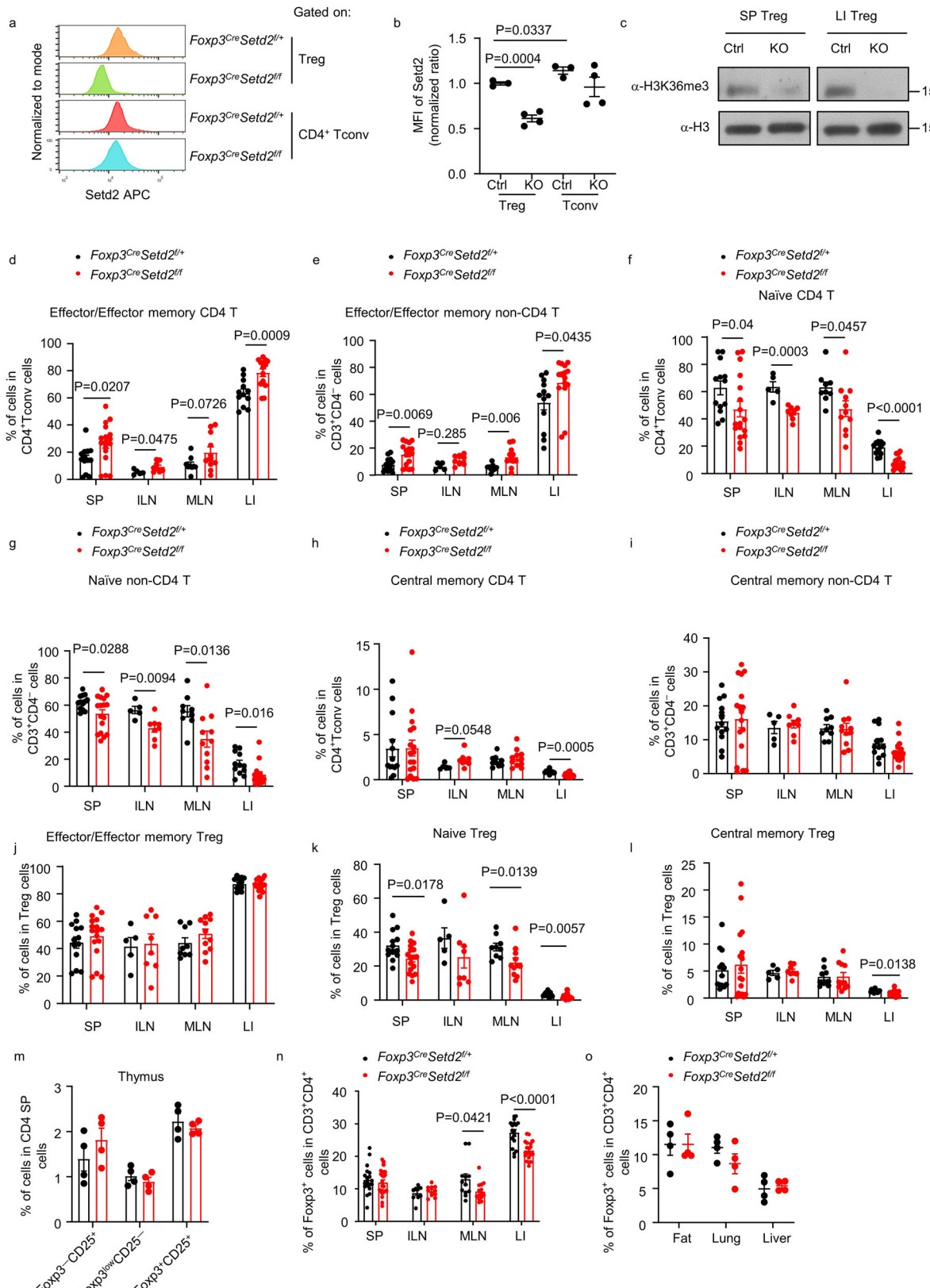

found that YFP+ Treg cells marking *Setd2*-deleted Treg cells constituted significantly less proportions compared with YFP− *Setd2*-competent Treg cells among total Foxp3+ cells (Supplementary Fig. 2b, c). Together, the data suggest that Setd2 supports Treg homeostasis in a cell-intrinsic manner.

Next, we purified splenic Treg cells from *Foxp3^Cre-YFP^Setd2^f/f^* and control mice and transferred them to *Rag2^−/−^* hosts (Fig. 2d). We confirmed that the purity of Treg cells from both genotypes were comparable and higher than 99% (Supplementary Fig. 2d). Interestingly, Setd2-deficient Treg cells showed decreased reconstitution in the

**Fig. 1 | Setd2 is essential for Treg cells to repress T cell activation. a, b** Expression of Setd2 gated on Treg cells (CD3$^+$CD4$^+$Foxp3$^+$) or CD4$^+$ Tconv cells (CD3$^+$CD4$^+$Foxp3$^-$) from the spleen of mice with indicated genotypes was analyzed by flow cytometry. The gating strategy was shown in Supplementary Fig. 1a. **b** MFI of Setd2 normalized to the average of the Ctrl group from each batch of the experiment is shown (Ctrl $n = 3$, KO $n = 4$). **b, c** Ctrl, *Foxp3*$^{Cre}$*Setd2*$^{f/+}$; KO, *Foxp3*$^{Cre}$-*Setd2*$^{f/f}$. **c** Level of H3K36me3 and H3 in purified splenic (SP) or large intestinal (LI) Treg cells (CD3$^+$CD4$^+$Foxp3-YFP$^+$) was analyzed by western blot. **d–l** Percentages of effector/effector memory cells (CD44$^+$CD62L$^{low}$) (**d, e, j**), naïve cells (CD44$^-$CD62L$^{high}$) (**f, g, k**), or central memory cells (CD44$^+$CD62L$^{high}$) (**h, i, l**) in CD4$^+$ Tconv cells (CD3$^+$CD4$^+$Foxp3-YFP$^-$) (**d, f, h**) or non-CD4 T cells (CD3$^+$CD4$^-$) (**e, g, i**), or Treg cells (**j, k, l**) from different organs were analyzed by flow cytometry. (SP: Ctrl $n = 13$, KO $n = 17$; ILN: Ctrl $n = 5$, KO $n = 8$; MLN: Ctrl $n = 9$, KO $n = 11$; LI: Ctrl $n = 12$, KO $n = 15$). **m** Percentages of Treg cells (Foxp3-YFP$^+$CD25$^+$), Foxp3-YFP$^-$CD25$^+$, and Foxp3-YFP$^{low}$CD25$^-$ Treg progenitors in CD4$^+$CD8$^-$ (CD4 single positive, SP) thymocytes were analyzed by flow cytometry. ($n = 4$ per group). **n** Percentages of Treg cells (Foxp3-YFP$^+$) in CD3$^+$CD4$^+$ cells from different organs were analyzed by flow cytometry. (SP: Ctrl $n = 19$, KO $n = 23$; ILN: Ctrl $n = 9$, KO $n = 10$; MLN: Ctrl $n = 12$, KO $n = 14$; LI: Ctrl $n = 17$, KO $n = 20$). **o** Percentages of Treg cells (Foxp3$^+$) in CD3$^+$CD4$^+$ cells from different organs were analyzed by flow cytometry. Fat was from epididymal adipose tissue. ($n = 4$ per group). **c–l, n** SP spleen, ILN inguinal lymph nodes, MLN mesenteric lymph nodes, LI large intestine. **a–o** Representative of 2–8 independent experiments. Data were means ± SEM. Source data are provided as a Source Data file.

intestine but not in the spleen (Fig. 2e). Some Treg cells lost Foxp3 expression in immunodeficient hosts and the percentage of Treg cells converted to Foxp3$^-$ cells (exTreg) was higher in the absence of Setd2 (Fig. 2f). The data suggest that Setd2 prevents fate conversion of Treg cells during homeostatic expansion. Nevertheless, the numbers of T cells with Treg and exTreg cells added up together were still fewer in the intestine of *Rag2*$^{-/-}$ hosts transferred with Setd2-deficient Treg cells (Fig. 2g). This indicates that Setd2-deficient donor cells, with Treg and exTreg cells, added up, have defective maintenance in *Rag2*$^{-/-}$ mice. This led us to think that there might be other mechanisms in addition to preventing fate conversion that mediates Setd2-supported Treg maintenance. T cells were expected to undergo homeostatic expansion in immunodeficient mice. Thus, defective Treg maintenance in the absence of Setd2 might occur during proliferation and activation. To test this idea, we treated *Foxp3*$^{Cre-YFP}$*Setd2*$^{f/f}$ mice with IL-2-α-IL-2 complex (IL-2c), which favors the expansion of Treg over Tconv cells[38,39]. Strikingly, we found that both splenic and intestinal Setd2-deficient Treg cells expanded less in response to IL-2 than control Treg cells, as demonstrated by the percentages and absolute numbers of Treg cells (Fig. 2h–k). This result was unlikely due to reduced proliferation since the percentage of Ki67$^+$ cells was not decreased in Treg cells from *Foxp3*$^{Cre-YFP}$*Setd2*$^{f/f}$ mice (Supplementary Fig. 2e, f). Rather, the proliferative rate of Treg cells was higher in splenic and intestinal Treg cells of PBS treated *Foxp3*$^{Cre-YFP}$*Setd2*$^{f/f}$ mice, possibly as a consequence of the proinflammatory environment (Supplementary Fig. 2e, f). In addition, cleaved Caspase 3$^+$ cells (apoptotic cells) among Treg cells were increased in both the spleen and intestine of *Foxp3*$^{Cre-YFP}$*Setd2*$^{f/f}$ mice with or without IL-2c treatment (Fig. 2l, m). Furthermore, blockade of cell apoptosis and necrosis by treatment with z-VAD-FMK and necrostatin-1 partially rescued the curtailed expansion of Setd2-deficient Treg cells in response to IL-2c treatment (Fig. 2n, o)[40]. The data collectively suggest that Setd2 is essential for Treg homeostatic expansion by preventing cell death.

## Setd2 supports Treg cells to inhibit CD45RB$^{high}$ T cell-induced colitis

In vitro, Setd2-deficient splenic Treg cells had a similar ability, whereas Setd2-deficient intestinal Treg cells had a reduced competency, to suppress conventional T-cell proliferation compared with control Treg cells (Supplementary Fig. 3a). Nevertheless, when transferred together with CD45RB$^{high}$ pathogenic T cells into *Rag2*$^{-/-}$ mice, splenic Setd2-deficient Treg cells had a reduced capacity to suppress colitis compared to control Treg cells in vivo (Fig. 3a, b), as indicated by the greater body weight loss and shortened colons of recipient mice (Fig. 3b–d). Histological analysis showed that mice transferred with Setd2-deficient Treg cells (T + KOTreg) had worsened colitis, as demonstrated by increased inflammatory cell infiltration, loss of goblet cells, damage and dysplasia of intestinal epithelial cells compared with mice transferred with control Treg cells (T + CtrlTreg) (Fig. 3e, f). Flow cytometry analysis confirmed that there was increased infiltration of neutrophils but not eosinophils in the lamina propria of the "T + KOTreg" group compared with

the "T + CtrlTreg" group (Fig. 3g, h). Previous studies have indicated that both Th1 and Th17 cells have pathogenic functions in colitis[41]. Consistently, control Treg cells but not Setd2-deficient Treg cells efficiently suppressed the accumulation of IFN-γ-producing T cells (Fig. 3i). The number of IL-17-producing CD4$^+$ T cells was also higher in "T + KOTreg" group than in the "T + CtrlTreg" group (Fig. 3j). In addition, the percentage of Treg cells was lower in the "T + KOTreg" group (Fig. 3k). Together, the data indicate that loss of Setd2 in Treg cells prevents them from inhibiting pathogenic T cell expansion during colitis in vivo. To find out if the reduced Treg cells in the "T + KOTreg" group was due to fate conversion, we transferred Thy1.1$^+$CD45RB$^{high}$ pathogenic T cells and Thy1.2$^+$Ctrl Treg or Thy1.2$^+$KO Treg cells to *Rag2*$^{-/-}$ mice, which were sacrificed for analysis when the mice developed colitis (Supplementary Fig. 3b). Significantly fewer percentages of Treg cells from Thy1.2$^+$Setd2-deficient donors maintained their Foxp3 expression in *Rag2*$^{-/-}$ hosts (Supplementary Fig. 3c). This data indicates that Setd2 is important for preventing fate conversion of Treg cells during CD45RB$^{high}$ T cell-induced colitis. However, the fate-converted Treg cells (exTreg) contributed marginally (less than 3%) to the pathogenic T cell pool and this proportion was much lower in the "T + KOTreg" group (Supplementary Fig. 3d). Furthermore, we observed a trend towards a reduction in absolute numbers of Thy1.2$^+$ T cells including Treg and exTreg cells from the large intestine of *Rag2*$^{-/-}$ mice receiving KO Treg cells (Supplementary Fig. 3e). The data raise a possibility that other mechanisms in addition to preventing fate conversion of Treg cells could contribute to Setd2-supported Treg sustentation in CD45RB$^{high}$ T cell-induced colitis.

## Setd2 facilitates GATA3 and ST2 expression in intestinal tTreg cells

To understand the mechanism by which Setd2 regulates Treg cells, we first assessed the Setd2-mediated H3K36me3 profile in intestinal Treg cells on a genome-wide scale. As intestinal *Foxp3*$^{Cre-YFP}$*Setd2*$^{f/f}$ Treg cells had barely detectable levels of H3K36me3 (Fig. 1c), we performed chromatin immunoprecipitation sequencing (ChIP-seq) for H3K36me3 in splenic and intestinal Treg cells from *Foxp3*$^{Cre-YFP}$ mice. Consistent with previous reports, H3K36me3 was enriched in gene bodies, with levels increasing towards the 3'UTR and decreasing at downstream of genes (Fig. 4a and Supplementary Fig. 4a)[34]. Intestinal Treg cells showed a slightly higher cumulative H3K36me3 signal than splenic Treg cells (Fig. 4a). The cumulative H3K36me3 signal of splenic Treg signature genes was higher in splenic Treg cells than in intestinal Treg cells (Fig. 4a and Supplementary Data 1). In parallel, the cumulative H3K36me3 signal of colonic Treg signature genes was higher in intestinal Treg cells than in splenic Treg cells (Fig. 4a and Supplementary Data 1). The mRNA expression of genes with a significantly different expression between intestinal Treg cells and splenic Treg cells was positively correlated with the H3K36me3 level in the gene locus (Fig. 4b and Supplementary Data 2). The data are consistent with previous findings indicating that H3K36me3 marks actively transcribed genes[26].

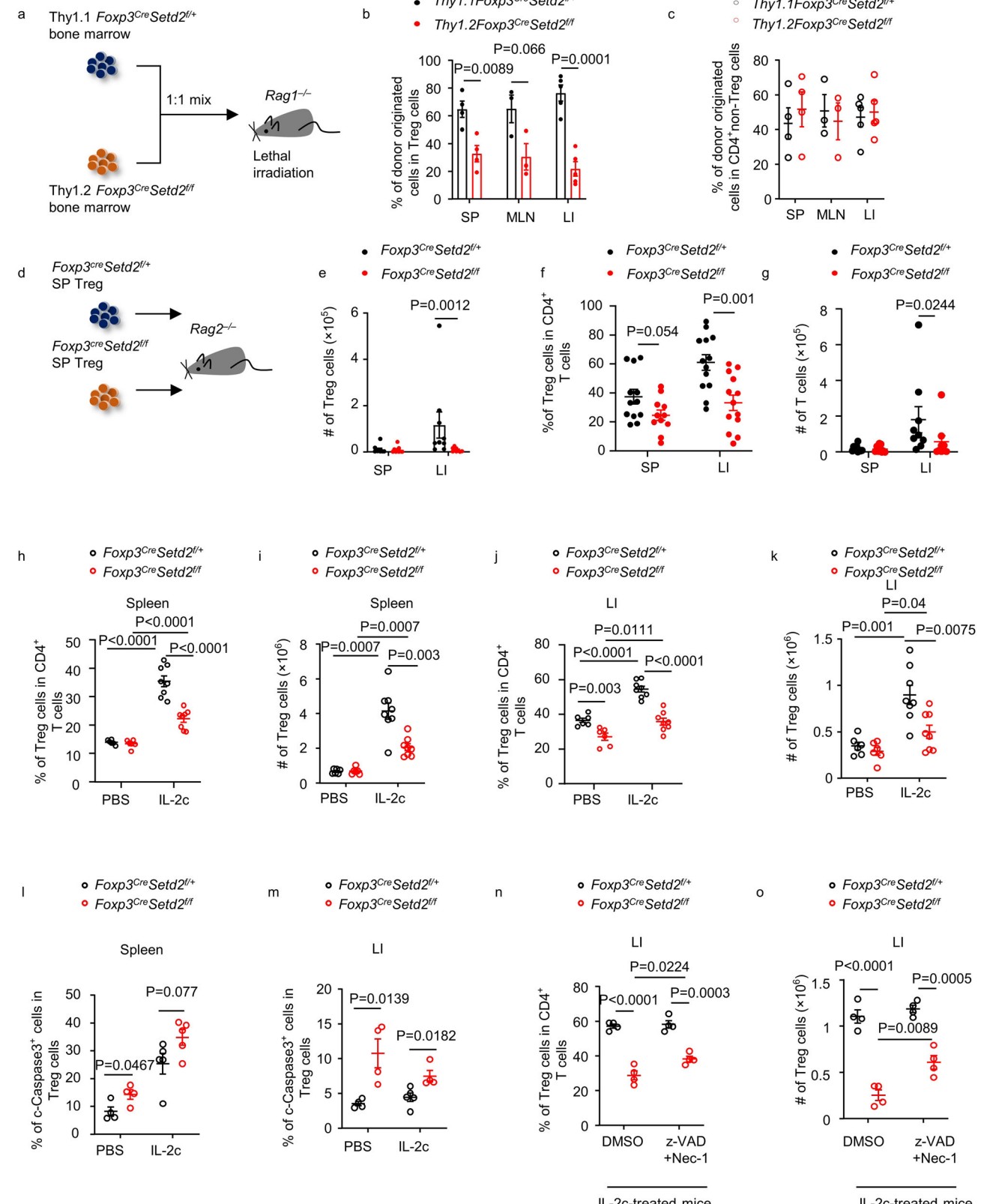

Next, we performed transcriptome sequencing (RNA-seq) of splenic and intestinal Treg cells sorted from *Foxp3^Cre-YFP^Setd2^f/f^* mice (KO) and *Foxp3^Cre-YFP^Setd2^f/+^* mice (control). Principal component analysis (PCA) indicated that PC1 distinguished splenic Treg cells from intestinal Treg cells, and PC2 separated Setd2-deficient from Setd2-sufficient Treg cells (Supplementary Fig. 4b). Splenic and intestinal Treg cells shared

249 upregulated and 172 downregulated genes upon Setd2 ablation (Supplementary Fig. 4c and Supplementary Data 3). These include decreased expression of *Il1rl1*, *Myc* and increased expression of *Cxcr3*, *Icos*, and *Gpr15*, which have been reported to mediate/regulate the migration or function of Treg cells (Supplementary Fig. 4d). We didn't observe altered GITR or reduced CTLA-4 expression, which could be

**Fig. 2 | Setd2 maintains Treg homeostasis by sustaining Treg survival.**
**a–c** Mixed bone marrow chimeric mouse was generated with bone marrows from donors of indicated genotypes as shown in (**a**). **a** Schematic strategy of generating bone marrow chimeric mouse. **b, c** Cells from the spleen (SP), mesenteric lymph nodes (MLN), and large intestinal lamina propria (LI) of recipient mice were isolated 6–8 weeks after bone marrow transfer. Percentages of Treg cells (CD3+CD4+Foxp3+) (**b**) and CD4+ non-Treg cells (CD3+CD4+Foxp3−) (**c**) from different donor origins were analyzed by flow cytometry. (SP: $n = 4$; MLN: $n = 3$; LI: $n = 5$). **d–g** Splenic Treg cells (CD3+CD4+Foxp3-YFP+) of mice with indicated genotypes were transferred to *Rag2−/−* mice as shown in (**d**) (LI: $n = 13$ per group; SP: $n = 12$ for Ctrl and $n = 11$ for KO). Mice were sacrificed for analysis 4 weeks after transfer. **e–g** Total cell number of Treg cells (CD3+CD4+Foxp3+) (**e**), percentages of Treg cells in CD4+ T cells

(CD3+CD4+ cells) (**f**), and the total number of CD4+ T cells (**g**) from SP or LI from recipient mice were analyzed by flow cytometry. **h–m** Mice with indicated genotypes were treated with PBS or IL-2-α-IL-2 complex (IL-2c), (PBS $n = 6$ per group, IL-2c $n = 8$ per group; cleaved caspase3 staining: $n = 4$ for PBS group and $n = 5$ for IL-2c group). **n, o** IL-2c-treated mice of indicated genotypes were injected with DMSO or z-VAD-FMK plus Necrostatin-1, ($n = 4$ per group). **h, j, n** Percentage of Treg cells (CD3+CD4+Foxp3+) in CD4+ T (CD3+CD4+) cells of spleen or LI was analyzed by flow cytometry. **i, k, o** Absolute numbers of Treg cells were shown. **l, m** Percentage of cleaved caspase3+ (c-caspase3) cells in splenic or LI Treg cells was analyzed by flow cytometry. **b, c, e–o** Representative of 2–5 independent experiments. Data were means ± SEM. Source data are provided as a Source Data file.

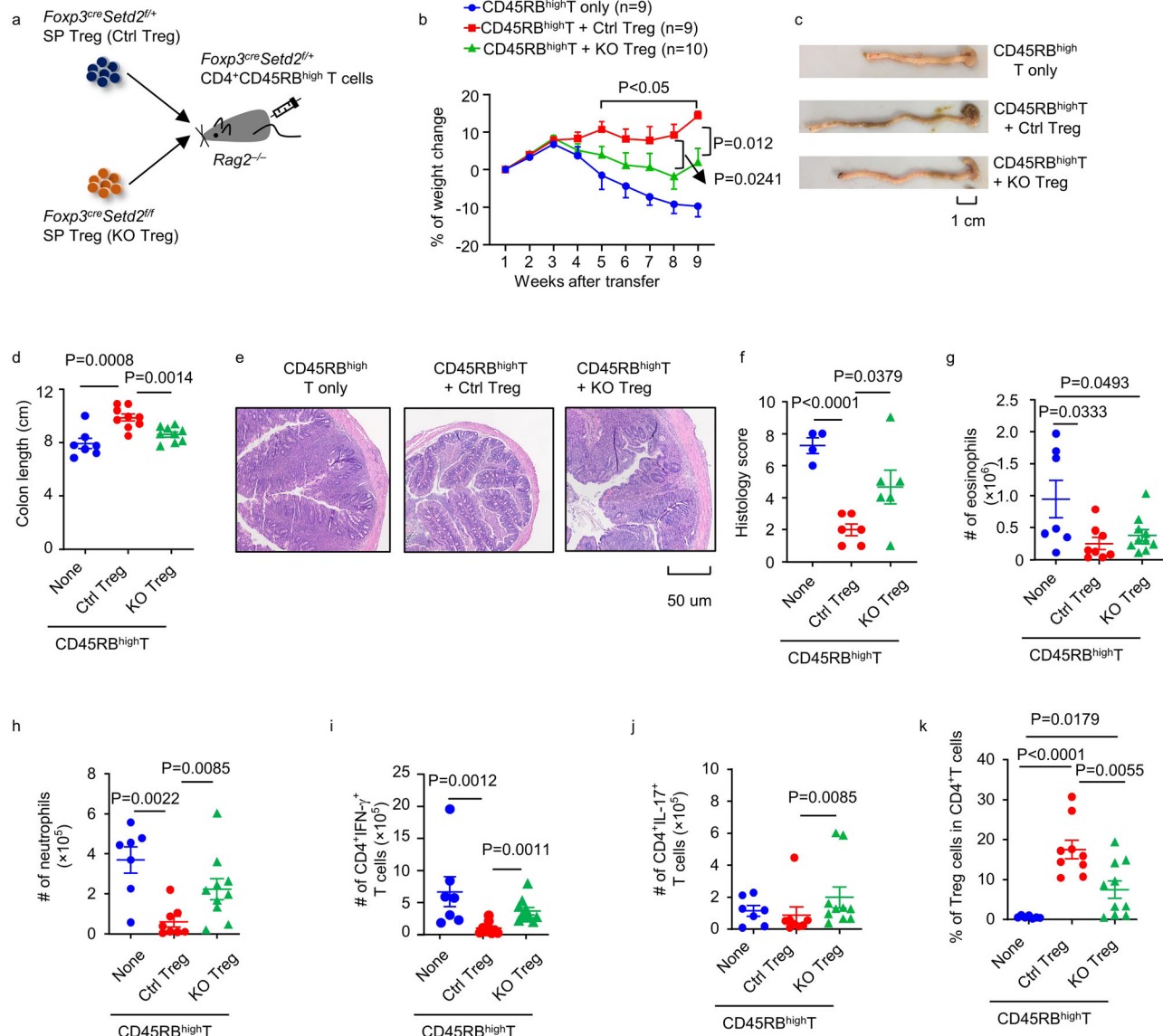

**Fig. 3 | Setd2 is required for Treg cells to suppress CD45RBhighT cell-induced colitis.** CD45RBhighT cells (CD3+CD4+Foxp3-YFP−CD45RBhigh) with or without splenic Treg cells (CD3+CD4+Foxp3-YFP+) purified from mice of indicated genotypes were transferred to *Rag2−/−* mice. Recipient mice were sacrificed for analysis 9 weeks after transfer. **a** Protocol of T cell transfer colitis. **b** Percentages of weight change (CD45RBhighT cells only group: $n = 9$; Ctrl group: $n = 9$; KO group: $n = 10$). **c** Images of large intestine. **d** Lengths of colons (CD45RBhighT cells only group: $n = 7$; Ctrl group: $n = 9$; KO group: $n = 9$). **e** Paraffin-embedded colon sections were stained with hematoxylin and eosin (H&E). Scale bar is 50 um. **f** Histological scores for H&E staining (CD45RBhighT cells only group: $n = 4$; Ctrl group: $n = 6$; KO group: $n = 6$).

**g–k** Large intestinal lamina propria lymphocytes were isolated. Absolute numbers of eosinophils (CD11b+Siglec-F+ cells) (**g**), neutrophils (CD11b+Ly6G+ cells) (**h**), CD4+IFN-γ+ T cells (CD3+CD4+IFN-γ+ cells) (**i**), and CD4+IL-17+ T cells (CD3+CD4+IL-17+ cells) (**j**) were analyzed by flow cytometry. **k** Percentage of Treg cells (CD3+CD4+Foxp3+) in CD4+ T (CD3+CD4+) cells was analyzed by flow cytometry. (CD45RBhighT cells only group: $n = 7$, Ctrl group: $n = 8$, KO group: $n = 10$ for **g, h, j**; CD45RBhighT cells only group: $n = 7$, Ctrl group: $n = 9$, KO group: $n = 10$ for **i, k**). **b, d, f–k** Representative of three independent experiments. Data were means ± SEM. Source data are provided as a Source Data file.

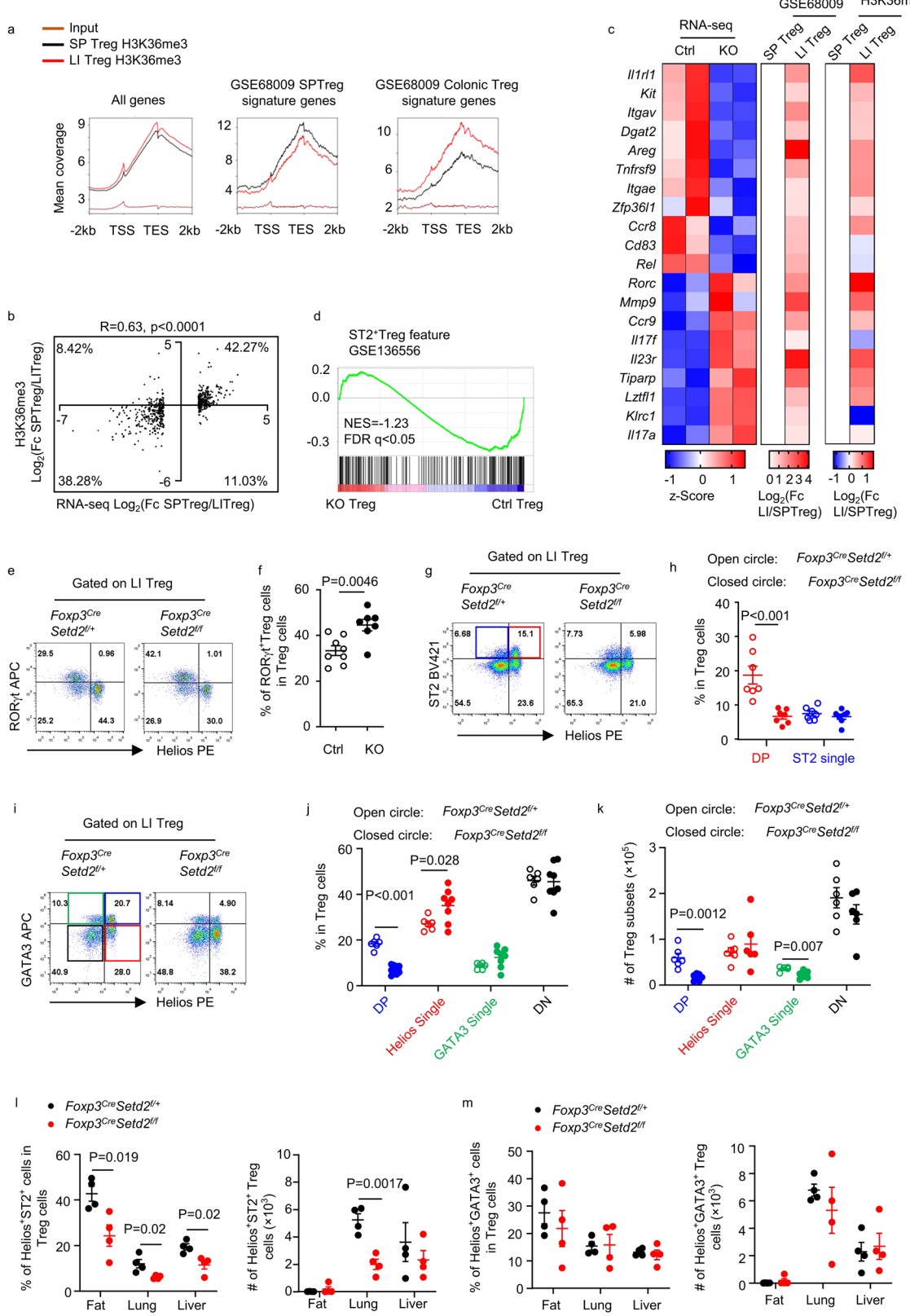

correlated with the defective function of Treg cells[42,43], at the mRNA or protein level in Setd2-deficient Treg cells (Supplementary Data 3 and Supplementary Fig. 4e, f). Among the differentially expressed genes in KO intestinal Treg cells compared with control intestinal Treg cells, 57 increased and 53 decreased genes were colonic Treg signature genes (Supplementary Fig. 4g, h and Supplementary Data 3), the majority of

which had higher H3K36me3 signal in their gene locus in colonic Treg cells than that in splenic Treg cells (Fig. 4a, c). These include *Il1rl1* and *Rorc*, likely implying differential branching of GATA3 and RORγt Treg subsets (Fig. 4c and Supplementary Fig. 4h). Indeed, gene set enrichment analysis (GSEA) showed that KO Treg cells had less enrichment of ST2+ Treg features[44], manifested by decreased *Il1rl1*, *Areg* and *Ccr8*

**Fig. 4 | Setd2 is important for supporting GATA3 and ST2 expression in intestinal tTreg cells. a, b** H3K36me3 ChIP-seq analysis on Treg cells (CD3⁺CD4⁺Foxp3-YFP⁺) sorted from spleen (SP) or large intestine (LI) of *Foxp3^(Cre-YFP)* mice. **a** Cumulative curve of average H3K36me3 signals distributed scaled regions of all genes, SPTreg signature genes and Colonic Treg signature genes. TSS transcription start site, TES transcription end site. **b** Correlation analysis on all SPTreg and Colonic Treg signature genes (GSE68009) using Log₂ (fold change) (Fc) of gene mRNA expression and Log₂ (Fc) of summed H3K36me3 signals. Each dot represents one gene. Percentages indicate the proportions of genes distributed in each quadrant. R is Pearson correlation coefficient. **c, d** LI Treg cells sorted from *Foxp3^(Cre-YFP)Setd2^(f/+)* (Ctrl) or *Foxp3^(Cre-YFP)Setd2^(f/f)* (KO) mice were subjected to RNA-seq analysis. **c** Heatmap showing differentially expressed genes based on *Z*-score of normalized FPKM value, and corresponding normalized Log₂Fc of LITreg over SPTreg in microarray data of GSE68009 and Log₂Fc of summed H3K36me3 signals in LITreg

over SPTreg. **d** GSEA analysis based on ST2⁺ Treg cells signature. **e–k** Cells were isolated from the LI. **e, f** Flow cytometry analysis of expression of RORγt and Helios gated on Treg cells (CD3⁺CD4⁺Foxp3⁺). **f** Percentage of Helios⁻RORγt⁺ cells. (Ctrl *n* = 8; KO *n* = 7). **g, h** Flow cytometry analysis of ST2 and Helios expression gated on Treg cells. (*n* = 7 per group). **h** Percentages of DP Treg cells (double positive, Helios⁺ST2⁺ cells) and ST2 Single Treg cells (ST2 single positive, Helios⁻ST2⁺ cells). **i, j, k** Flow cytometry analysis of GATA3 and Helios expression gated on Treg cells. (Ctrl *n* = 6; KO *n* = 8. For analyzing total cell number, *n* = 6 per group). Percentages (**j**) and absolute numbers (**k**) of DP Treg cells (Helios⁺GATA3⁺), Helios Single Treg cells (Helios⁺GATA3⁻), GATA3 Single Treg cells (Helios⁻GATA3⁺), and DN Treg cells (double negative, Helios⁻GATA3⁻). **l, m** Percentages and numbers of Helios⁺ST2⁺ Treg cells (**l**) and Helios⁺GATA3⁺ Treg cells (**m**) in indicated organs analyzed by flow cytometry. (*n* = 4 per group). **f, h, j–m** Representative of 2–4 independent experiments. Data were means ± SEM. Source data are provided as a Source Data file.

expression (Fig. 4c, d and Supplementary Data 1). In contrast, the RORγt⁺ Treg signature was enriched in KO Treg cells, represented by increased expression of *Rorc*, *Il23r*, *Il17a*, and *Il17f* (Fig. 4c, Supplementary Fig. 4i, and Supplementary Data 1). Consistently, we found that the percentage of RORγt⁺ Treg cells among total Treg cells was increased in *Foxp3^(Cre-YFP)Setd2^(f/f)* mice (Fig. 4e, f). However, the number of RORγt⁺ Treg cells was unchanged (Supplementary Fig. 4j).

Flow cytometry analysis confirmed the increased expression of IFN-γ and IL-17 in intestinal Setd2-deficient Treg cells identified by RNA-seq, likely indicating a state of "fragility" and decreased inhibitory function of these Treg cells[6,7,9,45,46] (Supplementary Fig. 4k, l and Supplementary Data 3). Consistently, *Foxp3^(Cre-YFP)Setd2^(f/f)* mice had more weight loss and shortened colons in TNBS-induced colitis, accompanied by enhanced IFN-γ production by CD4⁺ T cells in the colon (Supplementary Fig. 4m–o). However, no difference in IL-17 production by CD4⁺ T cells was found (Supplementary Fig. 4p). The data suggest that in TNBS colitis, Setd2-deficient Treg cells are defective in suppressing Th1 but not Th17 responses, which may be controlled by the retained inhibitory function of RORγt⁺ Treg cells[47].

GATA3 and ST2 can be expressed by both tTreg and pTreg cells[19,48]. However, Helios⁺ST2⁺ Treg and Helios⁺GATA3⁺ Treg cells of the tTreg cell subset, rather than Helios⁻GATA3⁺ Treg or Helios⁻ST2⁺ Treg cells, showed the most defects due to Setd2 ablation, as indicated by the extent of reduction in both percentages and absolute numbers (Figs. 4g–k and Supplementary Fig. 4q). Therefore, GATA3 and ST2 expressing tTreg cells are the hallmark intestinal Treg subsets supported by Setd2. The percentages of Helios⁺ST2⁺ Treg cells were also reduced in the liver, lung, and visceral adipose tissue of *Foxp3^(Cre-YFP)Setd2^(f/f)* mice (Fig. 4l). Numbers of Helios⁺ST2⁺ Treg cells were decreased in the lung of *Foxp3^(Cre-YFP)Setd2^(f/f)* mice (Fig. 4l). No difference in percentages or absolute numbers of Helios⁺GATA3⁺ Treg cells between *Foxp3^(Cre-YFP)Setd2^(f/f)* mice and control mice was found in the liver, lung or visceral adipose tissue (Fig. 4m). These data suggest that Setd2 broadly supports the maintenance of tissue ST2⁺ tTreg cells.

## Setd2 sustains the reciprocal relationship of GATA3 and ST2 in tTreg cells

GATA3 and ST2 have been shown to mutually promote the expression of each other in Th2 cells and Treg cells[19,20,49]. To investigate if the decreased expression of one causes a decrease in the expression of the other, we analyzed GATA3 and ST2 expression in splenic and intestinal Treg cells during ontogeny. GATA3 and ST2 expression among splenic and intestinal tTreg cells was analyzed (Fig. 5a–h). Strikingly, reduced proportions of GATA3⁺ST2⁺ cells among tTreg cells in both the spleen (Fig. 5a, c) and intestine (Fig. 5e, g) of *Foxp3^(Cre-YFP)Setd2^(f/f)* mice were found as early as 2 weeks of age, which preceded the increase of RORγt⁺ Treg cells occurring at 3 weeks of age (Supplementary Fig. 5a). In 2-week-old mice, there was also a decrease in GATA3⁻ST2⁺ tTreg cells in the spleen but not in the intestine of *Foxp3^(Cre-YFP)Setd2^(f/f)* mice (Fig. 5b, f). At all observed time points before weaning, no dramatic difference in the

percentage of total Treg or Helios⁺ tTreg cells was found in either the spleen or intestine of *Foxp3^(Cre-YFP)Setd2^(f/f)* mice (Supplementary Fig. 5b–e). Together, our data support that there is defective induction of GATA3 and ST2 expression in Helios⁺ tTreg cells in the absence of Setd2.

ST2 and GATA3 expression in Treg cells has been shown to be sustained or induced by a series of stimuli, including IL-2, IL-33, and TCR signaling activators[6,19], which may be more abundant in the intestine than in the SLOs[17,19,50]. Consistently, GATA3 expression in Treg cells was significantly increased by IL-2 and IL-33 in the presence of TCR signaling, which was defective in the absence of Setd2 (Fig. 5i, j). And ST2 expression, although not further increased by IL-2 or IL-33 in the presence of TCR signaling within 24 h of stimulation, was consistently lower in Setd2-deficient Treg cells (Fig. 5i, k). Importantly, there was a reduction in the percentage of GATA3⁺ST2⁺ Treg cells (DP Treg cells) among total ST2⁺ Treg and GATA3⁺ Treg cells in vitro (Fig. 5i, l, m), as well as in the intestine in vivo (Supplementary Fig. 5f–h). The data suggest an impaired reciprocal relationship of ST2 and GATA3 in Setd2-deficient Treg cells. Notably, upon overexpression of ST2 or GATA3 in Treg cells to a copious extent (Supplementary Fig. 5i, j), the expression of ST2 or GATA3 in Setd2-deficient Treg cells was partially but not completely restored (Fig. 5n, o). Strikingly, knocking out IL-33, the cytokine responsible for the reciprocal relationship between ST2 and GATA3 expression, caused a reduction in the percentage of GATA3⁺ST2⁺ tTreg cells in *Foxp3^(Cre-YFP)Setd2^(f/+)* mice but not in *Foxp3^(Cre-YFP)Setd2^(f/f)* mice (Fig. 5p). This also mitigated the discrepancy in the percentage of GATA3⁺ST2⁺ tTreg cells between *Foxp3^(Cre-YFP)Setd2^(f/f)* mice and controls (Fig. 5p). Together, the data strongly suggest that Setd2 sustains the reciprocal relationship between GATA3 and ST2 expression in Treg cells in vitro and in vivo, and Setd2 is required for the supportive effect of IL-33 on GATA3⁺ST2⁺ tTreg cells. The previous study has shown that tissue Treg cells are originated from splenic PD-1⁺KLRG1⁻ and PD-1⁺KLRG1⁺ Treg precursors[51]. We found no difference in percentages of PD-1⁻KLRG1⁻, PD-1⁺KLRG1⁻, or PD-1⁺KLRG1⁺ cells in splenic Treg cells between *Foxp3^(Cre-YFP)Setd2^(f/f)* mice and controls at the adult stage (Supplementary Fig. 5k, l). This suggests that Setd2 is unlikely to regulate the development or maintenance of tissue Treg progenitors. Interestingly, expression of ST2 was reduced in PD-1⁺KLRG1⁺ splenic Treg cells from *Foxp3^(Cre-YFP)Setd2^(f/f)* mice compared with controls. The data indicate that Setd2 supports ST2 expression in splenic tissue Treg progenitors (Supplementary Fig. 5m). STAT5, phosphorylation of which could be driven by IL-2, has been shown to be important for the mutually supportive network of GATA3 and ST2 expression in Th2 cells[20]. We observed slightly but significantly reduced CD25 (IL-2Rα) expression in splenic and intestinal Treg cells of *Foxp3^(Cre-YFP)Setd2^(f/f)* mice (Supplementary Fig. 5n, o), whereas the expression of CD122 (IL-2Rβ) in splenic Treg cells was increased in *Foxp3^(Cre-YFP)Setd2^(f/f)* mice (Supplementary Fig. 5p, q). Nevertheless, no difference in the level of phosphorylated STAT5 in tTreg cells from the spleen or the intestine of *Foxp3^(Cre-YFP)Setd2^(f/f)* mice was observed (Supplementary Fig. 5r, s), suggesting that signals downstream of IL-2 receptors are less likely to be breached by deletion of Setd2.

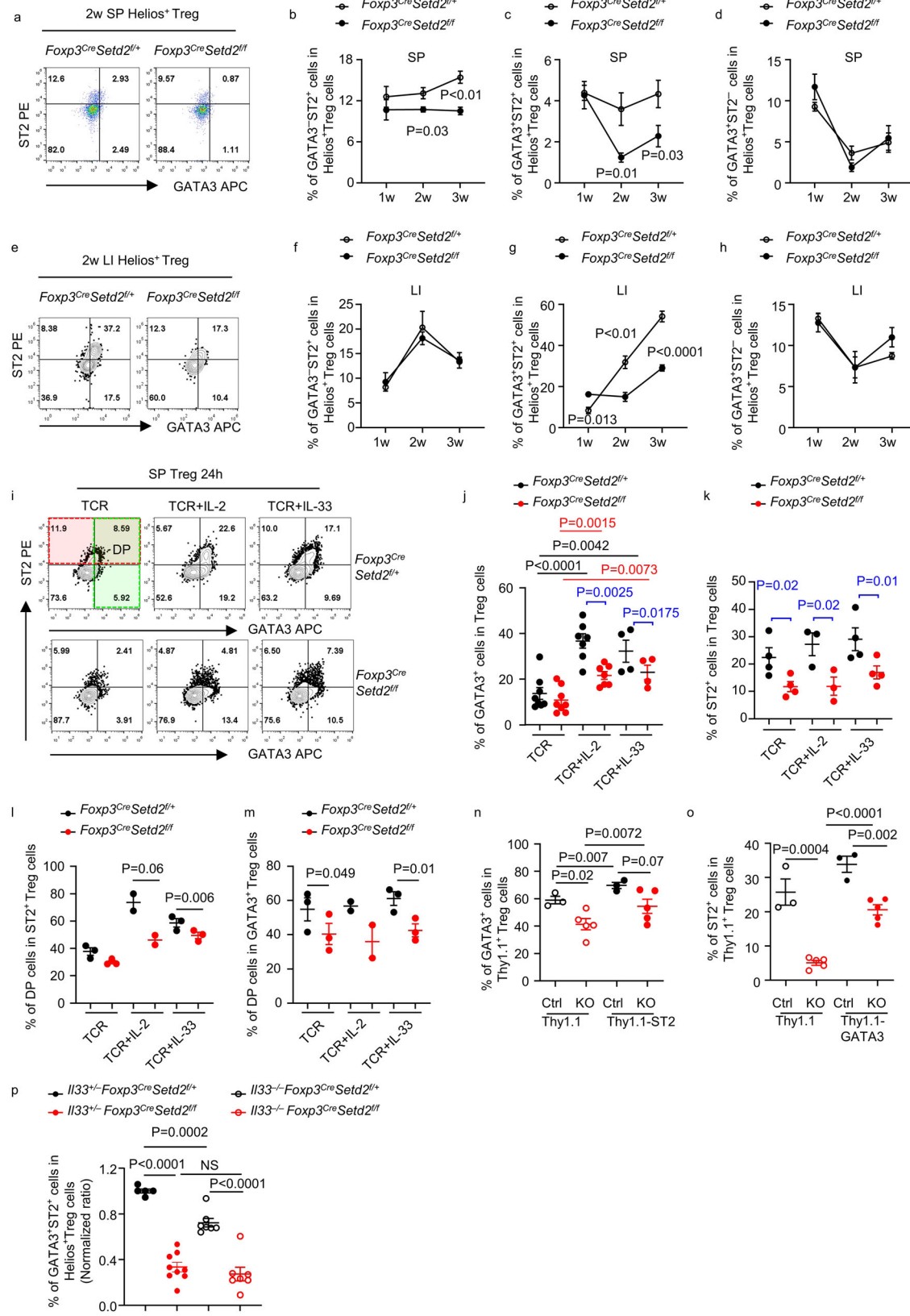

## Loss of Setd2 in Treg cells causes enhanced intestinal Th2 responses

Consistent with the function of GATA3-expressing Treg cells to suppress Th2 cells[8,10], we observed increased proportions of GATA3+ST2+ and GATA3+ST2− Th2 cells among CD4+ Tconv cells in the intestine of *Foxp3Cre-YFPSetd2f/f* mice (Fig. 6a, b). Moreover, the absolute number of

Th2 cells was significantly increased, suggesting a defective suppression of Th2 cells by Setd2-deficient Treg cells (Fig. 6c). With ST2 expressed on the cell surface, both Treg cells and Th2 cells might respond to IL-33 stimulation. Nevertheless, while GATA3+ Treg cells were more obviously promoted than Th2 cells by IL-33 in the control group, Th2 cells rather than GATA3+ Treg cells were preferentially

**Fig. 5 | Setd2 supports the reciprocal relationship of GATA3 and ST2 expression in tTreg cells. a–h** Splenocytes (SP) (**a–d**) and large intestinal (LI) LPLs (**e–h**) were isolated from 1-week (1w), 2-week (2w), or 3-week (3w)-old *Foxp3*$^{Cre-YFP}$*Setd2*$^{f/+}$ or *Foxp3*$^{Cre-YFP}$*Setd2*$^{f/f}$ mice. (1w: *n* = 3 per group; 2w: *n* = 6 per group; 3w: Ctrl *n* = 7; KO *n* = 6). **a, e** Representative FACS plot of ST2 and GATA3 expression gated on Helios$^+$ Treg cells. Percentages of GATA3$^-$ST2$^+$ cells (**b, f**), GATA3$^+$ST2$^+$ cells (**c, g**), and GATA3$^+$ST2$^-$ cells (**d, h**) in Helios$^+$ Treg cells (CD3$^+$CD4$^+$Foxp3$^+$Helios$^+$) were analyzed by flow cytometry and shown. **i–m** Splenic (SP) Treg cells (CD3$^+$CD4$^+$YFP$^+$) were sorted from mice of indicated genotypes and cultured with α–CD3/α–CD28 (TCR) with or without supplement of IL-2 (TCR + IL-2) or IL-33 (TCR + IL-33) for 24 h. **i** Expression of ST2 and GATA3 gated on Treg cells (CD3$^+$CD4$^+$Foxp3$^+$) was analyzed by flow cytometry. **j, k** Percentage of GATA3$^+$ cells (**j**) (TCR: *n* = 8 per group; TCR + IL-2: *n* = 7 per group; TCR + IL-33: *n* = 4 per group) or ST2$^+$ cells (**k**) (TCR and TCR + IL-33: *n* = 4 per group; TCR + IL-2: *n* = 3 per group) gated on Treg cells was shown. **l, m** Percentage of GATA3$^+$ST2$^+$ (double positive, DP) cells among total ST2$^+$ Treg cells (**l**) or GATA3$^+$ Treg cells (**m**) was shown. (TCR and TCR + IL-33: *n* = 3 per group; TCR + IL-2: *n* = 2 per group). **n, o** Splenic Treg cells sorted from Ctrl (*Foxp3*$^{Cre-YFP}$*Setd2*$^{f/+}$) or KO (*Foxp3*$^{Cre-YFP}$*Setd2*$^{f/f}$) mice were cultured with α-CD3, α-CD28, IL-2 and IL-33, and were infected with retrovirus expressing Thy1.1, Thy1.1-ST2, or Thy1.1-GATA3. The percentage of GATA3$^+$ cells (**n**) or ST2$^+$ cells (**o**) gated on Thy1.1$^+$ Treg cells (Thy1.1$^+$Foxp3$^+$) was analyzed by flow cytometry. (Ctrl *n* = 3; KO *n* = 5). **p** LI LPLs were isolated from mice of indicated genotypes (*n* = 5, 9, 7, 7, respectively). The percentage of GATA3$^+$ST2$^+$ cells gated on Helios$^+$ Treg cells (CD3$^+$CD4$^+$Foxp3$^+$Helios$^+$) was analyzed by flow cytometry and normalized to *Il33*$^{+/-}$ *Foxp3*$^{Cre}$*Setd2*$^{f/+}$ group as 1 based on littermates. **b–d, f–h, j–p** Data were means ± SEM. **a–h** Data were pooled from 3–7 mice per time point for each genotype. **i–p** Representative of 2–6 independent experiments. Source data are provided as a Source Data file.

---

boosted in *Foxp3*$^{Cre-YFP}$*Setd2*$^{f/f}$ mice (Fig. 6d–f). Under steady state, IL-4 expression in intestinal CD4$^+$ T cells and serum IgE levels were upregulated in *Foxp3*$^{Cre-YFP}$*Setd2*$^{f/f}$ mice, whereas IL-5 and IL-13 expression levels were comparable (Fig. 6g–j). Consistent with the promotive effect of IL-33 on Th2 cells, IL-33 significantly increased intestinal type 2 cytokine production and serum IgE concentration, and the difference in these parameters between *Foxp3*$^{Cre-YFP}$*Setd2*$^{f/f}$ and control mice was massively amplified by IL-33 (Fig. 6g–j). The data indicate that Setd2 expression by Treg cells is fundamental for them to suppress intestinal Th2 responses, especially upon IL-33 treatment.

GATA3 and ST2 expression in Treg cells has been implicated in the pathogenesis of intestinal tumors[21–24]. Interestingly, we observed that increased SETD2 expression was accompanied by enhanced GATA3 expression in Treg cells in cancerous tissues from colorectal cancer (CRC) patients compared with noncancerous colorectal tissues (Fig. 6k–n). Meanwhile, the percentage of Th2 cells, as well as the percentage of IL-13$^+$ cells in Tconv cells, was significantly lower in cancerous tissues (Fig. 6o, p). When human peripheral blood Treg cells were infected with retrovirus expressing short hairpin RNA (shRNA) targeting SETD2, a proportion of the infected Treg cells lost SETD2 expression (SETD2-null Treg cells), but some Treg cells still maintained SETD2 expression (SETD2-sufficient Treg cells) comparable to the scramble shRNA group (Fig. 6q, r). GATA3 expression in SETD2-sufficient Treg cells in the SETD2-knockdown group was comparable to that in the scramble shRNA group (Fig. 6s). In contrast, GATA3 expression was dramatically downregulated in SETD2-null Treg cells compared with SETD2-sufficient Treg cells in SETD2-knockdown group (Fig. 6t). Together, the data indicate that SETD2 supports GATA3 expression in Treg cells and may promote their function to suppress Th2 cells in human CRC patients.

### Setd2 modulates promoters and intragenic enhancer activity of Treg cells

H3K27ac, which has been previously shown to be affected by Setd2, is considered to be a histone modification marking active promoters and enhancers[29,52]. Due to the scarcity of intestinal Treg cells, we took advantage of the CUT&Tag technology to assess the genome-wide H3K27ac profile in intestinal Treg cells purified from 2-week-old *Foxp3*$^{Cre-YFP}$*Setd2*$^{f/f}$ mice and control mice. Because the intestinal Treg cells from 2-week-old mice were mainly composed of Helios$^+$ Treg cells (Supplementary Fig. 5e), the data were less likely to be confounded by Helios$^-$RORγt$^+$ Treg cells differentially distributed in adult intestinal Setd2-deficient Treg cells compared with controls. Statistical analysis implied that genes with significantly increased H3K27ac peaks preferentially overlapped with genes with increased mRNA expression (Supplementary Fig. 6a and Supplementary Data 4). In parallel, genes with significantly decreased H3K27ac peaks preferentially overlapped with genes with decreased mRNA expression (Supplementary Fig. 6a and Supplementary Data 4). Considering that gene transcription may be determined by collective H3K27ac signals in the gene locus,

aggregated analysis was performed on the mRNA expression of top 200 significantly changed genes with their average H3K27ac signal calculated from different regions of the gene locus. We found that the H3K27ac level in promoters and gene bodies, where H3K36me3 is typically deposited, but not in distal intergenic regions, had a significant positive correlation with the mRNA expression of the corresponding genes (Fig. 7a–c and Supplementary Data 5). The affected genes included *Txnl4b*, which regulates pre-mRNA splicing, and *Cxcr6*, *Ccr9*, and *Stat1*, which have been reported to be related to chemotaxis or function of Treg cells (Fig. 7a, b and Supplementary Data 5)[53–56]. In particular, the H3K27ac level in an intragenic region of *Il1rl1* with previously identified H3K4me1 signal and validated to be a functional enhancer[57,58], with presented H3K36me3 signal, was found to be reduced in Setd2-deficient Treg cells (Fig. 7d). The data suggest that Setd2 may regulate target gene transcription including *Il1rl1* by increasing the activity of intragenic enhancers.

Surprisingly, H3K27ac modification in the GATA3 locus was not decreased in KO Treg cells (Fig. 7e). To further determine if Setd2 regulates GATA3 at the transcriptional level, we performed CUT&Tag analysis on RNA polymerase II (Pol II) binding landscape using purified splenic Treg cells isolated from IL-2c-treated mice. We confirmed that Treg cells from IL-2c-treated mice were enriched for tTreg cells and that ST2$^+$ and GATA3$^+$ tTreg cells were significantly decreased in the spleen of *Foxp3*$^{Cre-YFP}$*Setd2*$^{f/f}$ mice (Supplementary Fig. 6b–d). Consistently, genes with significantly increased Pol II peaks preferentially overlapped with genes with increased mRNA expression, and vice versa (Supplementary Fig. 6e and Supplementary Data 6). Moreover, Pol II binding in both promoters and gene bodies was significantly correlated with mRNA expression for the top 200 significantly changed genes indicated by the RNA-seq analysis, confirming that Setd2 can regulate gene expression through modulating gene transcription at both the initiation and elongation stages (Fig. 7f, g and Supplementary Data 5). The affected genes included *Txnl4b*, *Cxcr6*, *Ccr9*, and *Stat1*, the H3K27ac levels of which were similarly altered by the loss of Setd2 (Fig. 7f, g, Supplementary Fig. 6f–i, and Supplementary Data 6). Pol II binding at the *Il1rl1* locus was dramatically reduced (Fig. 7d and Supplementary Data 6). Nevertheless, no difference was found in Pol II associated with the GATA3 locus (Fig. 7e). Consistently, while *Il1rl1* mRNA expression was decreased in splenic Treg cells of IL-2c-treated *Foxp3*$^{Cre-YFP}$*Setd2*$^{f/f}$ mice, *Gata3* mRNA expression was unaltered (Fig. 7h, i). We reason that Setd2 may support GATA3 expression through translational or posttranslational mechanisms in Treg cells, although we cannot rule out the possibility that Setd2 may promote the GATA3 expression at the transcription level within a specific subpopulation of Treg cells.

## Discussion
Epigenetic modifications are intricately regulated during cell development, differentiation, and functional commitment, and are closely related to transcriptional programs of nonimmune and immune cells,

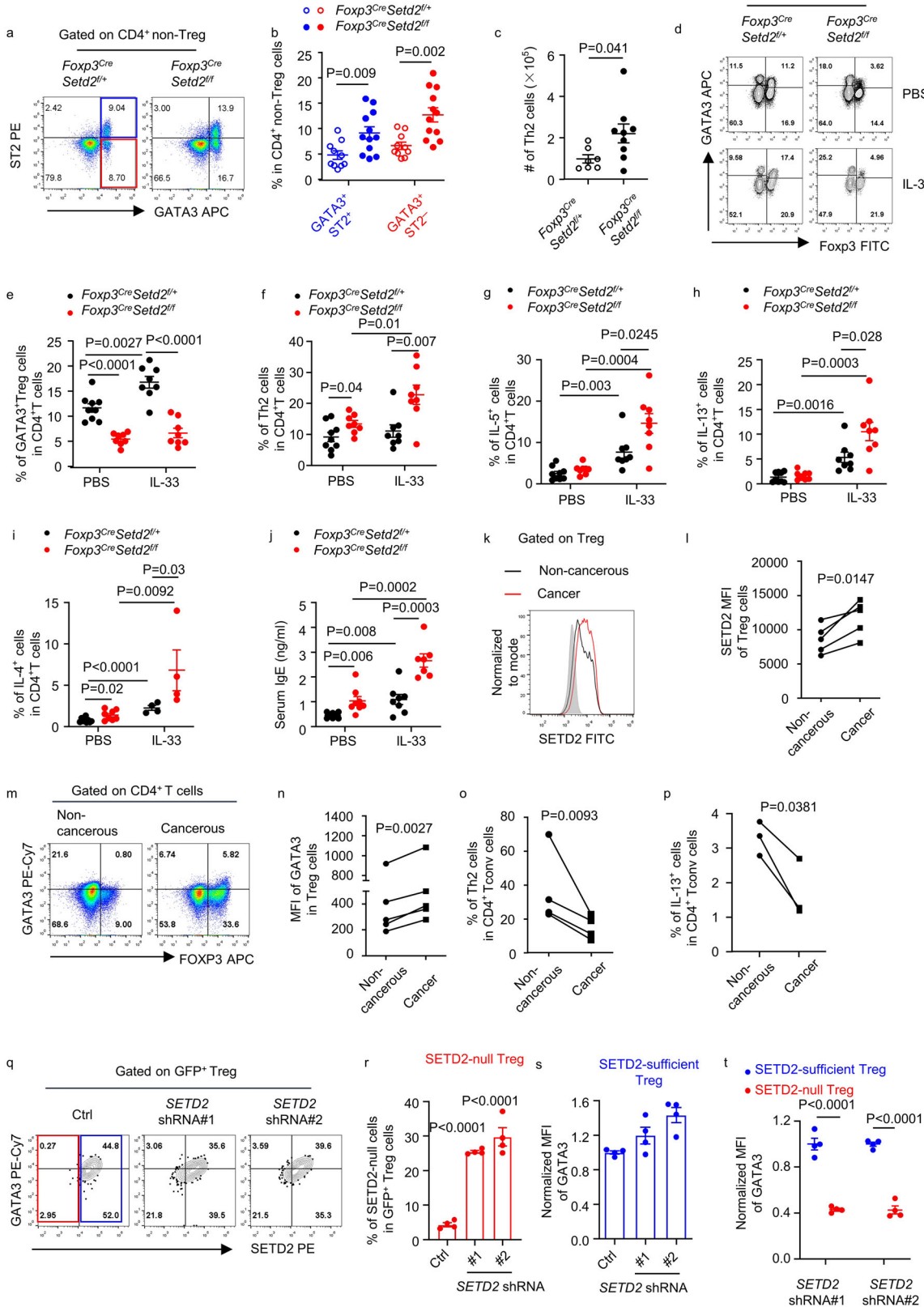

including Treg cells[25]. In Treg cells, the expression of these epigenetic modulators could alter with the status of cell activation or under pathological conditions. For example, Ezh2 mediating histone H3K27me3 modification is highly expressed by Treg cells and Ezh2 is upregulated in Treg cells with CD28 mediated co-stimulation[59]. Recently, the protein stability of Ezh2 has been shown to be disturbed

by Setd2[36], the expression of which has been found to be decreased in epithelial cells during colitis[60]. Furthermore, Nsd2 mediating histone H3K36me2 modification has been associated with the function of Treg cells to maintain fetal-maternal tolerance[61]. These findings indicate that the epigenetic modulators affecting histone modifications are critical in regulating the function of Treg cells and tissue homeostasis.

**Fig. 6 | Setd2 is critical for intestinal Treg cells to suppress Th2 responses. a**–**c** LI (large intestinal) LPLs were isolated. (Ctrl: $n = 10$; KO: $n = 12$. For analyzing total cell number, Ctrl: $n = 7$; KO: $n = 9$). **a** Flow cytometry analysis of ST2 and GATA3 expression in CD4$^+$ non-Treg cells. **b** Percentages of GATA3$^+$ST2$^+$ and GATA3$^+$ST2$^-$ cells in CD4$^+$ non-Treg cells. **c** Absolute number of Th2 cells (CD3$^+$CD4$^+$Foxp3$^-$ GATA3$^+$). **d**–**j** Mice were treated with PBS or IL-33. **d**–**i** LI LPLs were isolated. **d** Flow cytometry analysis of Foxp3 and GATA3 expression gated on CD4$^+$ T cells. **e**, **f** Flow cytometry analysis of percentages of GATA3$^+$ Treg cells (CD3$^+$CD4$^+$Foxp3$^+$GATA3$^+$) and Th2 cells in CD4$^+$ T cells. (PBS Ctrl $n = 9$; PBS KO $n = 8$; IL-33 $n = 8$ per group). **g**–**i** Flow cytometry analysis of expression of IL-5 (**g**), IL-13 (**h**), and IL-4 (**i**) in CD4$^+$ T cells. (IL-4 staining: PBS Ctrl $n = 9$; KO $n = 8$; IL-33 $n = 4$ per group.) **j** Serum IgE measured by ELISA. (PBS: $n = 8$ per group; IL-33 Ctrl $n = 8$; KO $n = 7$). **k**–**p** LPLs were isolated from patients with CRC. **p** Cells were stimulated with PMA and ionomycin. **k** Flow cytometry analysis on SETD2 expression gated on Treg cells (CD4$^+$FOXP3$^+$CD127$^-$). **l** MFI of SETD2 gated on Treg cells ($n = 5$). **m** Flow cytometry analysis on GATA3 and FOXP3 expression. **n** MFI of GATA3 ($n = 5$). **o**, **p** Percentage of Th2 cells (CD4$^+$FOXP3$^-$GATA3$^+$) ($n = 4$) (**o**) and IL-13$^+$ cells (**p**) ($n = 3$) in CD4$^+$ Tconv (CD4$^+$FOXP3$^-$) cells. **q**–**t** Human peripheral blood Treg cells were infected with retrovirus targeting SETD2 ($n = 4$ per group). **q** Flow cytometry analysis on GATA3 and SETD2 expression gated on GFP$^+$ Treg cells. The red box is SETD2-null Treg cells and the blue box is SETD2-sufficient Treg cells. **r** Percentage of SETD2-null cells in GFP$^+$ Treg cells. **s** Normalized MFI (to Ctrl as 1) of GATA3 gated on SETD2-sufficient Treg cells. **t** Normalized MFI (to SETD2-sufficient Treg cells as 1) of GATA3 gated on SETD2-sufficient Treg cells or SETD2-null Treg cells. **n**–**p** Connected lines are samples from the same patient. **b**, **c**, **e**–**j**, **r**–**t** Data were means ± SEM. **a**–**t** Representative of 2–5 independent experiments. Source data are provided as a Source Data file.

In this research, we report that Setd2 preferentially supports the homeostasis of GATA3$^+$ST2$^+$ tTreg cells, showing the specific transcriptional determination of Treg cells imprinted by Setd2. Considering Setd2-mediated H3K36me3 modification is broadly observed in actively transcribed genes, this specific transcriptional program imprinted by Setd2 in Treg cells is intriguing. We speculate that this specific manner of regulation is because Setd2 mainly affects the activities of cis-regulatory elements distributed in the promoters and gene bodies, where H3K36me3 is typically deposited. However, gene transcription may be determined by a group of cis-regulatory elements, including distal enhancers and enhancers upstream of a gene locus which may not be regulated by Setd2, in a cooperative way. Furthermore, Setd2 may exhibit H3K36me3-independent regulations by directly mediating protein methylation and stability[30,36]. These properties of Setd2 may allow it to imprint specific transcriptional programs of Treg cells.

The intestine harbors food antigens and trillions of microbiota that induces highly activated Tconv cells, which need to be controlled by Treg cells to prevent autoimmunity even under a steady state. We initially observed a reduction in the percentage of Treg cells in the GALTs but not in SLOs or other analyzed tissues. Interestingly, while Setd2 supports ST2$^+$ tTreg cells in both the intestine and other analyzed peripheral tissues, the decrease of GATA3$^+$ tTreg cells was observed in the large intestine but not in the fat, lung, or liver of *Foxp3$^{Cre-YFP}$Setd2$^{f/f}$* mice. This may suggest that IL-33-supported ST2$^+$GATA3$^+$ tTreg cells homeostasis dependent on Setd2 is more prominent in the intestine than in other organs under a steady state.

The Treg cells that are responsible for inhibiting specific CD4$^+$ conventional T responses seem to share expression of the same transcription factors, likely due to spatial proximity and/or competition for survival factors[4,5]. We showed that *Foxp3$^{Cre-YFP}$Setd2$^{f/f}$* mice had decreased GATA3$^+$ tTreg cells. Moreover, we found that *Foxp3$^{Cre-YFP}$Setd2$^{f/f}$* mice had enhanced IL-17 production by Treg cells and increased intestinal Th2 responses, as was similarly observed in *Foxp3$^{Cre-YFP}$Gata3$^{f/f}$* mice in previous studies[6–10]. Therefore, GATA3 deficiency in Treg cells of *Foxp3$^{Cre-YFP}$Setd2$^{f/f}$* mice possibly contributes to their reduced ability to inhibit Th2 responses. We showed that Setd2-deficient Treg cells had increased fate conversion in CD45RB$^{high}$T cell-induced colitis and during homeostatic expansion in *Rag2$^{-/-}$* mice. Previous studies indicate that GATA3 and ST2 are important for the maintenance of Treg stability during inflammation[6,19]. Therefore, it is likely that Setd2 prevents the fate conversion of Treg cells under inflammatory conditions by sustaining GATA3 and ST2. We also found that Setd2-deficient Treg cells had increased apoptosis, and blocking cell death partially rescued defective expansion of Setd2-deficient Treg cells in response to IL-2. So far, not much evidence supports the theory that GATA3 or ST2 could promote Treg survival. We speculate that Setd2 may support the survival of Treg cells through GATA3/ST2-independent mechanisms.

Although Setd2 was expressed at a similar level in mouse splenic Treg cells and intestinal Treg cells, we observed that SETD2 expression was increased in Treg cells from cancerous tissues compared with those from noncancerous tissues of CRC patients. Intriguingly, Setd2 expression in epithelial cells has been reported to be decreased in experimental colitis[60]. Mutation and deficiency of Setd2 is found to be associated with susceptibility to colorectal cancer[62]. These observations collectively suggest that SETD2 expression may be differentially regulated in different cell types under the same pathological conditions. The upstream signals regulating SETD2 expression in Treg cells remain to be elucidated. Consistent with the requirement of Setd2 expression in Treg cells for the inhibition of Th2 responses demonstrated in mice, we observed increased SETD2 expression in Treg cells together with decreased Th2 cells in cancerous tissues of CRC patients. Th2 cells have been indicated to have dual effects on cancers by affecting the formation of the vasculature, combating Th1 and CD8 cytotoxic functions, and recruiting eosinophils[63–65]. The precise function of Th2 cells in CRC is obscure. elucidating the effect of Th2 immunity in CRC will facilitate the development and assessment of potential therapeutic strategies directly or indirectly affecting SETD2 in Treg cells.

We have shown that Setd2 regulates gene transcription by modulating the activity of promoters and intragenic enhancers where H3K36me3 is preferentially deposited rather than enhancers of intergenic regions. This raises the possibility that Setd2-mediated H3K36me3 could affect the activity of adjacent promoters and enhancers. Deficiency of Setd2 can lead to both loss and gain of H3K27me3, H3K4me3, and H3K27ac modifications in a region-specific manner[29]. Nevertheless, deletion of Setd2 dominantly results in loss of DNA methylation in the gene bodies[29,31], which may be because DNA methyltransferases 3A/3B are recruited by H3K36me3[26]. In addition, Setd2 also affects H3K27me1[33]. The changes in histone modifications and DNA methylation induced by Setd2 could be the reason for the changed promoter/enhancer activity upon deletion of Setd2 in Treg cells. In addition, the diverse epigenetic modifications affected by Setd2 may explain why ST2/GATA3 expression was preferentially regulated by Setd2 in tTreg but not pTreg cells. tTreg and pTreg cells vary in transcription factor expression, DNA methylation status, and histone modifications, and these differences presumably lead to different consequences upon deletion of Setd2[25].

Although GATA3 protein expression was reduced in Setd2-deficient tTreg cells, we didn't observe decreased H3K27ac or Pol II accumulation marking active transcription in the *Gata3* locus, nor did we find decreased *Gata3* mRNA expression in IL-2c-treated Setd2-deficient splenic Treg cells. This possibly indicates that GATA3 in Treg cells may be regulated by Setd2 at the translational or posttranslational level. In consistent with this hypothesis, a previous study showed that H3K36me3 guided the *N*6-methyladenosine (m$^6$A) modification of mRNA, which had an impact on protein translation efficiency[66]. Further, it is also likely that the transcription regulation of *Gata3* by Setd2 may occur within only a specific subpopulation of tTreg cells, such as

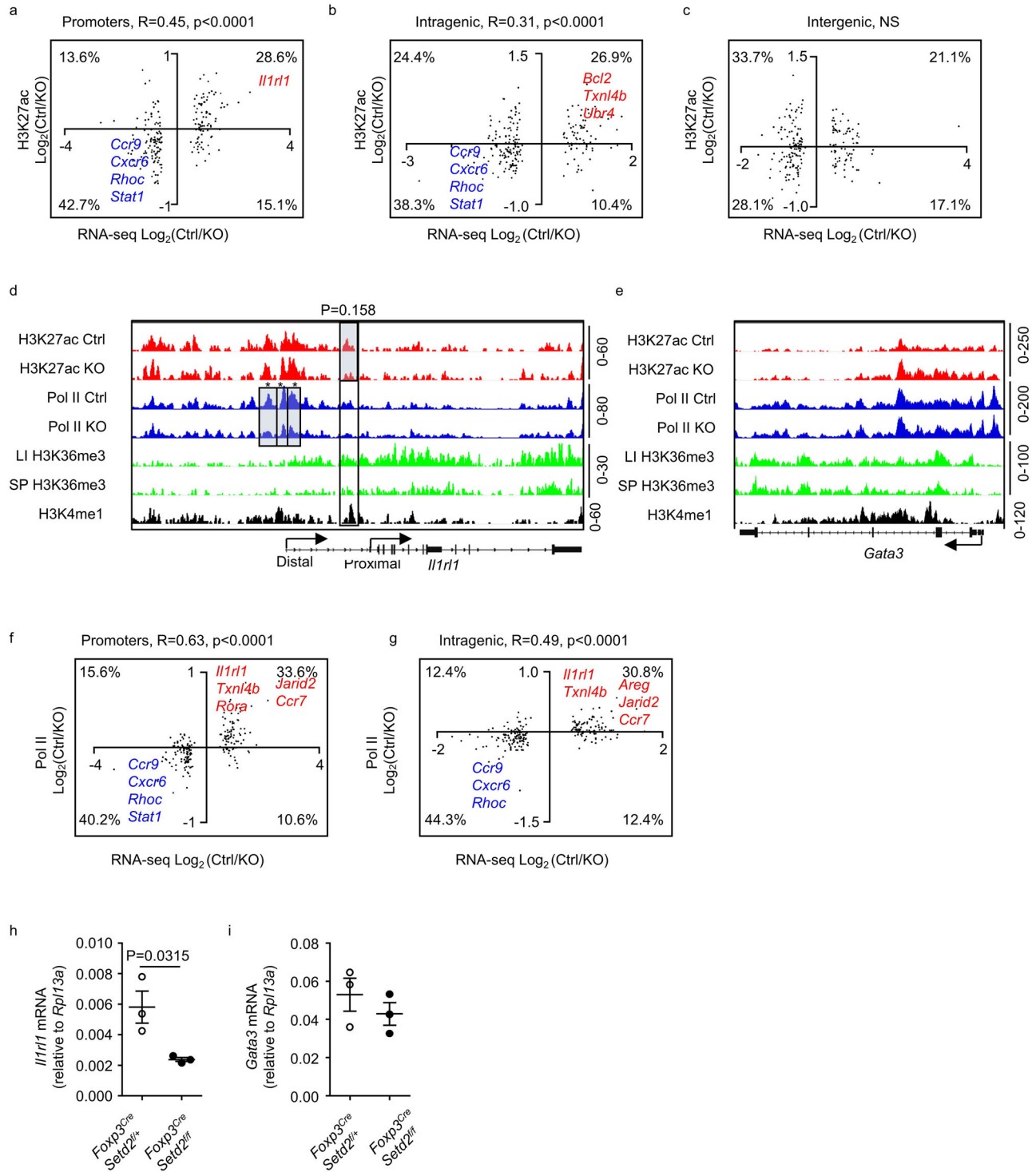

**Fig. 7 | Setd2 regulates gene transcription by modulating the activity of promoters and intragenic enhancers in tTreg cells. a–c** H3K27ac CUT&Tag analysis was performed with Treg cells purified from 2-week-old large intestinal LPLs of *Foxp3*^{Cre-YFP}*Setd2*^{f/+} (Ctrl) or *Foxp3*^{Cre-YFP}*Setd2*^{f/f} (KO) mice. **d, e** Genome browser tracks of H3K27ac, RNA Pol II CUT&Tag peaks, H3K36me3 peaks with published Treg H3K4me1(DRP003376) peaks at the *Il1rl1* and *Gata3* locus. Boxes highlight differentially expressed peaks with or close to reach statistically significant differences analyzed by DEseq2. **f, g** RNA Pol II CUT&Tag analysis was performed with splenic Treg cells nuclei of IL-2c-treated *Foxp3*^{Cre-YFP}*Setd2*^{f/+} (Ctrl) or *Foxp3*^{Cre-YFP}*Setd2*^{f/f} (KO) mice. **a–c, f, g** Correlation analysis was performed on top significantly changed genes identified by intestinal Treg RNA-seq using Log$_2$ (fold change) (Fc) of gene

mRNA expression and average Log$_2$ (fold change) (Fc) of H3K27ac peaks distributed at the promoters (**a**), H3K27ac peaks distributed at gene bodies (**b**), H3K27ac peaks distributed at intergenic regions (**c**), Pol II peaks distributed at the promoters (**f**), and Pol II peaks distributed at gene bodies (**g**). **a–c, f, g** Each dot represents one gene. Percentages indicate the proportions of genes distributed in each quadrant. R is the Pearson correlation coefficient. NS not significant. **h, i** Treg cells (CD3$^+$CD4$^+$YFP$^+$) were sorted from the spleen of IL-2c-treated mice of indicated genotypes. mRNA expression of *Il1rl1*(**h**) and *Gata3*(**i**) were analyzed by real-time RT-PCR (n = 3 per group). **h, i** Representative of two independent experiments. Data were means ± SEM. Source data are provided as a Source Data file.

the ST2$^+$GATA3$^+$ tTreg cells. As cells used for Pol II and H3K27ac CUT&Tag analyses were splenic Treg cells from IL-2c-treated mice and 2-week-old intestinal Treg cells, respectively, these cells were enriched for Helios$^+$ tTreg cells yet were heterogeneously including the ST2$^+$GATA3$^+$ tTreg, ST2$^+$GATA3$^-$ tTreg, ST2$^-$GATA3$^+$ tTreg, ST2$^-$GATA3$^-$ tTreg cells, and a small proportion of Helios$^-$ Treg cells. Regulation of GATA3 by Setd2 in different subsets of Treg cells is probably different, and Setd2-mediated molecular changes detected in one subset may be covered by the others in the H3K27ac and Pol II CUT&Tag analyses. This hypothesis remains to be further tested with tools allowing the purification of specific tTreg subpopulations, or with multi-omics approaches at the single-cell resolution.

## Methods

Study protocols used in this research complies with all relevant ethical regulations approved by "the institutional biomedical research ethics committee of the Shanghai Institutes for Nutrition and Health, Chinese Academy of Sciences" and "the Independent Ethics Committee of Shanghai Tongren Hospital".

### Mice

$Rag1^{-/-}$, $Rag2^{-/-}$ mice, $Foxp3^{Cre-YFP}$, and Thy1.1 mice were purchased from Jackson laboratory. $Il33^{-/-}$ mouse from Riken Center for Biosystems Research (accession number CDB0631K). $Setd2^{flox/flox}$ ($Setd2^{f/f}$) mouse was generated previously[29]. Mice labeled with $Foxp3^{Cre-YFP}$ ($Foxp3^{Cre}$) are $Foxp3^{Cre-YFP/y}$ for males and $Foxp3^{Cre-YFP/Cre-YFP}$ for females. Mice labeled with the genotype $Foxp3^{Cre-YFP/+}$ ($Foxp3^{Cre/+}$) are females. Mice used for in vivo studies were littermates, age-matched, and were 6–12 weeks old unless otherwise specified. Both male and female mice were used unless otherwise noted. All mice used in this study are on a C57BL/6 background, maintained in specific pathogen-free conditions, fed with a plain commercial diet (Silaikang, Shanghai), and housed in corn-cob-bedding cages in a room with the light-dark cycle (lights on at 6:00 and off at 18:00). The mice were kept at a constant temperature of $22 \pm 3 °C$ and a relative humidity of $35 \pm 5\%$. All animal experiments were performed in compliance with the "Guide for the Care and Use of Laboratory Animals" and approved by the institutional biomedical research ethics committee of the Shanghai Institutes for Nutrition and Health, Chinese Academy of Sciences, under the protocol numbers SINH-2020-QJ-2, SINH-2021-QJ-1, and SINH-2022-QJ-1.

### Isolation of mouse intestinal lamina propria lymphocytes (LPLs)

Large intestines were dissected and fat tissues were removed. Intestines were cut open longitudinally and washed in PBS. Intestines were then cut into 3-cm-long pieces, washed, and shaken in PBS containing 1 mM dithiothreitol (DTT) for 10 min at RT (room temperature). Intestines were incubated with shaking in PBS containing 30 mM ethylenediaminetetraacetic acid (EDTA) and 10 mM HEPES at 37 °C for 10 min for two cycles. The tissues were then digested in the RPMI 1640 medium (Thermo Fisher Scientific) containing DNase I (150 ug/ml, Sigma) and collagenase VIII (200 U/ml, Sigma) at 37 °C in a 5% $CO_2$ incubator for 1.5 h. The digested tissues were homogenized by vigorous shaking and passed through a 100-um cell strainer. Mononuclear cells were then harvested from the interphase of an 80 and 40% Percoll gradient after a spin at 2500 rpm for 20 min at RT. Isolating LPLs from 1-week-old mice was performed as the same approach above, except that the DTT incubation step was skipped, and only one cycle of EDTA incubation was performed. Isolating LPLs from 2-week-old mice was performed as the same approach above, except that only one cycle of EDTA incubation was performed.

### Cell suspension preparation from lymphoid organs of mice

Cell suspensions were prepared from the thymus, spleen, inguinal lymph nodes, and mesenteric lymph nodes by gentle mechanical disruption followed by passing through a 50-um nylon mesh.

### Isolation of mononuclear cells from lung, fat, and liver

For isolation of mononuclear cells from the lung, tissues were dissected, and blood clots, fat tissues, and bronchus were discarded. Lung tissues were cut into pieces and digested with 5 ml RPMI 1640 medium (Thermo Fisher Scientific) containing DNase I (75 ug/ml, Sigma-Aldrich) and collagenase VIII (250 U/ml, Sigma-Aldrich) at 37 °C for 40 min. The digested tissues were homogenized by vigorous shaking and passed through at 70 um cell strainer. Mononuclear cells were then harvested from the interphase of a 40 and 80% Percoll gradient after a spin at 2500 rpm for 20 min at RT.

For the isolation of mononuclear cells from fat, epididymal white adipose tissues were dissected from a male mouse. Tissues were cut into small pieces and then digested in 4 ml RPMI 1640 medium containing collagenase II (400 U/ml, Sigma-Aldrich) at 37 °C for 20 min. The digested tissues were homogenized by vigorous shaking and passed through a 100-um cell strainer. Mononuclear cells were then harvested after a spin at 2000 rpm for 10 min at RT.

For isolation of mononuclear cells from the liver, tissues were dissected and mechanically disrupted, followed by passing through a 100-um nylon mesh. The cell suspension was spun at $40 \times g$ for 3 min and the supernatant was harvested. Mononuclear cells were then harvested from the interphase of a 40 and 80% Percoll gradient after a spin at 2500 rpm for 20 min at RT.

### Mononuclear cell isolation from human samples

Human samples were obtained from the Department of General Surgery of Tongren Hospital with informed consent. The study was approved by the Independent Ethics Committee of Shanghai Tongren Hospital (approval numbers 2020-043-01 and 2019-052-01) for obtaining human PBMC and human colon tissues. Human samples were obtained from the Department of General Surgery of Tongren Hospital with informed consent. Human blood samples were collected from both female and male healthy donors, with ages ranging from 25 to 35. Human cancerous tissues and noncancerous tissues (≥10 cm away from the tumor) were collected from colorectal cancer patients during surgery. The study was performed in strict compliance with all institutional ethical regulations. Gender: Female ($n = 2$, 25%), Male ($n = 6$, 75%); Age at diagnosis: <70 ($n = 5$, 62.5%), >70 ($n = 3$, 37.5%); Pathological diagnosis: rectal tumor ($n = 6$, 75%), colon tumor ($n = 2$, 25%). Peripheral blood mononuclear cells (PBMC) from healthy adults were isolated via Lymphoprep (STEMCELL Technologies). For human LPL isolation, fresh tissues were washed with a pre-cold medium and the fat tissues and muscle were removed. The intestinal mucosa, about 2 cm × 2 cm was cut into pieces and shaken in PBS containing 1 mM DTT for 10 min at RT. Then, tissue pieces were further incubated in a shaker at 220 rpm in PBS containing 30 mM EDTA at 37 °C for 10 min twice. The tissues were then digested in 10 ml RPMI 1640 medium containing DNase I (Sigma-Aldrich, 300 ug/ml) and collagenase VIII (Sigma-Aldrich, 400 U/ml) at 37 °C in the cell culture incubator with 5% $CO_2$ for 6 h. Mononuclear cells were then harvested from the interphase of a 40 and 80% Percoll gradient after a spin at 2500 rpm for 20 min at room temperature.

### Western blot and quantification with ImageJ

Protein lysates from an equal number of sorted Treg cells or CD4$^+$ non-Treg cells from $Foxp3^{Cre-YFP}Setd2^{f/+}$ or $Foxp3^{Cre-YFP}Setd2^{f/f}$ mice were loaded per lane, resolved by 10% SDS-PAGE, transferred to polyvinylidene fluoride membranes and detected by immunoblotting with the Tanon-5200Multi system. Antibodies used were as follows: α-H3K36me3 (1:1000, Abcam), α-H3 (1:1000, Cell Signaling Technology). Quantification of signal strength was performed with ImageJ.

### Flow cytometry

Anti-mouse CD16/32 antibody was used to block the nonspecific binding to Fc receptors before all surface stainings. Dead cells were stained with live and dead violet viability kit (Invitrogen) and were

gated out in all analysis. Surface antibody staining was performed in PBS containing 2 mM EDTA and 0.1% BSA. For detection of nuclear factors, cells were fixed and permeabilized using a Mouse Regulatory T Cell Staining Kit (Thermo Fisher Scientific, 00-5523-00). For cytokine staining, cells were stimulated by PMA (50 ng/ml, Sigma) and ionomycin (500 ng/ml, Sigma) for 4 h. Brefeldin A (2 ug/ml, Sigma) was added for the last 2 h before cells were harvested for analysis. Flow cytometry data were collected using the Gallios flow cytometer (Beckman Coulter) and analyzed with FlowJo software (TreeStar). Sorting of cells were performed on the BD FACSAria III sorter (BD biosciences) or on the MoFlo Astrios sorter (Beckman Coulter). All antibodies used were listed in Supplementary Data 7.

### Bone marrow chimeras
Bone marrow cells from the femurs of Thy1.1$^+$ *Foxp3*$^{cre-YFP}$*Setd2*$^{f/+}$ mice were mixed with those from Thy1.2$^+$ *Foxp3*$^{cre-YFP}$*Setd2*$^{f/f}$ mice at a ratio of 1:1 and then were intravenous injected (5 × 10$^6$ cells per mouse) into lethally irradiated *Rag1*$^{-/-}$ mice (550 rads twice with 5 h interval). Chimeric mice were sacrificed for analysis 6-8 weeks after transfer.

### In vivo treatment of mice
IL-2/α-IL-2 complexes (IL-2c) were prepared as follows. About 1 ug recombinant mouse IL-2 (Peprotech) was mixed with 5 ug α-IL-2 (Bioxcell, clone JES6-1) and incubated at 37 °C for 30 min. Mice were injected intraperitoneally (i.p.) with IL-2c for 3 consecutive days and sacrificed for analysis 48 h later.

In the IL-2c treatment experiment, for blockade of cell death in vivo, 10 mg/kg Z-VAD-FMK (Apexbio Technology) and 5 mg/kg Necrostatin-1 (Abmole) or DMSO in PBS were injected i.p. daily for 5 days. In the IL-2c treatment experiment, for blockade of FASL in vivo, 400 ug IgG (Bioxcell) or α-FASL (Bioxcell) in PBS was injected i.p. on day −1, 1, and 3, and mice were sacrificed for analysis on day 5.

For IL-33 treatment in vivo, mice were i.p. injected with 500 ng recombinant murine IL-33 (BioLegend) or PBS for 4 consecutive days and sacrificed for analysis on day 5.

### Transfer of T cells to *Rag2*$^{-/-}$ mice
To induce colitis in *Rag2*$^{-/-}$ mice, CD3$^+$CD4$^+$ YFP$^+$ Treg cells from the spleen of Thy1.2$^+$ *Foxp3*$^{Cre-YFP}$*Setd2*$^{f/+}$ or *Foxp3*$^{Cre-YFP}$*Setd2*$^{f/f}$ mice and naive CD3$^+$CD4$^+$CD45RB$^{hi}$ YFP$^-$ Tconv cells from the spleen of Thy1.2$^+$ or Thy1.1$^+$ *Foxp3*$^{Cre-YFP}$*Setd2*$^{f/+}$ mice were prepared by cell sorting. Each *Rag2*$^{-/-}$ recipient mouse was intravenously injected with 4 × 10$^5$ Tconv cells alone or together with 2 × 10$^5$ Treg cells. The body weight of mice was monitored weekly.

For analysis of Treg homeostasis in *Rag2*$^{-/-}$ mice, CD3$^+$CD4$^+$YFP$^+$ Treg cells from *Foxp3*$^{Cre-YFP}$*Setd2*$^{f/+}$ or *Foxp3*$^{Cre-YFP}$*Setd2*$^{f/f}$ mice were sorted by flow cytometry. Each *Rag2*$^{-/-}$ recipient mouse was intravenously injected with 2.5 × 10$^5$ Treg cells and sacrificed for analysis 4 weeks later.

### Histological analysis
Tissues from the proximal colon were dissected and fixed with 4% paraformaldehyde. Tissues were then embedded in paraffin, sectioned at 5 um, and stained with H&E. Sections were then blindly analyzed using the light microscope (Olympus), and scored using four parameters used, include (i) the degree of inflammatory infiltration in the LP, range 1–3; (ii) goblet cell loss as a marker of mucin depletion, range 0–2; (iii) mucosal erosion to frank ulcerations, range 0–2; and (iv) submucosal spread to transmural involvement, range 0–2. The severity of inflammation in sections of the colon was based on the sum of the scores in each parameter (maximum score = 9).

### 2,4,6-Trinitrobenzenesulfonic acid (TNBS)-induced colitis
*Foxp3*$^{cre-YFP}$*Setd2*$^{f/+}$ and *Foxp3*$^{cre-YFP}$*Setd2*$^{f/f}$ mice were treated with TNBS (Sigma-Aldrich). On day 0, pelltobarbitalum natricum anesthetized

mice were carefully shaved a 1.5 × 1.5 cm field of the skin and 150 ul of vehicle control or the TNBS presensitization solution (1% (w/v) TNBS in acetone/olive oil mixture (4:1 volume ratio)) was applied to the shaved skin. On day 7, anesthetized mice were intra-rectal treated with 100 ul TNBS solution (2.5% (w/v) TNBS in 50% ethanol) or vehicle control by inserting the catheter into the colon 4 cm away from the anus. Remove the catheter gently from the colon and keep the mice with the head down in a vertical position for 30 min and then returned to their cages. Mice were sacrificed for analysis on day 9.

### Cell culture medium
RPMI 1640, DMEM, 10% fetal bovine serum, L-glutamine, and penicillin/streptomycin used for making a complete medium for cell culture were ordered from Thermo Fisher Scientific. IMDM and β-Mercaptoethanol (β-ME) were from Sigma-Aldrich. IMDM complete medium was prepared using IMDM medium supplemented with 10% fetal bovine serum, 2 mM L-glutamine, 50 uM β-ME, and 1% penicillin/streptomycin. DMEM complete medium was prepared using DMEM supplemented with 10% fetal bovine serum, 2 mM L-glutamine, and 1% penicillin/streptomycin. RPMI 1640 complete medium was prepared using RPMI 1640 medium supplemented with 10% fetal bovine serum, 2 mM L-glutamine, 50 uM β-ME, and 1% penicillin/streptomycin.

### In vitro Treg suppression assay
CD4$^+$CD62L$^{high}$CD44$^{low}$YFP$^-$ T-responder cells sorted from the spleen of Thy1.1$^+$ *Foxp3*$^{Cre-YFP}$ mice were labeled with CellTrace™ Far Red Cell Proliferation Kit (Thermo Fisher Scientific, C34572). CD4$^+$YFP$^+$ Treg cells were sorted from the spleen or large intestine of Thy1.2 *Foxp3*$^{cre-YFP}$*Setd2*$^{f/+}$ or Thy1.2 *Foxp3*$^{cre-YFP}$*Setd2*$^{f/f}$ mice. About 5 × 10$^4$ T-responder cells and Treg cells were cultured in 96-well round bottom plate at different ratios containing 2 × 10$^5$ mitomycin C (Selleck)-treated Thy1.2 WT splenocytes as antigen-presenting cells, soluble α-CD3 (1 ug/ml) and IL-2 (10 ng/ml). Cells were analyzed on day 3.

### GATA3 and ST2 overexpression in vitro
The full-length mouse GATA3 and ST2 cDNA were cloned into the MSCV-Thy1.1 retroviral vector. Phoenix cells were transfected with retroviral plasmids and the packaging plasmid 10A1 using polyethylenimine (PEI, Polysciences). Viral supernatant was collected after transfection. Splenic Treg cells (CD3$^+$CD4$^+$YFP$^+$) FACS-sorted from *Foxp3*$^{cre-YFP}$*Setd2*$^{f/+}$ and *Foxp3*$^{cre-YFP}$*Setd2*$^{f/f}$ mice were cultured with IMDM medium in anti-Hamster antibody (MP Biomedicals, antibody protein 40 ug/ml) coated 96-well plate supplied with soluble α-CD3 (0.5 ug/ml, Bioxcell), α-CD28 (1 ug/ml, Bioxcell), IL-2 (50 ng/ml, Peprotech), IL-33 (10 ng/ml, BioLegend) for 72 h. Then, spin-infection using virus supernatant was performed every 24 h twice with 8 ug/ml polybrene (Sigma-Aldrich) and with the presence of α-CD3, α-CD28, IL-2, and IL-33. Cells were analyzed 48 h later after the second round of spin infection.

### Retrovirus production and transduction of human Treg cells
MSCV-LTRmiR30-PIG (LMP) is a retroviral vector designed for the dual expression of GFP and short hairpin RNAs (shRNA) (Open Biosystems). *SETD2* shRNA#1 and *SETD2* shRNA#2 were generated by ligation of synthesized DNA oligonucleotides targeting coding regions of human *SETD2* to the LMP vector. Phoenix cells were transfected with retroviral plasmids and the packaging plasmid 10A1 using polyethylenimine (PEI, Polysciences). Viral supernatant was collected after transfection. FACS-sorted human peripheral blood Treg cells (CD4$^+$CD127$^-$CD25$^+$ cells) were cultured in α-CD3 (2 ug/ml, eBioscience)/α-CD28 (2 ug/ml, eBioscience) coated 96-well plate with RPMI 1640 complete medium containing recombinant human IL-2 (10 ng/ml, Peprotech), recombinant human IL-33 (10 ng/ml, Peprotech), α-human-IFN-γ (4 ug/ml, BD biosciences) for 5–8 days. Then, spin-infection was performed at 2500 rpm for 1.5 h at 30 °C and cultured with an additional 24 h in an

incubator with 5% $CO_2$, in virus supernatant containing 8 ug/ml poly-brene (Sigma-Aldrich), α-CD3, α-CD28, IL-2, IL-33, and α-IFN-γ. Cells were analyzed 36–48 h later after the second round of spin infection. The target sequences of *SETD2* are: #1 5′-ACTCACGGTGTTATGAA-TAAG-3′; #2 5′-TGTCTGGAACTCATACAGAAC-3′.

## Detection of serum IgE using ELISA

IgE from mouse serum was detected using a flat-bottom immuno 96-well plate with an unconjugated capture antibody (goat-anti-mouse IgE, 1 ug/ml, SouthernBiotech) and an HRP-conjugated detection antibody (goat-anti-mouse IgE, 1: 8000, SouthernBiotech). Absorbance at a wavelength of 450 and 570 nm was analyzed with a spectrophotometer (BioTek). The concentration of IgE in the serum was calculated according to the standard curve generated with the IgE standard (SouthernBiotech).

## Chromatin immunoprecipitation sequencing (ChIP-seq) on H3K36me3

CD3$^+$CD4$^+$YFP$^+$ Treg cells from the spleen and large intestine were sorted from Foxp3$^{Cre-YFP}$ mice (pooled cells from seven female and seven male mice) by flow cytometry. ChIP-sequencing on H3K36me3 was performed by Active Motif. Lymphocytes were fixed with 1% formaldehyde for 15 min and quenched with 0.125 M glycine. Chromatin was isolated by the addition of lysis buffer, followed by disruption with a Dounce homogenizer. Lysates were sonicated using the EpiShea Probe Sonicator (Active Motif, cat # 53051) with an EpiShea Cooled Sonication Platform (Active Motif, cat # 53080) and the DNA sheared to an average length of 300–500 bp. Genomic DNA (Input) was prepared by treating aliquots of chromatin with RNase, proteinase K, and heat for de-crosslinking (overnight at 65 °C) followed by ethanol precipitation. Pellets were resuspended and the resulting DNA was quantified on a NanoDrop spectrophotometer. Extrapolation to the original chromatin volume allowed quantitation of the total chromatin yield. Equal amounts of chromatin (1 ug) from the splenic or intestinal Treg cells were used per immunoprecipitation (IP) reaction (H3K36me3 (Active Motif 61101). Chromatin from splenic Treg cells was used as input. Single-end 75 nt sequencing was performed with Illumina NextSeq 500. BAM files were generated based on chastity-filtered reads with phred+33 quality scores of FASTQ files aligned to mouse genome (mm10). Then, peak calling was performed using the Galaxy web platform public server (https://usegalaxy.org) using MACS2 with a threshold *p* value of 0.01 as a cut-off based on input. For comparison of H3K36me3 signals, peaks of identified splenic and intestinal Treg cells were merged with "bedtools Multiple Intersect" followed by "bedtools MergeBED" function. Peak annotation was performed with "ChIPseeker". Bigwig files were generated using the Bamcoverage function normalized to reads per kilobase per million (RPKM). The cumulative curve of the H3K36me3 signal at the gene locus was performed with plotProfile based on bigwig files. Normalized counts for identified peaks were analyzed with multi-BigwigSummary. Fold change of H3K36me3 of splenic Treg cells compared with intestinal Treg cells was calculated based on the summed counts of all H3K36me3 peaks identified at a specific gene locus.

## Intestinal and splenic Treg signature genes

Intestinal and splenic Treg signature genes were analyzed based on published data of splenic Treg cells and colonic Treg cells from GSE68009 using GEO2R. The cutoff for *p* value was 0.05 and the cutoff value for fold change was 2.

## RNA-seq analysis

About $5 \times 10^5$ sorted large intestinal Treg cells (CD3$^+$CD4$^+$YFP$^+$ cells) or $1 \times 10^6$ sorted splenic Treg cells (CD3$^+$CD4$^+$YFP$^+$ cells) were pooled from littermate Foxp3$^{Cre-YFP}$Setd2$^{f/+}$ or Foxp3$^{Cre-YFP}$Setd2$^{f/f}$ mice. Each biological replicate was pooled from 2–3 mice. Cells were lysed in Trizol (Invitrogen) and total RNA was extracted. Library construction

and sequencing was performed by BGI Genomics, BGI-Wuhan. RNA were treated with DNase I and purified with magnetic beads with Oligo (dT). RNA were mixed with the fragmentation buffer and were fragmented into short fragments. Then cDNA was synthesized using the mRNA fragments as templates. Short fragments were purified and resolved with EB buffer for end reparation and single nucleotide A (adenine) addition. After that, the short fragments were connected with adapters. Suitable fragments were selected for the PCR amplification as templates. Agilent 2100 Bioanalyzer and ABI StepOnePlus Real-Time PCR System are used for quantification and qualification of the sample library. About 150 bp paired sequencing was performed using Illumina HiSeq4000. For analysis of intestinal Treg RNA-seq data, reads were mapped to Mouse Genome Assembly GRCm38 by STAR v2.5. Gene expression quantification was called by RSEM v1.2 with default parameters on the GENCODE mouse M16 gene annotation file. Differential expression analysis was performed by Bioconductor package edgeR v3.18.1. Splenic Treg RNA-seq data were analyzed using the Galaxy web platform public server (https://usegalaxy.org) mapped to Mouse Genome Assembly GRCm38. Briefly, quality-filtered reads were mapped to the mouse genome (mm10) using "HISAT2" (Galaxy Version 2.2.1). Gene raw counts were analyzed by "FeatureCounts" (Galaxy Version 2.0.1) and genes were annotated by "annotateMyIDs" (Galaxy Version 3.12.0). Differential gene expression was analyzed by edgeR (Galaxy Version 3.34.0). For both splenic and intestinal Treg cells, significantly changed genes were chosen according to three criteria: (1) significance level $p < 0.05$; (2) The genes were protein-coding genes. (3) expression level average FPKM values bigger than 5 in either treatment or control groups. Volcano plots and heatmaps based on the Z-score of FPKM value or fold change of mRNA expression were generated with the software GraphPad Prism 8.0. The standard score of a raw score x is $Z = \frac{x-\mu}{\sigma}$, where μ is the mean of the FPKM value of each sample and σ is the standard deviation of the FPKM value of each sample. Gene set enrichment analysis (GSEA) was performed using GSEA software (Broad Institute) based on customized gene lists generated from the following database. ST2$^+$ Treg signature was top 200 significantly higher expressed genes, ranked by *p* value, in ST2$^+$ Treg cells than ST2$^-$ Treg cells of published dataset GSE136556. RORγt$^+$ Treg signature was top 500 significantly higher expressed genes, ranked by fold change, in colonic RORγt$^+$ Treg cells than colonic RORγt$^-$ Treg cells from the published dataset GSE68009.

## H3K27ac and RNA Pol II CUT&Tag analysis

H3K27ac and RNA Pol II CUT&Tag library preparation was performed using Hyperactive In-Situ ChIP Library Prep Kit for Illumina (Vanzyme, TD901) according to the manufacturer's instructions. Briefly, for H3K27ac CUT&Tag analysis, $6 \times 10^4$ Treg cells (CD3$^+$CD4$^+$YFP$^+$ cells) as one biological replicate were pooled and sorted by flow cytometry from the large intestine of 3–5 mice of 2-week-old littermate Foxp3$^{Cre-YFP}$Setd2$^{f/+}$ or Foxp3$^{Cre-YFP}$Setd2$^{f/f}$ mice. For RNA Pol II CUT&Tag analysis, Treg cells (CD3$^+$CD4$^+$YFP$^+$ cells) were pooled and sorted by flow cytometry from the spleen of two mice of IL-2c-treated adult littermate Foxp3$^{Cre-YFP}$Setd2$^{f/+}$ or Foxp3$^{Cre-YFP}$Setd2$^{f/f}$ mice, and $8 \times 10^4$ nuclei were used as one biological replicate. Nuclei were prepared as previously described[67], cells were pelleted at 1400×*g* for 5 mins at 4 °C, supernatant was discarded, and cells were resuspended in 200 ul cold nuclear extraction (NE) buffer (20 mM HEPES-KOH, pH 7.9, 10 mM KCl, 0.5 mM Spermidine, 0.1% TritonX-100, 20% glycerol, and 1 × Protease inhibitor cocktail) for 10 min on ice. Nuclei were pelleted at 1400×*g* for 5 mins at 4 °C. Then, cells or nuclei were washed in Wash Buffer and attached to activated ConA-coated magnetic beads. The bead-bound cells were incubated with 50 ul Antibody Buffer containing the appropriate primary antibody (rabbit-anti-H3K27ac, 2 ug, ABCAM; mouse-anti-RNA Pol II, 0.5 ug, Active Motif; control rabbit IgG, 2 ug, ABCAM or control mouse IgG1, 0.5 ug, SouthernBiotech) for 2 h in RT. Cells were then incubated with a secondary antibody (Goat-anti-Rabbit

IgG antibody or Goat-anti-Mouse IgG antibody) diluted (1:100) in Dig-Wash buffer at RT for 60 min on a rotating platform. After the final wash, 100 ul of pA-Tn5 solution was added to the bead-bound cells and the cells were incubated at RT for 1 h on a rotating platform. Next, cells were resuspended in 100 ul Tagmentation buffer and incubated at 37 °C for 70 min. To stop tagmentation, 10 ul of 0.5 M EDTA, 3 ul of 10% SDS, and 2.5 ul of 20 mg/ml Proteinase K was added to 100 ul of the sample, which was incubated overnight at 37 °C. The next day, DNA was extracted using phenol (Sangon Biotech) -chloroform-alcohol and dissolved in 25 ul water. Library amplification was performed using TruePrep Index Kit V2 for Illumina (Vanzyme, TD202). A total volume of 50 ul of the sample was placed in a thermocycler using the following program: 72 °C for 3 min; 98 °C for 30 s; 15 cycles of 98 °C for 15 s, 60 °C for 30 s and 72 °C for 30 s; 72 °C for 5 min and hold at 4 °C. PCR products was purified with 1.2 × volumes of VAHTS DNA Clean Beads (Vanzyme) and eluted in 22 ul of ddH$_2$O. Libraries were sequenced on an Illumina NovaSeq platform and 150 bp paired-end reads were generated.

### CUT&Tag data analysis and integrated analysis with RNA-seq
Raw sequence reads were initially processed by FastQC for quality control, and then adapter sequences and poor-quality reads were removed. Mapping, peak calling, and differential analysis was performed using the Galaxy web platform public server (https://usegalaxy.org). Quality-filtered reads were mapped to the mouse genome (mm10) using Bowtie2, and only uniquely mapped reads were kept. Duplicates reads were removed with MarkDuplicates. Peak calling was done using MACS2 with a $p$ value of 0.01 as the cutoff against IgG control. Identified peaks were merged with "bedtools Multiple Intersect" followed by "bedtools MergeBED" function. Peak annotation was performed with "ChIPseeker". Bigwig files were generated using the Bamcoverage function normalized to reads per kilobase per million (RPKM). Raw counts based on the BED file of merged peaks were extracted using bedtools MultiCovBed. Differentially expressed peaks were analyzed using DESeq2. Visualization of read count data was performed by converting raw bam files to bigwig files using IGV tools.

For integrated analysis of CUT&Tag and RNA-seq data, the correlation of mRNA expression with H3K27ac or RNA Pol II signals was performed using Log$_2$ (fold change) of mRNA expression with average Log$_2$ (fold change) of signals in promoter and 5′UTR, genebody (exon + intron + 3′UTR), or intergenic regions annotated by ChIPseeker. The average fold change of signals was calculated by a fold of all observed peaks signals divided by the number of observed peaks, regardless of significance in differential expression. The top 200 significantly changed genes in RNA-seq analysis, ranked by $p$ value, were used for correlation analysis with H3K27ac or RNA Pol II signals. Pearson correlation analysis was performed with GraphPad Prism 8.0.

### Quantitative real-time RT-PCR
RNA was isolated with a Trizol reagent (Invitrogen). cDNA was synthesized using the GoScript Reverse Transcription kit (Promega, A5001). Real-time PCR was performed using SYBR Green (ROX). Reactions were run with the QuantStudio6 Q-PCR System (Agilent). The results were displayed as relative expression values normalized to *Rpl13a*. The sequences of quantitative PCR primers for the genes examined are listed below: *Rpl13a* forward primer 5′-CGGAGGGG-CAGGTTCTG−3′; *Rpl13a* reverse primer 5′-AGCGTACGACCACCACCTT−3′; *Gata3* forward primer 5′-AGAGGTGGACGTACTTTTTAAC−3′; *Gata3* reverse primer 5′-AGAGATCCGTGCAGCAGAG−3′; *Il1rl1* forward primer 5′-TCAATTCACACACGCGGAGA−3′; *Il1rl1* reverse primer 5′-ATCTGCCACAGGACATCAGC−3′.

### Statistical methods
Statistical analyses were performed using GraphPad Prism 8.0. Figure 5j–m CTRL and KO cells in the same groups, Fig. 5n, o CTRL or

KO cells before and after overexpression of ST2 or GATA3, Fig. 6l, n–p, were performed with two-tailed paired Student's $t$-test. Other statistical analyses were all performed with two-tailed unpaired Student's $t$-test except for the following data were analyzed with nonparametric Mann–Whitney test: Fig. 1n (MLN group), Figs. 2e, g, i, 3h, i (None vs Ctrl Treg cells), Figs. 3j, 5c (2w group), Fig. 6i (IL-33 group), and Supplementary Fig. 4m. Data used for unpaired Student's $t$-test passed the Kolmogorov–Smirnov or the Shapiro–Wilk normality test.

### Reporting summary
Further information on research design is available in the Nature Portfolio Reporting Summary linked to this article.

## Data availability
In this research: The Treg H3K36me3 ChIP-seq data, Treg RNA-seq data, Treg H3K27ac CUT&Tag, and Treg RNA Pol II CUT&Tag data generated in this study have been deposited in National Omics Data Encyclopedia (NODE) database under accession code OEP002600 and the Gene Expression Omnibus public database (GEO) database under accession code GSE182845. Spleen Treg RNA-seq data generated in this study have also been deposited in the NODE database under accession code OEP003364. The published dataset used by this research: Spleen and colonic signature genes and RORγt$^+$ Treg signature genes were analyzed from GSE68009[68]; ST2$^+$ Treg signature genes were analyzed from GSE136556[44]; Treg H3K4me1 ChIP-seq data were analyzed from NCBI SRA database with SRA accession number is DRP003376[69]. Source data are provided with this paper. Information required for reanalyzing data from this paper is available from the corresponding author upon request. Source data are provided with this paper.

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

## Acknowledgements

The authors would like to thank Dr. Jun Qin, Dr. Gang Wang, and Dr. Mofang Liu for their help in this research. We would like to thank Yanwen Ye and Dr. Shuangshuang Wei for their technical support on this research. We thank Xiang Miao, Kai Wang, Konglun Pan, Xinhui Yan, Yujia Zhai, Yifan Bu, and Jiayu Wu from the Institutional Center for Shared Technologies and Facilities of SINH, CAS, for technical assistance. This study was supported by grants 2020YFA0509103 (to J.Q.), 2019YFA0802502 (to J.Q.), from the Ministry of Science and Technology of China, grant 32022027 (to J.Q.), and 31970860 (to J.Q.), 82070730 (to H.S.), 81672083 (to H.S.), 81702071 (to H.S.) from the National Natural Science Foundation of China, grant 20ZR1466900 and 22ZR1481800 (to J.Q.), from Shanghai Science and Technology Committee (STCSM).

## Author contributions

Z.D., T.C., and X.Q. designed and performed mouse experiments. Z.D. and H.S. designed and performed experiments with human samples. Z.D., Y.J., L.Y., H.C., and J.Q. performed bioinformatics analyses and organized bioinformatics data. J.T., Y.Z., and Y.M. helped with experiments. J.Q. and Z.D. wrote the manuscript. H.Z., L.L., and H.S. facilitated experimental design, manuscript writing, and proofreading. L.L. generated the *Setd2 flox/flox* mouse. J.Q., H.S., and L.L. conceived, designed, and supervised the project.

## Competing interests

The authors declare no competing interests.
