## [Peer Review File · Nature Communications]

Setd2 supports GATA3+ST2+ thymic-derived Tregs and suppresses intestinal inflammationReviewer #1 (Remarks to the Author):

In their manuscript, Ding et al. study the role of the histone H3K36 methyltransferase Setd2 for Foxp3+ regulatory T cells (Tregs). It is well-known that Treg development and functional specialization is under epigenetic control, yet the molecular mechanisms controlling these processes are only incompletely understood. In the present study, the authors mainly use Treg-specific conditional Setd2 knockout mice to study the impact of Setd2 deficiency for the development, homeostasis, and functional properties of Foxp3+ Tregs. They provide some evidence that Setd2 deficiency is dispensable for the thymic Treg development, but affects the Treg homeostasis and suppressive capacity in peripheral tissues, particularly the intestine. Furthermore, the author could demonstrate that Setd2 regulates several target genes by modulating the activity of promoters and intragenic enhancers. Viewed as a whole, the study is of interest. Yet, before this manuscript can be published the whole manuscript requires an intense language editing to improve readability and a number of important specific issues need to be addressed.

Specific concerns

- It is not obvious why the authors have focused their analyses solely on the intestine. ST2+ Tregs are found in many peripheral tissues, and previously published studies have already demonstrated that tissue-resident Tregs have a distinct epigenetic signature when compared to Tregs from secondary lymphoid organs (e.g. Delacher et al., Nat Immunol 2017, PMID 28783152). Thus, the authors should include additional tissues (e.g. skin, lung, liver, visceral adipose tissue) into their analyses to find out if Setd2 plays a particular role for intestinal Tregs or for tissue-resident Tregs in general (also see statement from discussion, lines 365-366).
- Figure 1A and S1A: Gating strategy and isotype controls are missing. 'Teffs' should be renamed into Tconv throughout the manuscript if referring to Foxp3- CD4+ T cells. Furthermore, the authors should comment on their finding that Tconv express higher Setd2 levels when compared to Tregs and that Tregs seem to contain subsets with high vs. intermediate Setd2 expression. Finally, scatterplots summarizing independently generate data should be added.
- Figure 1C+E: Data from large intestine do not fit to each other as no difference was observed for effector/memory CD4+ T cells, but a significant difference was observed for naïve CD4+ T cells. Please check.
- Figure 1I: The authors should not only depict data on the frequency of Foxp3+ cells among CD4SP thymocytes, but in addition should also assess the frequency of the two recently identified thymic Treg precursors, e.g. by using the staining strategy described in Owen et al. (Nat Immunol 2019, PMID 30643267).
- Figure 2A-C: In addition to the BM chimeric mice, they authors should also study female mice that are heterozygous for the Foxp3-Cre transgene. In these mice, due to the random X chromosome inactivation only half of the Tregs express Cre, which allows to assess the impact of Setd2 deficiency in 'non-manipulated' mice and to directly compare Setd2-proficient and -deficient Tregs in the same environment, excluding any blurring by distinct inflammatory conditions.
- Figure 2D+E: It is widely accepted that Tregs rely on 'exogenous' IL-2 for their homeostasis and survival. Which cells are providing IL-2 if highly purified Tregs were adoptively transferred into Rag2-/- recipient mice? The authors should provide the purity of the transferred cells and also show data on the frequency of Foxp3+ cells among T cells at the analysis time point to assess the stability of Foxp3 expression.
- Figure 3: The authors should use congenic markers to discriminate between naïve T cells and co-transferred Tregs, and investigate if the reduced frequency of Foxp3+ Tregs in Rag2-/- mice that were co-transferred with Setd2-deficient Tregs (see panel K) is due to impaired stability of Foxp3 expression or impaired homeostasis of Setd2-deficient Tregs?
- Figure 4 and lines 202, 203, 213-214, 217-218: The authors should better define the different signatures used in these analyses, e.g. by providing gene lists in a Supplementary Table.
- Figure 4C+D: The authors should also generate RNAseq data from splenic Tregs (KO vs. control) to directly compare the impact of Setd2-deficiency in Tregs from secondary

lymphoid organs vs. peripheral tissues.

- **Figure 4G-K:** This interesting finding should be expanded (e.g. comparison of different organs) and explained better in the corresponding text (lines 234-237). Absolute cell numbers are not always given and the Figures are a little bit confusing; the authors might want to focus on relevant subsets instead of showing data for all of them.
- **Figure 5 and S5:** The authors should consider showing the representative dot plots first (now panel D) and moving the interesting spleen data into the main Figure. The authors should also analyze the impact of Setd2-deficiency on the recently described precursors of tissue Tregs, which are peaking in the spleen around day 10 after birth (Delacher et al., Immunity 2020, PMID 31924477).
- **Figure 5E-I:** The authors observed interesting phenotypes after stimulating Tregs with IL-2, wherefore they should also analyze the IL-2R expression (e.g. CD122 and CD25) on KO and control Tregs.
- **Figure 7:** It is not obvious from the corresponding text and Figure legend why data sets from intestinal Tregs and splenic Tregs of IL-2c-treated mice were merged. All correlation analyses should be only done with data generated from the same cells (e.g. the identification of RNA Pol II regions). Overall, the findings from this Figures are rather confusing and leave the author with a number of unanswered questions (e.g. the level at which Setd2 contributes to the regulation of GATA3 expression in Tregs).

Minor concerns

- The abstract needs to be reorganized to better describe the current knowledge and the key findings from the study.
- Lines 79+85: Cytokines (TGFb, IL-2 and IL-35) are not environmental factors.
- Line 103: The authors need to explain the term 'conservative regulation'.
- Line 120-121: The sentence "...confirming that Setd2 is the dominant H3K36 methyltransferase mediating H3K36me3 in Tregs" does not make any sense since the authors have stated in lines 96-97 that "Setd2 is the only known mammalian histone H3 lysine 36 (H3K36) methyltransferase mediating H3K36me3".
- Line 124 and Figure 1C+D: The authors should better use 'effector/memory' instead of 'activated' to describe the phenotype of CD44+CD62Llow cells.
- Line 135: The term 'periphery Tregs' is not commonly used and the authors should better explain which Tregs they are referring to.
- Line 148: Rephrase sentence as 'fate conversion of Tregs to non-Tregs' could only be studied with the help of a fate reporter that is missing in the present study.
- Line 149: Replace 'ratio' by 'frequency'.
- Line 167: What is meant with 'extreme effort'?
- **Figure S3:** It is difficult to draw any conclusions from the suppression assay since the conditions of the assay (e.g. cell number or stimulus) need to be optimized to enable complete inhibition of naïve T cell proliferation if Tregs were added at a 1:1 ratio.
- Line 221-222: The authors should consider rephrasing since experimental evidence for this specific assumption is lacking and Setd2-deficiency could also simply affect the differentiation of Treg subsets.
- **Figure 4 and S4:** The authors might want to consider using the exact phenotypes instead of the terms 'DP' and 'DN' since they are frequently used for the description of thymocyte subsets.
- **Figure 5L:** It is not obvious why the authors have depicted a 'normalized ratio' for the frequency of GATA3+ST2+ cells among Helios+ Tregs.

Reviewer #2 (Remarks to the Author):

This paper reports the effect of a deletion of Setd2 in Tregs.

In figure 1, the authors report a statistically significant reduction of Tregs in LI and MLN, accompanied by an increase in activated CD4 and CD8 T cells.

As a "Treg defect" is the starting observation, it would be interesting to analyze the phenotype of Setd2- Tregs in more details. What is their naïve/memory status?

**How are expressed functional markers such as CD25, CTLA4, GITR...?
How is their suppressive activity in vitro.**

In figure 2, the authors report a functional disadvantage of Setd2- Tregs that they then link to a poor response to IL-2. Given the pleiotropic consequence of the Setd2- KO on the gene expression as analyzed in fig 4, it would be interesting to analyze if this poor response to IL-2 could be link to modifications of the IL-2 receptor activation pathway, for example by analysing IL-2-induced Stat5 phosphorylation.

Fig3 convincingly show that Setd2- Tregs are less efficient in controlling the induction of colitis by Teff.

In figure 4, the authors analyzed the transcriptome of Setd2- Tregs. It would have been useful to first show a global comparison between Setd2- Tregs and plain Tregs using for example a PCA, and then to show a volcano plot with p values for variations... This would have helped to more convincingly point to Gata3 and ST2 expression.

In figure 5, in F to I the representation is quite misleading: it is invalid to draw line between the values obtained from different mice.

In figure 6, the defect of Setd2- Tregs does not induce any pathology. Thus, how significant is this defect?

Reviewer #3 (Remarks to the Author):

This is an interesting study that addresses the role of the histone H3K36 methyltransferase Setd2 in regulatory T cells (Tregs). Deficiency for this enzyme in the Treg lineage results in an unexpectedly selective phenotype, that seems to especially affect a subset of thymic derived Tregs that express the transcription factor Gata3 and the cytokine receptor ST2 in the intestinal lamina propria and the local draining lymph nodes. The selectivity of this phenotype is unexpected, because the enzyme is expressed in all Treg cells.

On the whole, the analysis of the mice was done well. I have only minor technical comments, which I will list below. In my view, a weakness of the article is that it remains rather descriptive and that it is not all that clear what new insights we have obtained from these results. No clear rationale is provided why the role of SetD2 should be examined in Tregs and there is no clear explanation why this enzyme, that one would expect to have a rather general role in control of gene expression, would have such a very selective role in regulation of just a particular subtype of Tregs. I believe that at least some discussion of these questions and possible explanations would make the article more interesting. Some discussion is also necessary regarding the finding in Figure 2 that there is a more general survival defect in the Tregs upon adoptive transfer into immunodeficient mice. How does this result fit with the very selective phenotype during steady state in mice? Is survival in the adoptive transfer model also dependent on Gata3 and/or ST2?

Minor points

Figure 1: it would be good to provide absolute cell numbers in addition to the percentages shown here.

Figure 4: I am missing the point about the H3K36me3 levels show in Figure 4C. It seems to me that there is absolutely no correlation between these levels and expression of the genes that are dependent on SetD2. This needs clarification.

Figure 4S: the text claims that there is no increase in IL17+ Tregs. However, it looks to

me like there is a clear trend towards higher IL17+ Tregs. While the difference in these data does not reach statistical significance, that does not prove that there is no increase.

Figure 6: injection of IL33 elicits a greater increase in ST2+ Th2 cells in the knock out mice than in the wild type. I wonder what the right interpretation of this result is. One possibility is that the SetD2-dependent Treg population inhibits the generation of ST2+ Th2 cells in steady state, such that there are more cells in the knock out mice with the capacity to respond to his cytokine and proliferate. A second possible explanation is that the SetD2-dependent Treg population directly regulates the response by conventional T cells to IL33. Finally, it is possible that Set2D-dependent Tregs transdifferentiate into ST2+ Th2 cells. There are sophisticated lineage tracing approaches that can address this latter possibility. However, I realise that doing those would be a large time investment. I would be satisfied with a mixed chimera experiment, which will at least address whether the effect is cell intrinsic (which would argue in favour of transdifferentiation) or not (which would support the other two possibilities).

Figure 6M/N: I am not very convinced about the elevated number of Th2 cells in the cancer samples. This is mostly because the Gaat3 staining is very weak. I think these data need to be strengthened or taken out of the manuscript. One way to strengthen the results would be to restimulate the cells and measure Th2 cytokines. Alternatively, there might already be single cell RNAseq data publicly available that can be used to address this point.

Finally, there is no information about the statistical tests used in any of the figures. This seems especially relevant for Figure 2, 3 5, 6, where sometimes multiple comparisons were made which should be analyzed with something like ANOVA.

Response to Reviewers' comments:

We wish to thank all the Reviewers for their constructive comments on our manuscript, "Setd2 supports GATA3⁺ST2⁺ thymic-derived Tregs and suppresses intestinal inflammation" (NCOMMS-21-47229). To address all Reviewers' questions, additional experiments have been performed, resulting in 10 additional figure panels in the main figures (Fig. 1b, 1j-1l, 1o, 2f, 2g, 4l, 4m and 6p), 23 panels in the supplementary figures (Supplementary Fig. 1a, 1g, 1h, 2b-2d, 3b-3e, 4b-4f, 4h, 5k-5q), 1 supplementary table (Supplementary Table 1) and 6 figures for Reviewers (Review Fig. 1-6 in the response letter). Data in Fig. 1d-1i, Supplementary Fig. 1b and 3a, and Supplementary Table 3 have been updated.

Major changes upon formatting our manuscript following *Nature Communications* instructions are listed as following. 1) Subheadings were shortened to less than 60 characters without changing the scientific meanings. 2) Exact P values calculated from statistical analyses for null hypothesis testing in all our Figures were provided. 3) Numbers of references were limited to less than 70. In addition, as we prepared the "Source Data" file containing raw data for generating bar charts and scatter plots, we noticed some mistakes when preparing for the "Source Data" file. Fig. 3b (the "n" of 2 groups were switched), 3i (one dot outside of the Y axis was included), Fig. S2a (some misplaced data were corrected) and S4g (397 was changed to 395) have been corrected and updated. These changes didn't affect our original statistical and scientific conclusions. We apologize for these mistakes in the previous version of manuscript.

Furthermore, English language editing recommended to be performed by Reviewer 1 has been performed throughout the manuscript to improve readability. Point-by-point responses were provided below marked with "author response" to answer every comment from the Reviewers. Changes that have been made in the text of the revised manuscript, including the English language editing recommended to be performed by Reviewer 1, are highlighted in yellow except for corrections of grammatical mistakes or typos. We hope that our revised manuscript is now appropriate for publication by *Nature Communications*.

Reviewer #1 (Remarks to the Author):

In their manuscript, Ding et al. study the role of the histone H3K36 methyltransferase Setd2 for Foxp3⁺ regulatory T cells (Tregs). It is well-known that Treg development and functional specialization is under epigenetic control, yet the molecular mechanisms controlling these processes are only incompletely understood. In the present study, the authors mainly use Treg-specific conditional Setd2 knockout mice to study the impact of Setd2 deficiency for the development, homeostasis, and functional properties of Foxp3⁺ Tregs. They provide some evidence that Setd2 deficiency is dispensable for the thymic Treg development, but affects the Treg homeostasis and suppressive capacity in peripheral tissues, particularly the intestine. Furthermore, the author could demonstrate that Setd2 regulates several target genes by modulating the activity of promoters and intragenic enhancers. Viewed as a whole, the study is of interest. Yet, before this manuscript can be published the whole manuscript requires an intense

language editing to improve readability and a number of important specific issues need to be addressed.

Author response: We would like to thank Reviewer 1 for the comments. The English language of the manuscript has been edited by the Springer Nature language editing service.

Specific concerns

1. It is not obvious why the authors have focused their analyses solely on the intestine. ST2⁺ Tregs are found in many peripheral tissues, and previously published studies have already demonstrated that tissue-resident Tregs have a distinct epigenetic signature when compared to Tregs from secondary lymphoid organs (e.g. Delacher et al., Nat Immunol 2017, PMID 28783152). Thus, the authors should include additional tissues (e.g. skin, lung, liver, visceral adipose tissue) into their analyses to find out if Setd2 plays a particular role for intestinal Tregs or for tissue-resident Tregs in general (also see statement from discussion, lines 365-366).

Author response: We would like to thank Reviewer 1 for raising this important point. The intestine harbors food antigens and trillions of microbiota that induces highly activated Tconvs, which need to be controlled by Tregs to prevent autoimmunity even under steady state. We initially observed a reduction in the percentage of Tregs in the GALTs but not in SLOs or other analyzed tissues. The above points have been added in the Discussion section (line 464-467). Intriguingly, Setd2 expression in epithelial cells has been reported to be decreased in experimental colitis¹. Mutation and deficiency of Setd2 is found to be associated with susceptibility to colorectal cancer²(line 495-497). These observations/reports intrigued us to focus on studying the regulation of Tregs by Setd2.

We agree with Reviewer 1 that it is important to examine Tregs in other tissues of *Foxp3^{Cre}Setd2^{ff}* mice. We found that there was no difference in percentages or absolute numbers of Tregs from the fat (epididymal adipose tissue), liver or lung of *Foxp3^{Cre}Setd2^{ff}* mice compared with littermate controls (Fig. 1o and Supplementary Fig. 1h). To conjunctively address question 10 from Reviewer 1, we also examined the expression of ST2 and GATA3 in Tregs from the fat, liver and lung. Interestingly, percentages of Helios⁺ST2⁺ Tregs (ST2⁺ tTregs) among total Tregs were reduced in the lung, fat and liver of *Foxp3^{Cre}Setd2^{ff}* mice (Fig. 4l). Numbers of ST2⁺ tTregs were decreased in the lung of *Foxp3^{Cre}Setd2^{ff}* mice (Fig. 4l). However, there were no difference in percentages or numbers of the Helios⁺GATA3⁺ Tregs in the lung, fat and liver of *Foxp3^{Cre}Setd2^{ff}* mice compared with controls (Fig. 4m). Importantly, we found no reduction in percentages of GATA3⁺ST2⁺ cells among total ST2⁺ tTregs, reflecting a reciprocal relationship of ST2 and GATA3 in tTregs, in the lung, fat and liver of *Foxp3^{Cre}Setd2^{ff}* mice (Review Fig. 1), whereas we showed that such percentages were reduced in the large intestine of *Foxp3^{Cre}Setd2^{ff}* mice in the previous version of manuscript (Supplementary Fig. 5g). Collectively, these data indicate that Setd2 broadly supports the homeostasis of tissue ST2⁺ tTregs, but more prominently supports the Treg homeostasis and the reciprocal relationship of GATA3 and ST2 in tTregs from the

intestine. We speculate that this may be due to higher level of IL-33 in the intestine compared to the lung, fat or liver, but this hypothesis remains further investigation. We discussed these points in the Discussion section : “Interestingly, while *Setd2* supports $ST2^+$ tTregs in both the intestine and other analyzed peripheral tissues,This may suggest that IL-33-supported $ST2^+$ GATA3⁺ tTregs homeostasis dependent on *Setd2* is more prominent in the intestine than in other organs under steady state.”(line467-474)

Review Fig. 1. Expression of GATA3⁺ST2⁺ cells in ST2⁺tTregs in different tissues. Mononuclear cells were isolated from fat (epididymal adipose tissue), lung and liver of littermate *Foxp3^{Cre}Setd2^{fl/fl}* mice or *Foxp3^{Cre}Setd2^{fl/f}* male mice. Expression of GATA3 and ST2 gated on Helios⁺ Tregs (CD3⁺CD4⁺Foxp3⁺ cells) was analyzed by flow cytometry. Percentages of GATA3⁺ST2⁺ cells gated on ST2⁺ tTregs (CD3⁺CD4⁺Foxp3⁺Helios⁺ST2⁺ cells) are shown. Representative of 2 independent experiments. Data are means ± SEM.

Fig. 1o, Supplementary Fig. 1h and Fig. 4l-4m have been added to the revised manuscript. The findings have been described in the Results section. (line 153-155, 299-304)

2. Figure 1A and S1A: Gating strategy and isotype controls are missing. ‘Teffs’ should be renamed into Tconv throughout the manuscript if referring to Foxp3- CD4+ T cells. Furthermore, the authors should comment on their finding that Tconv express higher Setd2 levels when compared to Tregs and that Tregs seem to contain subsets with high vs. intermediate Setd2 expression. Finally, scatterplots summarizing independently generate data should be added.

Author response: We would like to thank Reviewer 1 for raising the above questions. We have changed the nomenclature of “Teffs” referring to the “Foxp3-CD4⁺ T cells” to “conventional T cells (Tconv)” throughout manuscript.

Gating strategy for Fig. 1a and secondary antibody only (anti-rabbit APC) as control was provided as Supplementary Fig. 1a and 1b respectively. As the negative staining control in Supplementary Fig. 1b confirmed the specificity of Setd2 staining in Tregs, we didn’t include a negative staining control in Fig. 1a. Furthermore, Setd2 staining on Tregs from *Foxp3^{Cre}Setd2^{fl/f}* mice served as a genetic negative control for specific Setd2 expression by

Tregs and Tconvs (Fig. 1a).

Upon calculating the mean fluorescence intensity (MFI) of Setd2 expressed by splenic CD4⁺ Tconvs and Tregs of *Foxp3^{Cre}Setd2^{fl/fl}* and *Foxp3^{Cre}Setd2^{fl/+}* mice, we found that Setd2 expression in Tconv was indeed slightly but significantly higher than that in Tregs from *Foxp3^{Cre}Setd2^{fl/+}* mice (Fig. 1b). We think that this was possibly due to one deficient allele of *Setd2* in Tregs of *Foxp3^{Cre-YFP}Setd2^{fl/+}* mice. Consistent with this notion, we found that Setd2 expression in Tconvs and Tregs from wild-type (WT) mouse was comparable (Review Fig. 2a and 2b). Notably, no difference in percentages of Tregs in the intestine or GATA3⁺ expression among intestinal Helios⁺Tregs (tTregs) was observed between *Foxp3^{Cre-YFP}Setd2^{fl/+}* mice and *Foxp3^{Cre-YFP}Setd2^{+/+}* mice (Review Fig. 2c and 2d). As the percentages of intestinal Tregs and GATA3⁺ expression in tTregs were two of the key deficiencies observed in *Foxp3^{Cre-YFP}Setd2^{fl/fl}* mice compared with *Foxp3^{Cre-YFP}Setd2^{fl/+}* mice (Fig. 1n, 4i-4k) and were not perturbed by heterozygous deficiency of Setd2 in Tregs (Review Fig. 2c and 2d), the data support that *Foxp3^{Cre-YFP}Setd2^{fl/+}* mice could be served as an efficient control for studying the function of Setd2 in Tregs.

Review Fig. 2. Analysis of Setd2 expression in WT mice and Tregs in *Foxp3^{Cre-YFP}Setd2^{fl/+}* mice

(a and b) Splenocytes were isolated from WT mice. Expression of Setd2 gated on Tregs (CD3⁺CD4⁺Foxp3⁺ cells) or CD4⁺ Tconvs (CD3⁺CD4⁺Foxp3⁻ cells) was analyzed by flow cytometry. (a) Secondary antibody only on splenic Treg from WT mice was used as a negative staining control. (b) Mean fluorescence intensity (MFI) of Setd2 was shown. (c and d) Large intestinal lamina propria lymphocytes were isolated from *Foxp3^{Cre}Setd2^{+/+}* mice or *Foxp3^{Cre}Setd2^{fl/+}* mice. (c) Percentage of Tregs (CD3⁺CD4⁺Foxp3⁺) in CD4⁺ T (CD3⁺CD4⁺) cells was analyzed by flow cytometry. (d) Expression of GATA3 gated on Helios⁺ Tregs (CD3⁺CD4⁺Foxp3⁺Helios⁺) was analyzed by flow cytometry. (a-d) Representative of 2 independent experiments. (b-d) Data are means ± SEM.

Fig. 1b and Supplementary Fig. 1a have been added to the manuscript. Supplementary Fig. 1b has been updated. In the results section, we have added “We noticed a slight but significant increase in Setd2 expression by Tconvs compared with Tregs in *Foxp3^{Cre-YFP}Setd2^{fl/+}* mice (Fig. 1a and 1b). This was possibly due to one deficient allele of Setd2 in Tregs of *Foxp3^{Cre-YFP}Setd2^{fl/+}* mice.” (line 122-125)

3. Figure 1C+E: Data from large intestine do not fit to each other as no difference was observed for effector/memory CD4⁺ T cells, but a significant difference was observed for naïve CD4⁺ T cells. Please check.

Author response: We appreciate Reviewer 1 for raising this important point. We revisited the data and noticed that we missed some CD44⁺CD62L^{low} events when analyzing the intestinal flow cytometry staining data in Fig. 1d and 1f (Figure 1C+E in the previous version of manuscript). We originally used the same gating strategy for CD44 and CD62L for spleen, inguinal lymph nodes (ILN), mesenteric lymph nodes (MLN) and large intestine (LI). However, the LI contains significantly higher percentages of CD44⁺CD62L^{low} cells than the secondary lymphoid organs and some events close to the Y axis were not included in the analysis. We have reanalyzed the LI flow cytometry data and recovered these lost events. We are very sorry for this mistake. We have ensured that this problem is not present in flow cytometry analysis of CD44 and CD62L expression in the secondary lymphoid organs (SP, ILN and MLN).

Upon correction of the data, we found that the percentage of effector/effector memory CD4⁺ T cells among total CD4⁺ T cells was significantly increased in the LI of *Foxp3^{Cre}Setd2^{ff}* mice compared with controls (Fig. 1d). The corrected analyses further strengthened our original conclusion that “Setd2-deficient Tregs have reduced capability to inhibit T cell activation in the secondary lymphoid organs (SLOs) and the intestine”. Other results and conclusions were not affected after the data correction, except that we observed significantly decreased percentage (previously only a trend was observed) of central memory cells in CD4⁺ T cells in the LI of *Foxp3^{Cre}Setd2^{ff}* mice (Fig. 1h). This reduced level of central memory CD4⁺ T (Tcm) cells was possibly seconded to increased effector/effector memory cells CD4⁺ T cells (Tem) in the LI of *Foxp3^{Cre}Setd2^{ff}* mice. This may also imply that there is a conversion of Tcm to Tem cells possibly triggered by pro-inflammatory environment in the LI of *Foxp3^{Cre}Setd2^{ff}* mice³. But these hypotheses remain further investigation.

As a note, Reviewer 2 asked us to analyze the "naive/memory" status of Tregs (See response to question 1 from Reviewer 2) (Fig. 1j-1l). We observed reduced percentages of naive cells without an increase in the percentages of effector/effector memory cells in Tregs from the LI, MLN and spleen of *Foxp3^{Cre}Setd2^{ff}* mice (There was a slight increase on average in the percentages of effector/effector memory cells in Tregs from the MLN and spleen of *Foxp3^{Cre}Setd2^{ff}* mice). This may be because there was only a mild reduction in the percentage of naive Tregs, which only contributed to a small proportion among total Tregs.

LI data in Fig. 1d-1i have been updated in the revised manuscript. The correction of our data doesn't affect our original description on the change of effector/effector memory CD4⁺ T cells (line 130-134). The following description has been added: “*The percentages of central memory (CD44⁺CD62L^{high}) CD4⁺ Tconvs but not CD4⁻ T cells were reduced in the large intestine (Fig. 1h and 1i)*”. (line 135-137)

4. Figure 1I: The authors should not only depict data on the frequency of Foxp3⁺ cells among CD4SP thymocytes, but in addition should also assess the frequency of the two recently identified thymic Treg precursors, e.g. by using the staining strategy described in Owen et al. (Nat Immunol 2019, PMID 30643267).

Author response: We would like to thank Reviewer 1 for the advice. We have performed the experiments as the Reviewer 1 suggested. We found that there was no difference in percentages of Foxp3⁻CD25⁺, Foxp3^{low}CD25⁻ and CD25⁺Foxp3⁺ cells among thymic CD4⁺ single positive cells in *Foxp3^{Cre}Setd2^{ff}* mice compared with controls (Fig. 1m). The data indicate that Setd2 deficiency does not affect Treg development in the thymus. Fig. 1m has been updated. The findings have been added to the Results section “*The development of Tregs in the thymus was normal in Foxp3^{Cre-YFP}Setd2^{ff} mice, as indicated by the similar percentages of thymic Tregs and two Treg precursor subsets (Foxp3⁻CD25⁺ and Foxp3^{low}CD25⁻ cells) among CD4⁺ single-positive T cells (Fig. 1m)*”(line 145-148).

5. Figure 2A-C: In addition to the BM chimeric mice, they authors should also study female mice that are heterozygous for the Foxp3-Cre transgene. In these mice, due to the random X chromosome inactivation only half of the Tregs express Cre, which allows to assess the impact of Setd2 deficiency in ‘non-manipulated’ mice and to directly compare Setd2-proficient and -deficient Tregs in the same environment, excluding any blurring by distinct inflammatory conditions.

Author response: We would like to thank Reviewer 1’s suggestion. We analyzed YFP⁺ and YFP⁻ Tregs indicating Setd2-deficient and Setd2-sufficient Tregs in the intestine of female *Foxp3^{Cre/+}Setd2^{ff}* mice. The percentage of YFP⁺ cells was significantly lower than percentage of YFP⁻ cells among Foxp3⁺ Tregs (Supplementary Fig. 2b and 2c). The data indicate that Setd2 is required for the homeostatic maintenance of Tregs in the intestine. Supplementary Fig. 2b and 2c have been added. These findings have been added to the Results section (line171-175). To distinguish the genotype of *Foxp3^{Cre}Setd2^{ff}* and *Foxp3^{Cre/+}Setd2^{ff}* in this study, we have added the following description in the “Mouse” of the Methods section: “Mice labeled with *Foxp3^{Cre-YFP}(Foxp3^{Cre})* are *Foxp3^{Cre-YFP/Y}* for males and *Foxp3^{Cre-YFP/Cre-YFP}* for females. Mice labeled with the genotype *Foxp3^{Cre-YFP/+} (Foxp3^{Cre/+})* are females.” (line687-689)

6. Figure 2D+E: It is widely accepted that Tregs rely on ‘exogenous’ IL-2 for their homeostasis and survival. Which cells are providing IL-2 if highly purified Tregs were adoptively transferred into Rag2^{-/-} recipient mice? The authors should provide the purity of the transferred cells and also show data on the frequency of Foxp3⁺ cells among T cells at the analysis time point to assess the stability of Foxp3 expression.

Author response: We would like to thank Reviewer 1 for raising this question. Previous studies have consistently shown that purified Tregs could undergo homeostatic expansion in *Rag2^{-/-}* hosts^{4, 5}. Group 3 innate lymphoid cells (ILC3s) have been found to be one of the sources for IL-2 in the intestine in immunodeficient mice⁶. Indeed, possibly due to limited

amount of IL-2 compared with immunocompetent mice, about 50% of Tregs lost Foxp3 expression (this was consistent with published findings^{4,5}), indicating conversion of Tregs to Tconvs. We found that Setd2-deficient Tregs had increased fate conversion compared to control Tregs, as was suggested by the reduced percentage of Foxp3⁺ cells among CD4⁺ T cells (Fig. 2f). We have confirmed that the sorting purity for both Setd2-deficient and control Tregs reached more than 95% before transfer (Supplementary Fig. 2d). The data suggest that Setd2 prevents fate conversion of Tregs during homeostatic expansion. Importantly, absolute numbers of total CD4⁺ T cells including Tregs and fate-converted Tregs (exTregs) in *Rag2*^{-/-} transferred with Setd2-deficient Tregs were lower than that in *Rag2*^{-/-} transferred with control Tregs in the intestine (Fig. 2g). This indicates that Setd2-deficient donor cells, with Tregs and exTregs added up, have defective maintenance in *Rag2*^{-/-} mice. This led us to think that there might be other mechanisms in addition to preventing fate conversion that mediate Setd2-supported Treg maintenance. As we have demonstrated Setd2 sustains Treg survival, we think that these mechanisms may collectively contribute to Treg homeostasis supported by Setd2.

Fig. 2f and 2g, Supplementary Fig. 2d have been added to the revised manuscript. The above findings have been incorporated to the Results section. (line 178-189).

7. Figure 3: The authors should use congenic markers to discriminate between naïve T cells and co-transferred Tregs, and investigate if the reduced frequency of Foxp3⁺ Tregs in *Rag2*^{-/-} mice that were co-transferred with Setd2-deficient Tregs (see panel K) is due to impaired stability of Foxp3 expression or impaired homeostasis of Setd2-deficient Tregs?

Author response: We thank Reviewer 1 for raising this point. We performed experiments as Reviewer 1 suggested. Thy1.1⁺CD4⁺CD45RB^{high} T cells were transferred with Thy1.2 Setd2-deficient Tregs (KOTregs) or control Tregs (CtrlTregs) to *Rag2*^{-/-} mice (Supplementary Fig. 3b), which were sacrificed for analysis when the mice developed colitis. We found that the percentages of Tregs gated on Thy1.2 origin was significantly lower from *Rag2*^{-/-} mice receiving KOTregs, suggesting increased fate conversion of Tregs to Tconvs (exTregs) in the absence of Setd2 (Supplementary Fig. 3c). This data indicates that Setd2 is important for preventing fate conversion of Tregs during CD45RB^{high}T cell-induced colitis. However, the fate converted Tregs (exTregs) contributed marginally (less than 3%) to the pathogenic T cell pool and this proportion was much lower in the “T+KOTreg” group (Supplementary Fig. 3d). Furthermore, we observed a trend towards a reduction in absolute numbers of Thy1.2⁺ T cells including Tregs and exTregs from the large intestine of *Rag2*^{-/-} mice receiving KOTregs (Supplementary Fig. 3e). The data raise a possibility that other mechanisms in addition to preventing fate conversion of Tregs may contribute to Setd2-supported Treg sustenance in CD45RB^{high}T cell-induced colitis.

Supplementary Fig. 3c-e have been added. The above findings have been included in the Results section. (Line 226-239)

8. Figure 4 and lines 202, 203, 213-214, 217-218: The authors should better define the different signatures used in these analyses, e.g. by providing gene lists in a Supplementary Table.

Author response: We would like to thank Reviewer 1 for this suggestion. Lines 202, 203, 213-214, 217-218 from previous version of manuscript are lines 250, 252, 271 and 276 in the current version of manuscript. To make the nomenclature uniform, we have changed the description of “gut signature genes” to “colonic signature genes” in our manuscript. Splenic signature genes, colonic signature genes, ROR γ ⁺Treg signature genes and ST2⁺Treg signature genes were provided as independent sheets in Supplementary Table 1. The 57 increased and 53 decreased genes in Setd2-deficient intestinal Tregs that are colonic Treg signature genes were provided as an independent sheet in Supplementary Table 3.

9. Figure 4C+D: The authors should also generate RNAseq data from splenic Tregs (KO vs. control) to directly compare the impact of Setd2-deficiency in Tregs from secondary lymphoid organs vs. peripheral tissues.

Author response: We would like to thank Reviewer 1 for this suggestion. We performed RNA-seq analysis on purified splenic Tregs from *Foxp3^{Cre}Setd2^{fl/fl}* and *Foxp3^{Cre}Setd2^{fl/+}* mice. Principal component analysis (PCA) revealed that PC1 distinguished splenic Tregs from intestinal Treg cells, and PC2 separated Setd2-deficient from Setd2-sufficient Tregs (Supplementary Fig. 4b). Splenic and intestinal Tregs shared a series of upregulated (249) and downregulated (172) genes upon Setd2 ablation (Supplementary Fig. 4c and Supplementary Table 3). This includes the decreased expression of *Il1rl1*, *Myc* and increased expression of *Cxcr3*, *Icos* and *Gpr15*, which have been reported to mediate/regulate the migration or function of Tregs (Supplementary Fig. 4d).

Supplementary Fig. 4b-4d were added to the revised manuscript. Supplementary Table 3 was updated. The above findings have been included in the Result section (line 261-267).

We also analyzed the expression of genes listed in Fig. 4c in splenic Tregs (Review Fig. 3). Similar to their changed expression in intestinal Tregs upon ablation of Setd2, the expression of *Il1rl1*, *Itgae* and *Zfp3611* were decreased, and the expression of *Mmp9*, *Ccr9*, *Tiparp*, *Lztf11* and *Rorc* (with p value close to 0.05) were increased (Review Fig. 3). Opposite to their changed expression in intestinal Tregs upon ablation of Setd2, the expression of *Dgat2*, *Tnfrsf9*, *Ccr8* were increased (Review Fig. 3). The rest of the genes showed no significant difference in expression between splenic Setd2-deficient Tregs and control Tregs (Review Fig. 3). It is interesting that *Il1rl1* (encoding ST2) was consistently appeared to be a hallmark target of Setd2 demonstrated by both splenic and intestinal Tregs. It may not be surprising that some genes were differentially regulated in splenic and intestinal Tregs by Setd2, as the genes listed in Fig. 4c are colonic signature genes and they could be expressed at a relatively low level by splenic Tregs. Review Fig. R3 was not incorporated in the manuscript because disease-relevant phenotypes could be better interpreted by the transcriptional change of intestinal Tregs regulated by Setd2.

Review Fig 3. The expression of lists of colonic signature genes by splenic Tregs. Splenic Tregs (CD3⁺CD4⁺Foxp3-YFP⁺) sorted from *Foxp3*^{Cre-YFP}*Setd2*^{f/+} (Ctrl) or *Foxp3*^{Cre-YFP}*Setd2*^{ff} (KO) mice were subjected to RNA-seq analysis. The protocol for RNA-seq analysis was provided in the Methods section of the manuscript. Heatmap shows list of colonic signature genes, which are differentially expressed by intestinal *Setd2*-deficient Tregs in Fig. 4c, generated based on the Z score of FPKM value of splenic Treg RNA-seq analysis. Genes colored in red are differentially expressed genes (P<0.05).

10. Figure 4G-K: This interesting finding should be expanded (e.g. comparison of different organs) and explained better in the corresponding text (lines 234-237). Absolute cell numbers are not always given and the Figures are a little bit confusing; the authors might want to focus on relevant subsets instead of showing data for all of them.

Author response: We appreciate Reviewer 1's suggestion. We analyzed the RORγ⁺Treg population because they were identified as an increased signature by RNA-seq (Fig. 4c), which we feel it is necessary to be verified at the protein level. We have analyzed ST2⁺ and GATA3⁺ cells from both Helios⁺ and Helios⁻ cells because we found *Setd2* mainly affect ST2⁺ tTregs and GATA3⁺ tTregs rather than Helios⁻ST2⁺ and Helios⁻GATA3⁺ Tregs (Fig. 4g-4k and Supplementary Fig. 4q). Absolute numbers for these populations have been shown in the previous version of manuscript (now as Fig. 4k, Supplementary Fig. 4j and 4q in current version of manuscript). We feel that it is helpful to demonstrate progressively of how the key target Treg subsets affected by *Setd2* were identified.

We agree with Reviewer 1 that it is important to expand the finding by examination of other tissue Tregs. Reduced percentages of Helios⁺ST2⁺ Tregs in the liver, lung and visceral adipose tissue, and reduced numbers of Helios⁺ST2⁺ Tregs in the lung were observed in *Foxp3*^{Cre-YFP}*Setd2*^{ff} mice compared with *Foxp3*^{Cre-YFP}*Setd2*^{f/+} mice (Fig. 4l). Percentages or

absolute numbers of Helios⁺GATA3⁺ Tregs from the above analyzed tissues showed no difference in *Foxp3^{Cre-YFP}Setd2^{ff}* mice (Fig. 4m). The data suggest that Setd2 broadly supports the maintenance of tissue ST2⁺ tTregs.

Fig. 4l and 4m have been added to the revised manuscript and the findings have been incorporated into the Results section. (line299-304) The specific regulation of GATA3⁺ tTregs by Setd2 in the intestine but not in other analyzed tissues was explained in the Discussion section (line467-474). Please also see author response to Question 1 from Reviewer 1.

11. Figure 5 and S5: The authors should consider showing the representative dot plots first (now panel D) and moving the interesting spleen data into the main Figure. The authors should also analyze the impact of Setd2-deficiency on the recently described precursors of tissue Tregs, which are peaking in the spleen around day 10 after birth (Delacher et al., Immunity 2020, PMID 31924477).

Author response: We would like to thank Reviewer 1 for the suggestions. We have placed the representative dot plots before the figure panels of statistical analyses and have moved the spleen data into the main figure (Fig. 5a-5h). Previous data suggest that tissue Tregs are derived from Foxp3⁺Nfil3⁺KLRG1⁻ and Foxp3⁺ Nfil3⁺KLRG1⁺ progenitors in the spleen⁷. The research has also demonstrated PD-1 serves as a surrogate marker for Nfil3. We observed no difference in the percentages in PD-1⁻KLRG1⁻ cells, or PD-1⁺KLRG1⁻/PD-1⁺KLRG1⁺ tissue Treg progenitors among spleen Tregs from *Foxp3^{Cre-YFP}Setd2^{ff}* compared with control mice (Supplementary Fig. 5k and 5l). This suggests that Setd2 is unlikely to regulate development or maintenance of tissue Treg progenitors. ST2 were most highly expressed by PD-1⁺KLRG1⁺ Tregs, in which ST2 was reduced in the absence of Setd2 (Supplementary Fig. 5m). The data indicate that Setd2 supports ST2 expression in splenic tissue Treg progenitors.

Supplementary Fig. 5k-5m have been added. The above findings have been incorporated in the Results section. (line 340-348)

12. Figure 5E-I: The authors observed interesting phenotypes after stimulating Tregs with IL-2, wherefore they should also analyze the IL-2R expression (e.g. CD122 and CD25) on KO and control Tregs.

Author response: We would like to thank Reviewer 1 for raising this question. CD122 and CD25 expression was analyzed in KO and control Tregs. We observed slightly but significantly reduced CD25 (IL-2R α) expression in splenic and intestinal Tregs of *Foxp3^{Cre-YFP}Setd2^{ff}* mice (Supplementary Fig. 5n and 5o), whereas the expression of CD122 (IL-2R β) in splenic Setd2-deficient Tregs was increased (Supplementary Fig. 5p and 5q). Since we have shown in the previous version of the manuscript that the level of phosphorylated STAT5 was comparable in tTregs from the spleen or the intestine of *Foxp3^{Cre-YFP}Setd2^{ff}* mice and control mice (Supplementary Fig. 5r-5s), we think that the direct signal downstream of IL-2 is unlikely to be breached by deletion of Setd2 in Tregs.

Supplementary Fig. 5n-5q have been added. The above findings have been incorporated in the Results section.(line350-353)

13. Figure 7: It is not obvious from the corresponding text and Figure legend why data sets from intestinal Tregs and splenic Tregs of IL-2c-treated mice were merged. All correlation analyses should be only done with data generated from the same cells (e.g. the identification of RNA Pol II regions). Overall, the findings from this Figures are rather confusing and leave the author with a number of unanswered questions (e.g. the level at which Setd2 contributes to the regulation of GATA3 expression in Tregs).

Author response: We would like to thank Reviewer 1 for raising this question. Our data indicate that GATA3⁺ ST2⁺ tTregs are the hallmark targets affected by Setd2 in Tregs. When studying the molecular mechanisms, we used Tregs enriched for Helios⁺ Tregs (tTregs) for analysis including the 2-week-old intestinal Tregs (Supplementary Fig. 5e) and splenic Tregs from IL-2 treated mice (Supplementary Fig. 6b). We have confirmed that these cells have key phenotypes including decreased ST2 expression at both the mRNA and protein level (Fig. 7h and Supplementary Fig. 6d), decreased GATA3 expression at the protein level (Supplementary Fig. 6c) and defective homeostatic expansion (Fig. 2h and 2i) in the absence of Setd2, which have been observed in adult Setd2-deficient intestinal Tregs. Correlation analysis revealed that the transcriptome of adult Setd2-deficient intestinal Tregs could be largely interpreted by the landscape of H3K27ac modifications and Pol II binding analyzed from Helios⁺ Treg enriched 2-week-old intestinal Tregs and IL-2-treated splenic Tregs respectively (Fig. 7a-7c and 7f-7g). These data indicate that Setd2 (although not entirely) regulates most target genes in similar manners in adult intestinal Tregs, 2-week-old intestinal Tregs and IL-2-treated splenic Tregs.

To further address Reviewer 1's concern, we performed RNA-seq analysis on splenic Tregs purified from IL-2c (IL-2- α -IL-2 complex)-treated *Foxp3^{Cre-YFP}Setd2^{ff}* mice and control group (Review Fig. 4a-4e). We found that fold changes of genes differentially expressed by Setd2-deficient Tregs in the intestine and are positively correlated with fold change of genes differentially expressed by Setd2-deficient Tregs in the IL-2c-treated spleen (Review Fig. 4a). Gene set enrichment analysis (GESA) revealed that top 500 decreased genes in intestinal KOTregs were enriched to be expressed in IL-2c-treated splenic CtrlTregs (Review Fig. 4b). Likewise, top 500 increased genes in intestinal KOTregs were enriched to be expressed in IL-2c-treated splenic KOTregs (Review Fig. 4c). The data further validate that majority of genes (73.6% calculated by 34.7%+38.9%) are regulated by Setd2 in similar manners in both adult intestinal Tregs and IL-2c-treated splenic Tregs (Review Fig. 4a). These data serve as a rationale for using IL-2c-treated splenic Tregs as a surrogate for adult intestinal Tregs to perform RNA Pol II CUT&Tag analysis

Next, we performed correlation analysis on fold change of mRNA expression and Pol II binding using the same kinds of cells (IL-2c-treated spleen Tregs) (Review Fig. 4d-4e). We confirmed that genes with increased expression in Setd2-deficient Tregs have increased Pol II binding at both the promoter (Review Fig. 4d) and gene bodies (Review Fig. 4e), and *vice*

versa (Review Fig. 4d-4e), as was suggested by correlation coefficient and p value. These data support our original conclusion that “Setd2 could regulate gene expression through modulating gene transcription at both the initiation and elongation stages”.

Our data indicate that the regulation of GATA3 expression in Tregs by Setd2 is less likely to occur at the mRNA level (Fig. 7e and 7i). To test if Setd2 regulates the protein stability of GATA3, we treated splenocytes of *Foxp3^{Cre-YFP}Setd2^{ff}* mice and controls *in vitro* with MG132 preventing proteasome-mediated protein degradation and with chloroquine inhibiting lysozyme-mediated protein degradation (Review Fig. 4f). We found that treatment of MG132 or chloroquine, or the combination of the two compounds failed to mitigate the difference of GATA3 expression between Setd2-deficient Tregs and control Tregs (Review Fig. 4f). MG132 with and without chloroquine, but not chloroquine alone, prevented GATA3 degradation in both Setd2-deficient Tregs and control Tregs at a similar rate (Review Fig. 4f). Furthermore, tracking GATA3 expression after treatment of cycloheximide, which can stop protein synthesis, revealed a similar rate of GATA3 degradation in Tregs from *Foxp3^{Cre-YFP}Setd2^{ff}* mice compared with controls (Review Fig. 4g). The above data suggest that Setd2 is less likely to sustain GATA3 expression in Tregs by preventing proteasome or lysozyme-mediated protein degradation. We suspect that GATA3 may be regulated by Setd2 at the translational level. Such a possibility has been discussed in the discussion section (line 522-527). The following description has been deleted from the Discussion section: “*Further, Setd2 has been indicated to decrease protein stability by mediating protein ubiquitination and proteasome degradation*”.

Review Fig. 4 Bioinformatics analysis on IL-2-treated splenic Tregs and the effect of Setd2 on the stability of GATA3

(a-e) SP (spleen) Tregs sorted from IL-2c-treated *Foxp3^{Cre-YFP}Setd2^{fl/+}* (Ctrl) or *Foxp3^{Cre-YFP}Setd2^{fl/ff}* (KO) mice were subjected to RNA-seq analysis. IL-2c (IL-2- α -IL-2Ab complex) injection was performed using the same method as Fig. 7f and 7g. Bioinformatics analysis was performed using the same methods described in the Methods section for analyzing RNA-seq of splenic Tregs. (a-c) RNA-seq analysis performed on LI Tregs sorted from Ctrl or KO mice has been described in the manuscript. (d and e) RNA Pol II CUT&Tag analysis was performed with splenic Tregs nuclei of IL-2c-treated Ctrl or KO mice has been described in the manuscript. (a) Correlation analysis was performed on the fold change (Ctrl/KO) of mRNA expression in intestinal and IL-2-treated splenic Tregs on the differentially expressed genes identified by LI RNA-seq. (d and e) Correlation analysis was performed on significantly changed genes identified by IL-2c-treated SP Treg RNA-seq using

Log₂ (fold change) (Fc) of gene mRNA expression and average Log₂ (fold change) (Fc) of Pol II peaks distributed at the promoters (d) and Pol II peaks distributed at gene bodies (e) using the Methods described for the analysis in Fig. 7f and 7g. (a and d-e) Each dot represents one gene. Percentages indicate proportions of genes distributed in each quadrant. R is Pearson correlation coefficient. (b and c) GSEA analysis on IL-2c-treated SP Treg RNA-seq data using the custom gene sets, which are the top 500 decreased genes (b) and top 500 increased genes (c) in Setd2-deficient intestinal Tregs respectively. (f) Splenocytes from Ctrl or KO mice were cultured with MG132 (0.5uM), Chloroquine (10uM) or MG132 (0.5uM) plus Chloroquine (10uM) for 12h before cells were harvested for analysis. DMSO was used as vehicle control. The expression of GATA3 gated on Tregs (CD4⁺Foxp3⁺) was analyzed by flow cytometry. (g) Splenocytes from Ctrl or KO mice were cultured with the presence of cycloheximide (150ug/ml) before collected for analysis at indicated time points. The expression of GATA3 gated on Tregs (CD4⁺Foxp3⁺) was analyzed by flow cytometry. Mean fluorescence intensity (MFI) for GATA3 in Tregs was normalized to that of 0h. (f and j) Data are representative of 2 pairs of littermate mice. (g) Data are means ± SEM.

Minor concerns

1. The abstract needs to be reorganized to better describe the current knowledge and the key findings from the study.

Author response: We thank Reviewer 1 for the suggestion. Due to word limit (150 words) of the abstract, we couldn't expand more details about our findings. We would like to keep the current organization of the abstract to ensure the accuracy of the key conclusions of our findings. We have modified some of the descriptions according to the recommendations from the language editing service.

2. Lines 79+85: Cytokines (TGFb, IL-2 and IL-35) are not environmental factors.

Author response: We thank Reviewer 1 for the correction. We have changed the description of "environmental factors" to "factors such as TGF-β and retinoic acid from the tissue microenvironment". (line 80-81) We also deleted the "environmental" in previous description of "environmental triggers inducing GATA3". (line 85)

3. Line 103: The authors need to explain the term 'conservative regulation'.

Author response: We thank Reviewer 1 for the correction. We have modified the description as following, "*We demonstrated that the support of GATA3 expression in Tregs by Setd2 was conserved in both mice and humans, ...*" (line105-106)

4. Line 120-121: The sentence "...confirming that Setd2 is the dominant H3K36 methyltransferase mediating H3K36me3 in Tregs" does not make any sense since the authors have stated in lines 96-97 that "Setd2 is the only known mammalian histone H3 lysine 36 (H3K36) methyltransferase mediating H3K36me3".

Author response: We thank Reviewer 1 for pointing this out. We think that these two statements are not conflict with each other. There could be unknown and undefined H3K36 methyltransferase mediating H3K36me3 in mammalian cells. Such a case may be present in Tregs since H3K36me3 in Tregs haven't been investigated before. Therefore, examining H3K36me3 in Setd2-deficient and sufficient Tregs is required before making the conclusion of "Setd2 is the dominant H3K36 methyltransferase mediating H3K36me3 in Tregs".

5. Line 124 and Figure 1C+D: The authors should better use 'effector/memory' instead of 'activated' to describe the phenotype of CD44⁺CD62L^{low} cells.

Author response: We would like to thank Reviewer 1's suggestion. We agree with the Reviewer 1 and have modified the description of "activated" to "effector/effector memory" cells throughout the manuscript (also in figures and legends). We have also changed the description of "memory T cells" marked as CD62L^{high}CD44⁺ to "central memory T cells".

6. Line 135: The term 'periphery Tregs' is not commonly used and the authors should better explain which Tregs they are referring to.

Author response: We appreciate Reviewer 1's suggestion. We have specified the description of "periphery Tregs" throughout the manuscript.

"periphery" was modified to "secondary lymphoid organs (SLOs)". (line 145)

"periphery" was modified to "spleen and ILN". (line 149)

"periphery" was modified to "SLOs". (line 323)

"periphery" in the subtitle of Fig. 1 was removed. (line 530)

"periphery" in the first subtitle of the Results section was removed due to shortening of the subtitles. (line 114)

7. Line 148: Rephrase sentence as 'fate conversion of Tregs to non-Tregs' could only be studied with the help of a fate reporter that is missing in the present study.

Author response: We would like to thank Reviewer 1 for the correction. We have modified the description as "*Furthermore, increased proportions of Tconvs derived from Foxp3^{Cre-YFP}Setd2^{ff} donors, which might happen if Setd2-deficient Tregs lost Foxp3 expression and converted to Tconvs, was not observed (Fig. 2c).*" (line169-171)

8. Line 149: Replace 'ratio' by 'frequency'.

Author response: We have changed "ratios" to "frequencies". (line 167)

9. Line 167: What is meant with 'extreme effort'?

Author response: We apologize for causing this confusion. We have deleted this description and modified the description as "blockade of cell apoptosis and necrosis..." (line202).

10. Figure S3: It is difficult to draw any conclusions from the suppression assay since the conditions of the assay (e.g. cell number or stimulus) need to be optimized to enable complete inhibition of naïve T cell proliferation if Tregs were added at a 1:1 ratio.

Author response: We thank Reviewer 1 for pointing this out. We have optimized protocol for T cell suppression assay by analyzing T cell proliferation at an earlier time point (on day 3 of culture). A higher suppressive efficiency of Tregs to inhibit the proliferation of T responder cells was observed. We found that *Setd2*-deficient Tregs from the large intestine had reduced suppressive capacity to inhibit the proliferation of T responder cells, whereas the suppressive function of splenic *Setd2*-deficient Tregs was normal (Supplementary Fig. 3a). We have corrected this conclusion in the Results section as “*In vitro, Setd2-deficient splenic Tregs had a similar ability, whereas Setd2-deficient intestinal Tregs had a reduced competency, to suppress conventional T-cell proliferation compared with control Tregs (Supplementary Fig. 3a).*” (line 208-210)

11. Line 221-222: The authors should consider rephrasing since experimental evidence for this specific assumption is lacking and *Setd2*-deficiency could also simply affect the differentiation of Treg subsets.

Author response: We thank Reviewer 1 for raising this point. We have deleted this sentence from our manuscript (line 282 in the revised manuscript). It is very likely that *Setd2* also regulate the proliferation, survival, differentiation and function of ROR γ ⁺ Tregs under other context (such as during inflammation). This requires further investigation. Since we didn't focus on studying the regulation of ROR γ ⁺ Tregs by *Setd2*, we try to be careful with any assumptions and comments.

12. Figure 4 and S4: The authors might want to consider using the exact phenotypes instead of the terms ‘DP’ and ‘DN’ since they are frequently used for the description of thymocyte subsets.

Author response: We would like to thank the Reviewer for this advice. We feel that the DP, DN and SP terms are helpful for people to easily catch the Treg populations we are talking about instead of relying on reading the “+” and “-” superscript. This nomenclature also simplifies our labeling on the figures (Fig. 5i and Supplementary Fig. 5f). The terms of DP and SP have also been applied for describing the intestinal dendritic cells^{8,9}. As we have described the populations as DP Tregs, DN Tregs, GATA3 single Tregs and ST2 single Tregs without describing anything related to the thymus in Fig. 4 and 5, we think this wouldn't cause misunderstanding.

13. Figure 5L: It is not obvious why the authors have depicted a ‘normalized ratio’ for the frequency of GATA3+ST2+ cells among Helios+ Tregs.

Author response: We performed this normalization because we observed variations among batches of experiments perhaps due to variations of microbiota among different litters or

variations in the robustness of antibody staining from different batches of experiments. We presented the original data without normalization, which was with less robust statistics but support the same conclusion, as below for the Reviewer's reference (Reviewer Fig. 5).

Reviewer Fig. 5 Original of data without normalization for manuscript Fig. 5p (Figure 5L in the previous version of the manuscript).

Reviewer #2 (Remarks to the Author):

This paper reports the effect of a deletion of Setd2 in Tregs.

1. In figure 1, the authors report a statistically significant reduction of Tregs in LI and MLN, accompanied by an increase in activated CD4 and CD8 T cells. As a “Treg defect” is the starting observation, it would be interesting to analyze the phenotype of Setd2-Tregs in more details. What is their naïve/memory status?

Author response: We would like to thank Reviewer 2 for raising this question. We have analyzed the naïve/memory status of Tregs in *Foxp3^{Cre-YFP}Setd2^{ff}* and control mice (Fig. 1j-11). We found that the percentage of central memory (CD44⁺CD62L^{high}) cell in Tregs was reduced in the large intestine (Fig. 1l). Percentages of naïve cells were also decreased or showed a trend towards a decrease in Tregs, reflecting a loss of the quiescence status similar to the CD4⁺Tconv (Fig. 1k). Percentages of effector/effector memory cells in Tregs were comparable in the spleen, ILN, MLN and LI of *Foxp3^{Cre-YFP}Setd2^{ff}* mice compared to controls (Fig. 1j).

Fig. 1j-11 haven been added to the revised manuscript. In the Result section, we have added “For Tregs, the percentages of naïve cells were also decreased in spleen, MLNs and large intestine, and percentages of central memory cells were decreased in the large intestine, possibly reflecting a loss of the quiescence status of the Tregs (Fig. 1j-11)” (line 137-140)

2. How are expressed functional markers such as CD25, CTLA4, GITR...?

Author response: We thank Reviewer 2 for raising this question. We have analyzed the expression of CD25, CTLA-4 and GITR expression by splenic and intestinal Tregs from *Foxp3^{Cre-YFP}Setd2^{ff}* mice and controls. We observed slightly but significantly reduced CD25 (IL-2R α) expression in splenic and intestinal Tregs of *Foxp3^{Cre-YFP}Setd2^{ff}* mice

(Supplementary Fig. 5n and 5o), whereas the expression of CD122 (IL-2R β) in splenic Setd2-deficient Tregs was increased (Supplementary Fig. 5p and 5q). Since we have shown in the previous version of the manuscript that the level of phosphorylated STAT5 was comparable in tTregs from the spleen or the intestine of *Foxp3^{Cre-YFP}Setd2^{ff}* mice and control mice (Supplementary Fig. 5r and 5s), we think that the direct signal downstream of IL-2 is unlikely to be breached by deletion of Setd2 in Tregs. Please also see author response to major Question 12 from Reviewer 1.

We found no difference in GITR expression in Tregs from spleen or large intestine of *Foxp3^{Cre-YFP}Setd2^{ff}* mice compared with controls (Supplementary Fig. 4e). We observed increased expression of CLTA-4 in Tregs from the spleen but not large intestine of *Foxp3^{Cre-YFP}Setd2^{ff}* mice (Supplementary Fig. 4f). As CLTA-4 has been indicated to support the function of Tregs and we observed reduced function in Setd2-deficient Tregs¹⁰, we think that the changed expression of CLTA-4 is less likely to participate in the regulation of Tregs by Setd2.

Supplementary Fig. 4e, 4f, and 5n-5q have been added. And these findings have been described in the Results section of the revised manuscript. (line 267-269, 350-353)

3. How is their suppressive activity in vitro.

Author response: We would like to thank the reviewer for the question. We have optimized the *in vitro* Treg suppression assay and found that Setd2-deficient Tregs from the large intestine but not the spleen showed a reduced suppressive capacity of inhibiting the proliferation of responder T cells *in vitro* (Supplementary Fig. 3a). Please also see author response to minor concerns question 10 from Reviewer 1.

4. In figure 2, the authors report a functional disadvantage of Setd2- Tregs that they then link to a poor response to IL-2. Given the pleiotropic consequence of the Setd2- KO on the gene expression as analyzed in fig 4, it would be interesting to analyze if this poor response to IL-2 could be link to modifications of the IL-2 receptor activation pathway, for example by analysing IL-2-induced Stat5 phosphorylation.

Author response: We thank Reviewer 2 for this question. IL-2-induced STAT5 phosphorylation in Tregs from the spleen and large intestine of *Foxp3^{Cre-YFP}Setd2^{ff}* mice and control mice has been analyzed in the previous version of the manuscript. The data have been presented as Supplementary Fig. 5r and 5s, which showed that the level of phosphorylated STAT5 was comparable in tTregs from the spleen or the intestine of *Foxp3^{Cre-YFP}Setd2^{ff}* mice and control mice. Therefore, the direct signal downstream of IL-2 is unlikely to be breached by deletion of Setd2 in Tregs (line353-356).

5. Fig3 convincingly show that Setd2- Tregs are less efficient in controlling the induction of colitis by Teff.

Author response: We thank the Reviewer for the positive comment.

6. In figure 4, the authors analyzed the transcriptome of Setd2- Tregs. It would have been useful to first show a global comparison between Setd2- Tregs and plain Tregs using for example a PCA, and then to show a volcano plot with p values for variations... This would have helped to more convincingly point to Gata3 and ST2 expression.

Author response: We would like to thank the Reviewer's advice. As Reviewer 1 also suggested us to perform RNA-seq on splenic Tregs (see response to major Question 9 from Reviewer 1), we performed principal component analysis (PCA) on Tregs isolated from both the spleen and large intestine of *Foxp3^{Cre-YFP}Setd2^{ff}* mice and controls. We found that PC1 distinguished splenic Tregs from intestinal Treg cells, and PC2 separated Setd2-deficient from Setd2-sufficient Tregs (Supplementary Fig. 4b). This suggests that the major difference in transcriptome profiles from splenic and intestinal Tregs of *Foxp3^{Cre-YFP}Setd2^{ff}* mice and controls could be explained by the deficiency of Setd2 and tissue microenvironment.

Since more than 1000 genes were found to be differentially expressed by intestinal Setd2-deficient Tregs compared with controls, we have screened candidate genes by overlapping with colonic Treg signatures (Supplementary Fig. 4g). We then generated volcano plot based on the overlapped genes in Supplementary Fig. 4g (Supplementary Fig. 4h). *Il1rl1* and other genes of interests have been highlighted for visualization based on fold change and P values as a supplement to Fig. 4c.

In this study, we found no difference in GATA3 mRNA expression by Setd2-deficient Tregs compared with controls (Fig. 7e and 7i). We suspect that GATA3 may be regulated by Setd2 at the translational level. Please also see response to major Question 13 from Reviewer 1.

Supplementary Fig. 4b and 4h have been added. These findings have been added to the Results section of the revised manuscript. (line261-263 and 272)

7. In figure 5, in F to I the representation is quite misleading: it is invalid to draw line between the values obtained from different mice.

Author response: We thank the Reviewer for this correction. We originally used joint lines to indicate cells from littermate mice from the same batch of experiment. We agree that it was inappropriate to draw lines between values obtained from different mice. We have removed the lines and the data are now presented as means \pm SEM. Fig. 5j-5m have been updated.

8. In figure 6, the defect of Setd2- Tregs does not induce any pathology. Thus, how significant is this defect?

Author response: We would like to thank Reviewer 2 for this comment. Although we didn't observe any pathological changes in the intestine of *Foxp3^{Cre-YFP}Setd2^{ff}* mice under steady state, we found that IL-4 production by CD4⁺ T cells and serum IgE were elevated. After IL-33 treatment, IL-4, IL-5 and IL-13 production by CD4⁺ T cells were increased in addition to elevated serum IgE in *Foxp3^{Cre-YFP}Setd2^{ff}* mice. We suspect that *Foxp3^{Cre-YFP}Setd2^{ff}* mice

might have increased capacity to control parasite infection considering Th2 responses have been known to be important for defense against extracellular pathogens¹¹. Previous studies have shown that IL-33 promotes IgE production through an IL-4-dependent mechanism and exacerbates skin anaphylaxis by increasing mast cell degranulation activity¹². Therefore, we suspect that *Foxp3^{Cre-YFP}Setd2^{ff}* mice may be more prone to skin anaphylaxis. As we have demonstrated Setd2 expression in intestinal Tregs is important for inhibiting colitis and may also be relevant to the pathogenesis of colorectal cancer, we didn't further extend the scope of significance into other disease contexts such as intestinal infections and allergy. They will be of interest to be investigated in the future.

Reviewer #3 (Remarks to the Author):

This is an interesting study that addresses the role of the histone H3K36 methyltransferase Setd2 in regulatory T cells (Tregs). Deficiency for this enzyme in the Treg lineage results in an unexpectedly selective phenotype, that seems to especially affect a subset of thymic derived Tregs that express the transcription factor Gata3 and the cytokine receptor ST2 in the intestinal lamina propria and the local draining lymph nodes. The selectivity of this phenotype is unexpected, because the enzyme is expressed in all Treg cells.

On the whole, the analysis of the mice was done well. I have only minor technical comments, which I will list below. In my view, a weakness of the article is that it remains rather descriptive and that it is not all that clear what new insights we have obtained from these results. No clear rationale is provided why the role of SetD2 should be examined in Tregs and there is no clear explanation why this enzyme, that one would expect to have a rather general role in control of gene expression, would have such a very selective role in regulation of just a particular subtype of Tregs. I believe that at least some discussion of these questions and possible explanations would make the article more interesting. Some discussion is also necessary regarding the finding in Figure 2 that there is a more general survival defect in the Tregs upon adoptive transfer into immunodeficient mice. How does this result fit with the very selective phenotype during steady state in mice? Is survival in the adoptive transfer model also dependent on Gata3 and/or ST2?

Author response: We would like to thank Reviewer 3 for the positive comments and critiques. To address why the role of Setd2 should be examined in Tregs, we have added the following text in the Discussion section: "*In Tregs, the expression of these epigenetic modulators could alter with the status of cell activation or under pathological conditions. For example, Ezh2 mediating histone H3K27me3 modification is highly expressed by Tregs and Ezh2 is upregulated in Tregs with CD28 mediated co-stimulation¹³. Recently, the protein stability of Ezh2 has been shown to be disturbed by Setd2¹⁴, the expression of which has been found to be decreased in epithelial cells during colitis¹. Furthermore, Nsd2 mediating histone H3K36me2 modification has been associated with function of Tregs to maintain fetal-maternal tolerance¹⁵.*

These findings indicate a critical role of the epigenetic modulators affecting histone modifications in regulating the function of Tregs and tissue homeostasis.” (line 442-450)

To speculate why Setd2 determines a specific transcriptional profile of Tregs while Setd2 deposits H3K36me3 broadly in actively transcribed genes, we have added the following text in the Discussion section: “*Considering Setd2-mediated H3K36me3 modification is broadly observed in actively transcribed genes, this specific transcriptional program imprinted by Setd2 in Tregs is intriguing. We speculate that this specific manner of regulation is because Setd2 mainly affects the activities of cis-regulatory elements distributed in the promoters and gene bodies, where H3K36me3 is typically deposited. However, gene transcription may be determined by a group of cis-regulatory elements, including distal enhancers and enhancers upstream of a gene locus which may not be regulated by Setd2, in a cooperative way. Furthermore, Setd2 may exhibit H3K36me3-independent regulations by directly mediating protein methylation and stability^{14, 16}. These properties of Setd2 may allow it to imprint specific transcriptional programs of Tregs.*” (line 453-462)

To speculate whether loss of Treg homeostasis (shown in Fig. 2) in the absence of Setd2 is relevant to the deficiency of GATA3 and ST2, the following text have been added in the Discussion section: “*We showed that Setd2-deficient Tregs had increased fate conversion in CD45RB^{high}T cell-induced colitis and during homeostatic expansion in Rag2^{-/-} mice. Previous studies indicate that GATA3 and ST2 are important for the maintenance of Treg stability during inflammation^{17, 18}. Therefore, it is likely that Setd2 prevents fate conversion of Tregs under inflammatory conditions through sustaining GATA3 and ST2. We also found that Setd2-deficient Tregs had increased apoptosis, and blocking cell death partially rescued defective expansion of Setd2-deficient Tregs in response to IL-2. So far, not much evidence supports the theory that GATA3 or ST2 could promote Treg survival. We speculate that Setd2 may support the survival of Tregs through GATA3/ST2-independent mechanisms*”. (line483-491) New data mentioned in this discussion are provided as Fig. 2f, 2g and Supplementary Fig. 3b-3e in the revised manuscript. For details, please see author response to major Question 6 and 7 from Reviewer 1.

Minor points:

1. Figure 1: it would be good to provide absolute cell numbers in addition to the percentages shown here.

Author response: We would like to thank Reviewer 3 for this suggestion. As a note, we changed the nomenclature of “effector T cells, Teffs” to “T conventional cells (Tconvs)” based on Reviewer 1’s suggestion (See response to major question 2 from Reviewer 1). We found that there were no difference in numbers of Tconvs or numbers of CD4⁺ T cells cells from the spleen, MLN, ILN and large intestine (LI) of *Foxp3^{Cre-YFP}Setd2^{fl/fl}* mice compared with control mice (Supplementary Fig. 1g). Therefore, the percentages of cells were supposed to be consistent with absolute numbers of cells. A trend towards a decrease in numbers of Tregs was observed in the MLNs and large intestine of *Foxp3^{Cre-YFP}Setd2^{fl/fl}* mice although this didn’t reach a statistical significance (Supplementary Fig. 1g). These data are

consistent with our observation that percentages of Tregs were decreased in the MLN and large intestine of *Foxp3^{Cre-YFP}Setd2^{fl/fl}* mice (Fig. 1n), and are therefore support our conclusion that “Setd2-deficient Tregs have reduced capability to inhibit T cell activation in the secondary lymphoid organs (SLOs) and the intestine” and “Setd2 is important for the homeostasis of Tregs in the intestine and gut associated lymphoid tissues”.

Supplementary Fig. 1g has been added, and this finding has been added in the Results section (line 141-143, 150-153).

2. Figure 4: I am missing the point about the H3K36me3 levels show in Figure 4C. It seems to me that there is absolutely no correlation between these levels and expression of the genes that are dependent on SetD2. This needs clarification.

Author response: We apologize for causing this confusion. Representative genes shown in Fig. 4c are colonic Treg signature genes. As we have shown in Fig. 4a, the majority of colonic Treg signature genes have higher H3K36me3 signal in their locus (as suggested by the cumulative curve) in colonic Tregs than in splenic Tregs. This has also been reflected in the list of genes in Fig. 4c. We have clarified this point in the Results section: “*Among the differentially expressed genes in KO intestinal Tregs compared with control intestinal Tregs, 57 increased and 53 decreased genes were colonic Treg signature genes (Supplementary Fig. 4g, 4h and Supplementary Table 3), the majority of which had higher H3K36me3 signal in their gene locus in colonic Tregs than that in splenic Tregs (Fig. 4a and 4c)*”. (line269-273)

3. Figure 4S: the text claims that there is no increase in IL17+ Tregs. However, it looks to me like there is a clear trend towards higher IL17+ Tregs. While the difference in these data does not reach statistical significance, that does not prove that there is no increase.

Author response: We thank Reviewer 3 for noticing this point. We actually claimed that IL-17 was increased in Tregs in the previous version of the manuscript: “Flow cytometry analysis confirmed the increased expression of IFN- γ and IL-17 in intestinal Setd2-deficient Tregs identified by RNA-seq..” (line 284-286).

4. Figure 6: injection of IL33 elicits a greater increase in ST2+ Th2 cells in the knock out mice than in the wild type. I wonder what the right interpretation of this result is. One possibility is that the SetD2-dependent Treg population inhibits the generation of ST2+ Th2 cells in steady state, such that there are more cells in the knock out mice with the capacity to respond to his cytokine and proliferate. A second possible explanation is that the SetD2-dependent Treg population directly regulates the response by conventional T cells to IL33. Finally, it is possible that Set2D-dependent Tregs transdifferentiate into ST2+ Th2 cells. There are sophisticated lineage tracing approaches that can address this latter possibility. However, I realise that doing those would be a large time investment. I would be satisfied with a mixed chimera experiment, which will at least address whether

the effect is cell intrinsic (which would argue in favour of transdifferentiation) or not (which would support the other two possibilities).

Author response: We would like to thank Reviewer 3 for raising this point. Bone marrow cells from Thy1.1⁺ *Foxp3^{cre-YFP}Setd2^{ff/+}* mice and Thy1.2⁺ *Foxp3^{cre-YFP}Setd2^{ff}* mice were mixed at a 1:1 ratio and transferred to lethally irradiated *Rag1^{-/-}* mice, which were sacrificed for analysis 6-8 weeks (for analysis under steady state) and 12 weeks (for analysis upon IL-33 treatment) after transfer (Review Fig. 6a). We observed significantly lower ST2⁺ Tregs and GATA3⁺ Tregs derived from *Foxp3^{cre-YFP}Setd2^{ff}* donors compared with that derived from *Foxp3^{cre-YFP}Setd2^{ff/+}* donors both under steady state and upon IL-33 injection (Review Fig. 6b, 6c, 6f and 6g). This suggests that *Setd2* supports the homeostasis of GATA3⁺ Tregs and ST2⁺ Tregs through a cell intrinsic manner. Nonetheless, no significant difference was observed in proportion of cells derived from *Foxp3^{cre-YFP}Setd2^{ff}* and *Foxp3^{cre-YFP}Setd2^{ff/+}* donors contributing to the ST2⁺ Tconv pool or the GATA3⁺ Th2 cell pool under steady state (Review Fig. 6d and 6e) or after IL-33 treatment (Review Fig. 6h and 6i). There was a trend towards a decrease in proportion of *Foxp3^{cre-YFP}Setd2^{ff}* donor-derived cells among ST2⁺ Tconv cells under steady state (Review Fig. 6e).

Using another strategy of analysis, we observed no significant difference in percentages of GATA3⁺ cells or ST2⁺ cells among Tconvs analyzed from *Foxp3^{cre-YFP}Setd2^{ff}* donors compared with that analyzed from control donors after IL-33 treatment (Review Fig. 6j and 6k). Together, these data indicate that the increased Th2 cells under steady state or upon IL-33 treatment in *Foxp3^{cre-YFP}Setd2^{ff}* mice was less likely due to transdifferentiation of Tregs to Th2 cells or ST2⁺Tconvs, but more likely due to defective function of Tregs to inhibit the expansion of Th2 cells or ST2⁺ Tconvs in *Foxp3^{cre-YFP}Setd2^{ff}* mice.

Review Fig. R6 Analysis of Th2 cells and ST2⁺Tconvs in bone marrow chimeric mice (a-k) Mixed bone marrow chimeric mouse was generated with bone marrows mixed at 1:1 ratio from donors with indicated genotypes as shown in (a). (a) Schematic strategy of generating bone marrow chimeric mouse. (b-e) Large intestinal laminal propria lymphocytes of recipient mice were isolated 6-8 weeks after bone marrow transfer for flow cytometry analysis. (f-k) 12 weeks after bone marrow transfer, recipient mice were injected with 500ng of IL-33 daily for 3 consecutive days. Large intestinal laminal propria lymphocytes of recipient mice were isolated for flow cytometry analysis. (b-i) Percentages of cells derived from *Foxp3^{cre-YFP}Setd2^{ff}* donors or *Foxp3^{cre-YFP}Setd2^{ff/+}* donors distinguished by Thy1.2 and Thy1.1 gated on GATA3⁺Tregs (CD4⁺Foxp3⁺GATA3⁺) (b and f), ST2⁺Tregs (CD4⁺Foxp3⁺ST2⁺) (c and g), GATA3⁺Th2 cells (CD4⁺Foxp3⁻GATA3⁺) (d and h), ST2⁺Tconvs (CD4⁺Foxp3⁻ST2⁺) (e and i) respectively were analyzed by flow cytometry. (j-k) Percentages of GATA3⁺ cells (j) and ST2⁺ cells (k) in Tconvs (CD4⁺Foxp3⁻) from indicated

donor origin are shown. (b-e) Representative of 2 independent experiments. (f-k) Data were pooled from 2 bone marrow chimeric mice. (b-k) Each pair of connected line indicates data obtained from the same host mouse.

5. Figure 6M/N: I am not very convinced about the elevated number of Th2 cells in the cancer samples. This is mostly because the Gaat3 staining is very weak. I think these data need to be strengthened or taken out of the manuscript. One way to strengthen the results would be to restimulate the cells and measure Th2 cytokines. Alternatively, there might already be single cell RNAseq data publicly available that can be used to address this point.

Author response: We would like to thank Reviewer 3 for raising this point. We stimulated lamina propria lymphocytes from cancerous and non-cancerous tissues from patients with colorectal cancer with PMA and ionomycin. Expression of IL-5 and IL-13 were then analyzed by flow cytometry. IL-5 expressed by Tconvs was barely detected. Interestingly, IL-13 expression in Tconvs from cancerous tissues was lower compared with that from non-cancerous tissues (Fig. 6p). The data is consistent with reduced level of Th2 cells identified by GATA3 staining in cancerous tissues compared with non-cancerous tissues.

Fig. 6p has been added to the revised manuscript. In the Results section, we have added “*Meanwhile, percentage of Th2 cells, as well as percentage of IL-13⁺ cells in Tconvs, was significantly lower in cancerous tissues (Fig. 6o and 6p)*”. (line378-379)

6. Finally, there is no information about the statistical tests used in any of the figures. This seems especially relevant for Figure 2, 3 5, 6, where sometimes multiple comparisons were made which should be analyzed with something like ANOVA.

Author response: We would like to thank Reviewer 3 for raising this question. We stated statistical tests used in this study in the “Statistical methods” section in the previous version of the manuscript (line1022-1030 in this revised version of manuscript). In this revised version of manuscript, we modified the statistical analysis methods by changing some analyses using “Mann-Whitney test” to “unpaired Student’s t-test” as long as the data pass “Klomogorov–Smirnov” or “Shapiro-Wilk” normality tests. The change of statistical methods didn’t change any of our original scientific conclusions. All the statistical tests have been described in detail in the “Statistical methods” section. Exact P values have been added in all the Figures to comply with the Nature Communications “Reporting Summary” document for “Statistics”. In data with more than 3 groups, we actually made multiple comparisons between each two groups using Student’s t-test or Mann-Whitney test. Therefore, we didn’t use ANOVA in our study.

References:

1. Liu M, *et al.* The histone methyltransferase SETD2 modulates oxidative stress to attenuate experimental colitis. *Redox biology* **43**, 102004 (2021).

2. Yuan H, *et al.* Histone methyltransferase SETD2 modulates alternative splicing to inhibit intestinal tumorigenesis. *The Journal of clinical investigation* **127**, 3375-3391 (2017).
3. Sallusto F, Geginat J, Lanzavecchia A. Central memory and effector memory T cell subsets: function, generation, and maintenance. *Annual review of immunology* **22**, 745-763 (2004).
4. Komatsu N, Mariotti-Ferrandiz ME, Wang Y, Malissen B, Waldmann H, Hori S. Heterogeneity of natural Foxp3+ T cells: a committed regulatory T-cell lineage and an uncommitted minor population retaining plasticity. *Proceedings of the National Academy of Sciences of the United States of America* **106**, 1903-1908 (2009).
5. Duarte JH, Zelenay S, Bergman ML, Martins AC, Demengeot J. Natural Treg cells spontaneously differentiate into pathogenic helper cells in lymphopenic conditions. *European journal of immunology* **39**, 948-955 (2009).
6. Zhou L, *et al.* Innate lymphoid cells support regulatory T cells in the intestine through interleukin-2. *Nature* **568**, 405-409 (2019).
7. Delacher M, *et al.* Precursors for Nonlymphoid-Tissue Treg Cells Reside in Secondary Lymphoid Organs and Are Programmed by the Transcription Factor BATF. *Immunity* **52**, 295-312 e211 (2020).
8. Flores-Langarica A, *et al.* CD103(+)CD11b(+) mucosal classical dendritic cells initiate long-term switched antibody responses to flagellin. *Mucosal immunology* **11**, 681-692 (2018).
9. Mortha A, *et al.* Microbiota-dependent crosstalk between macrophages and ILC3 promotes intestinal homeostasis. *Science* **343**, 1249288 (2014).
10. Wing K, Yamaguchi T, Sakaguchi S. Cell-autonomous and -non-autonomous roles of CTLA-4 in immune regulation. *Trends in immunology* **32**, 428-433 (2011).
11. Anthony RM, *et al.* Memory T(H)2 cells induce alternatively activated macrophages to mediate protection against nematode parasites. *Nature medicine* **12**, 955-960 (2006).
12. Komai-Koma M, *et al.* Interleukin-33 amplifies IgE synthesis and triggers mast cell degranulation via interleukin-4 in naive mice. *Allergy* **67**, 1118-1126 (2012).
13. DuPage M, *et al.* The chromatin-modifying enzyme Ezh2 is critical for the maintenance of regulatory T cell identity after activation. *Immunity* **42**, 227-238 (2015).

14. Yuan H, *et al.* SETD2 Restricts Prostate Cancer Metastasis by Integrating EZH2 and AMPK Signaling Pathways. *Cancer cell* **38**, 350-365 e357 (2020).
15. Zhang L, *et al.* Histone methyltransferase Nsd2 ensures maternal-fetal immune tolerance by promoting regulatory T-cell recruitment. *Cellular & molecular immunology* **19**, 634-643 (2022).
16. Chen K, *et al.* Methyltransferase SETD2-Mediated Methylation of STAT1 Is Critical for Interferon Antiviral Activity. *Cell* **170**, 492-506 e414 (2017).
17. Wohlfert EA, *et al.* GATA3 controls Foxp3(+) regulatory T cell fate during inflammation in mice. *The Journal of clinical investigation* **121**, 4503-4515 (2011).
18. Schiering C, *et al.* The alarmin IL-33 promotes regulatory T-cell function in the intestine. *Nature* **513**, 564-568 (2014).

Reviewer #1 (Remarks to the Author):

In the revised version of their manuscript, the authors have addressed all my concerns and modified the manuscript accordingly. Thank you. In my opinion, the manuscript is now suitable for publication in Nature Communications.

A few minor points should be changed in the final version:

- 1) In Suppl. Figure 2d, the authors show a purity >99%. Yet, in the corresponding text (line 180) they mention "higher than 95%". This should be harmonized.
- 2) Neither in the legend for Suppl. Figure 5k-m nor in the corresponding text in the results section, the age of the mice is mentioned. This missing information should be added.

Reviewer #2 (Remarks to the Author): Reviewer #2 Comments were mediated by reviewer #3

Anyway, I had a look at the response to reviewer 2. This reviewer did not have very serious comments to begin with and the authors have done a good job to address most of these points.

There is one thing that should still be addressed better, though: the response to comment 6 draws attention to the fact that Gata3 gene expression is apparently not reduced in SETD2 deficient Tregs, even though the percentages of Gata3+ Tregs (as measured by FACS) are clearly down. This is an important point, because the authors make a big case that SETD2 controls the reciprocal relationship between ST2 and Gata3. Especially because SETD2 is a known regulator of transcription, the immediate assumption of the reader is that Gata3 gene expression would be regulated by this factor, but this is apparently not so. This issue will likely confuse readers. Both reviewers 1 and 2 had questions about this, and now, so do I.

The authors try to explain the loss of Gata3 protein but not mRNA by invoking a mechanism based on translation, for which they do not provide evidence. I wonder whether the explanation is not simpler: that the SETD2 deficiency causes loss of the Gata3+ subset. Perhaps SETD2 regulates a critical factor for the differentiation or maintenance of these cells (ST2?).

For as far as I can see, the FACS data on the mouse Tregs can all be explained by a loss of the population. In principle, the loss of the population should also lead to loss of mRNA in the RNAseq experiment (even if this has nothing to do with direct control of the Gata 3 gene), but the authors say that Gata3 expression was not found to be diminished. I wonder whether perhaps the focus on genes that were reduced in both splenic and large intestinal Tregs SETD2 knock out Tregs made them "miss" the effect on Gata3. The effect on the Tregs is much more modest in the spleen and also there are fewer ST2+Gata3+ Tregs in the spleen. Is Gata3 expression lower in the comparison wt vs ko when looking at the LI only?

If the LI RNAseq data do not reveal a reduction in Gata3, it is also possible that the lost population is too small to allow detecting this loss: Figure 4j suggests that Gata3+Helios+ cells are a relatively small population even in the LI and that there is also a Gata3 single positive population of Tregs in the LI. Perhaps the loss of the population is just too small to pick up, especially if it is associated with an increase in the Gata3 single positive population (which seems to be the case, even if it is not significant).

The only result that suggests a more direct effect on Gata3 expression is the knock down in the human Tregs. However, the mRNA expression of Gata3 was not tested there, so perhaps in that setting SETD2 does regulate Gata3 gene expression or the cells are in the process of dedifferentiating and Gata3 is lost via some other mechanism.

I would not want to ask the authors to provide more data on this. However, I do think it is important that the authors discuss why they think that loss of the population is not the explanation

(if that is what they believe) or else discuss the pro's and con's of the competing interpretations (translation of Gata3 versus loss of a population). Such a discussion would help avoid confusion by the readers and would also alert them to the issue. They can then determine for themselves whether they agree with the interpretation provided by the authors.

Reviewer #3 (Remarks to the Author):

I have no further comments.

Response to Reviewers' comments

We are delighted to receive the positive comments from all the Reviewers on our manuscript entitled “Setd2 supports GATA3⁺ST2⁺ thymic-derived Tregs and suppresses intestinal inflammation” (NCOMMS-21-47229A), and we appreciate the Reviewers' further constructive suggestions. Corrections according to Reviewer 1's suggestions have been made. To address Reviewer 3's concern (mediating Reviewer 2's previous comments), we provided one Figure for Reviewer (Figure R1) and discussed additional mechanisms in the Discussion section. Point-by-point responses started with “author response” are provided below. Changes made to the revised manuscript are highlighted in yellow, except for that in the Supplementary information required to be prepared without highlights. We hope that our manuscript is now appropriate for publication by *Nature Communications*.

REVIEWERS' COMMENTS

Reviewer #1 (Remarks to the Author):

In the revised version of their manuscript, the authors have addressed all my concerns and modified the manuscript accordingly. Thank you. In my opinion, the manuscript is now suitable for publication in Nature Communications.

Author response: We thank Reviewer 1 for the positive comments.

A few minor points should be changed in the final version:

1) In Suppl. Figure 2d, the authors show a purity >99%. Yet, in the corresponding text (line 180) they mention "higher than 95%". This should be harmonized.

Author response: We thank Reviewer 1 for the correction. We have modified the description in the text as “higher than 99%” (line 179).

2) Neither in the legend for Suppl. Figure 5k-m nor in the corresponding text in the results section, the age of the mice is mentioned. This missing information should be added.

Author response: We thank Reviewer 1 for the suggestion. In the text, we have described the results as “We found no difference in percentages of PD-1⁻KLRG1⁻, PD-1⁺KLRG1⁻ or PD-1⁺KLRG1⁺ cells in splenic Tregs between *Foxp3*^{Cre-YFP}*Setd2*^{ff} mice and controls at the adult stage (Supplementary Fig. 5k and 5l)”.

Figure legends for Supplementary Fig. 5k-m were corrected as “(k-m) Flow cytometry analysis of KLRG1 and PD-1 expression in splenic Tregs from 6-12-week-old mice (n=4 per group).”

Reviewer #2 (Remarks to the Author): Reviewer #2 Comments were mediated by reviewer #3

Anyway, I had a look at the response to reviewer 2. This reviewer did not have very serious comments to begin with and the authors have done a good job to address most of these points.

Author response: We thank Reviewer 3 for the positive comments on our revised manuscript.

There is one thing that should still be addressed better, though: the response to comment 6 draws attention to the fact that Gata3 gene expression is apparently not reduced in SETD2 deficient Tregs, even though the percentages of Gata3+ Tregs (as measured by FACS) are clearly down. This is an important point, because the authors make a big case that SETD2 controls the reciprocal relationship between ST2 and Gata3. Especially because SETD2 is a known regulator of transcription, the immediate assumption of the reader is that Gata3 gene expression would be regulated by this factor, but this is apparently not so. This issue will likely confuse readers. Both reviewers 1 and 2 had questions about this, and now, so do I.

The authors try to explain the loss of Gata3 protein but not mRNA by invoking a mechanism based on translation, for which they do not provide evidence. I wonder whether the explanation is not simpler: that the SETD2 deficiency causes loss of the Gata3+ subset. Perhaps SETD2 regulates a critical factor for the differentiation or maintenance of these cells (ST2?).

For as far as I can see, the FACS data on the mouse Tregs can all be explained by a loss of the population. In principle, the loss of the population should also lead to loss of mRNA in the RNAseq experiment (even if this has nothing to do with direct control of the Gata 3 gene), but the authors say that Gata3 expression was not found to be diminished. I wonder whether perhaps the focus on genes that were reduced in both splenic and large intestinal Tregs SETD2 knock out Tregs made them “miss” the effect on Gata3. The effect on the Tregs is much more modest in the spleen and also there are fewer ST2+Gata3+ Tregs in the spleen. Is Gata3 expression lower in the comparison wt vs ko when looking at the LI only?

If the LI RNAseq data do not reveal a reduction in Gata3, it is also possible that the lost population is too small to allow detecting this loss: Figure 4j suggests that Gata3+Helios+ cells are a relatively small population even in the LI and that there is also a Gata3 single positive population of Tregs in the LI. Perhaps the loss of the population is just too small to pick up, especially if it is associated with an increase in the Gata3 single positive population (which seems to be the case, even if it is not significant).

The only result that suggests a more direct effect on Gata3 expression is the knock down in the human Tregs. However, the mRNA expression of Gata3 was not tested there, so perhaps in that setting SETD2 does regulate Gata3 gene expression or the cells are in the process of dedifferentiating and Gata3 is lost via some other mechanism.

I would not want to ask the authors to provide more data on this. However, I do think it is important that the authors discuss why they think that loss of the population is not the explanation (if that is what they believe) or else discuss the pro's and con's of the competing interpretations (translation of *Gata3* versus loss of a population). Such a discussion would help avoid confusion by the readers and would also alert them to the issue. They can then determine for themselves whether they agree with the interpretation provided by the authors.

Author response:

We appreciate Reviewer 3's constructive comments on the molecular regulation of GATA3 by *Setd2* in Tregs.

We checked the RNA-seq data and found that *Gata3* mRNA expression was decreased in *Setd2*-deficient intestinal Tregs (Figure R1a). The P value for RNA-seq statistical analysis was below the cut off value (0.05), so this change was not highlighted. *Gata3* mRNA expression was comparable between KO and WT splenic Tregs (Figure R1b). Notably, mRNA expression of *Il1rl1* was significantly reduced in KO Tregs isolated from both the intestine and the spleen (Figure R1c-1d). The above data suggest that compared with *Gata3*, *Il1rl1* is more profoundly affected at the transcription level by *Setd2* in Tregs.

Figure R1 mRNA expression of *Gata3* and *Il1rl1* in intestinal and splenic Tregs analyzed by RNA-seq

Large intestinal (LI) (a and c) and splenic (SP) (b and d) Tregs sorted from *Foxp3^{Cre-YFP}Setd2^{fl/fl}* (Ctrl) or *Foxp3^{Cre-YFP}Setd2^{fl/fl}* (KO) mice were subjected to RNA-seq analysis. Statistical analysis was performed by edgeR. Fragments per kilobase of exon model per million mapped fragments (FPKM) value of *Gata3* mRNA (a and b) and *Il1rl1* mRNA (c and d) are shown.

Although it seems that the changed expression of GATA3 could be interpreted at the mRNA level in the intestinal Tregs, we think that this could be partially attributed to proportional loss of Helios⁺Tregs (tTregs) in KO and correspondingly proportional enrichment of Helios⁻RORγt⁺ Tregs (Fig. 4e and 4f). Cell ontogeny analysis revealed that the reduction of

GATA3⁺ST2⁺ tTregs preceded the proportional increase of RORγt⁺ Tregs (Fig. 5e and 5g, Supplementary Fig. 5a), supporting the notion that GATA3⁺ST2⁺ tTregs are key targets of Setd2 not seconded to the change of RORγt⁺ Tregs. As a result, we utilized tTreg-enriched cells, the 2-week-old intestinal Tregs or the splenic Tregs from IL-2c-treated mice for the following CUT&Tag analyses, because the percentages of Helios⁻ Tregs in both of these cells were very low or were comparable between KO and controls (Supplementary Fig. 5e and 6b). This was to avoid possible confounding effects of proportionally enriched RORγt⁺ Tregs in the KO group.

We agree with the Reviewer that loss of GATA3⁺ tTregs is a central phenotype that we observed in KO mice, but this may not be sufficient to explain the reduced GATA3 protein (Supplementary Fig. 6b and 6c) expression with unchanged *Gata3* mRNA expression (Fig. 7i). Although the reduction of GATA3⁺ST2⁺ tTregs in the absence of Setd2 was less obvious in the spleen than that was observed in the intestine under steady state (Fig. 5a-5h), the decrease of GATA3⁺ tTregs in the spleen of KO mice was consistently clear upon IL-2c treatment (Supplementary Fig. 6b and 6c). With a loss of these GATA3⁺ tTregs in KO mice, we would expect at least a trend towards a reduction in *Gata3* transcription if it was regulated by Setd2, which was not the case as suggested by the H3K27ac/Pol II CUT&Tag analyses (Fig. 7d-7e). Based on the evidences above, we think that Setd2 may regulate GATA3 expression at the translational/post-translational level.

The Reviewer's insightful comments enlightened us on another possibility: Setd2 probably regulates GATA3 transcription within a specific subpopulation of Tregs. The cells used for H3K27ac and Pol II CUT&Tag analyses were enriched for tTregs yet were heterogenous by containing the GATA3⁺ST2⁻ tTregs, GATA3⁺ST2⁺ tTregs, GATA3⁻ST2⁺ tTregs, GATA3⁻ST2⁻ tTregs and a small subset of Helios⁻ Tregs (Fig. 5e, Supplementary Fig. 5d and 6b). It is likely that transcription regulation of *Gata3* by Setd2 only occurs in one of these Treg subsets. For example, Setd2 may promote the induction of GATA3 from GATA3⁻ST2⁺ tTregs or GATA3⁻ST2⁻ tTregs by transcriptional mechanisms while not affecting the GATA3⁺ST2⁺ tTregs or GATA3⁺ST2⁻ tTregs. Alternatively, Setd2 may sustain the transcription of *Gata3* in the GATA3⁺ST2⁺ tTregs while not affecting other Treg subsets. In this scenario, current analyses using bulk Tregs (even the cells were enriched for tTregs) may not be sensitive enough to capture the transcriptional regulation of *Gata3* by Setd2 within a specific Treg subset. Analyses using reporter mice allowing purification of specific Treg subsets or single-cell multi-omics approaches will be needed to test this hypothesis.

We also would like to thank Reviewer 3's the comment on human Tregs. As Setd2 is a large protein, retrovirus expressing Setd2 shRNA only partially ablated Setd2 expression (Fig. 6q). Cell permeabilization followed by flow cytometry analysis was required to detect these Setd2-deficient cells, making further mRNA analysis very tricky.

In the Results section, we have added, "*although we cannot rule out the possibility that Setd2 may promote the GATA3 expression at the transcription level within a specific subpopulation of Tregs*". We also explained these possibilities in the Discussion section: "*Although GATA3*

protein expression was reduced in Setd2-deficient tTregs, we didn't observe decreased H3K27ac or Pol II accumulation marking active transcription in the Gata3 locus, nor did we find decreased Gata3 mRNA expression in IL-2c-treated Setd2-deficient splenic Tregs." "Further, it is also likely that the transcription regulation of Gata3 by Setd2 may occur within only a specific subpopulation of tTregs, such as the ST2⁺GATA3⁺ tTregs. As cells used for Pol II and H3K27ac CUT&Tag analyses were splenic Tregs from IL-2c-treated mice and 2-week-old intestinal Tregs respectively, these cells were enriched for Helios⁺ tTregs yet were heterogeneously including the ST2⁺GATA3⁺ tTregs, ST2⁺GATA3⁻ tTregs, ST2⁻GATA3⁺ tTregs, ST2⁻GATA3⁻ tTregs and a small proportion of Helios⁻ Tregs. Regulation of GATA3 by Setd2 in different subsets of Tregs is probably different, and Setd2-mediated molecular changes detected in one subset may be covered by the others in the H3K27ac and Pol II CUT&Tag analyses. This hypothesis remains to be further tested with tools allowing purification of specific tTreg subpopulations, or with multi-omics approaches at the single cell resolution."

Reviewer #3 (Remarks to the Author):

I have no further comments.

Author response: We thank Reviewer 3 for this response.